

# New insights into the Devonian sea spiders of the Hunsrück Slate (Arthropoda: Pycnogonida)

Romain Sabroux[1], Russell J. Garwood[2,3], Davide Pisani[1,4], Philip C. J. Donoghue[1] and Gregory D. Edgecombe[3]

[1] Bristol Palaeobiology Group, School of Earth Sciences, University of Bristol, Bristol, United Kingdom
[2] Department of Earth and Environmental Sciences, University of Manchester, Manchester, United Kingdom
[3] Natural History Museum, London, United Kingdom
[4] Bristol Palaeobiology Group, School of Biological Sciences, University of Bristol, Bristol, United Kingdom

Corresponding author
Romain Sabroux,
romain.sabroux@bristol.ac.uk

## ABSTRACT

**Background:** The sea spiders (Pycnogonida Latreille, 1810) of the Hunsrück Slate (Lower Devonian, ~400 million years ago) are iconic in their abundance, exquisite pyritic preservation, and in their distinctive body plan compared to extant sea spiders (Pantopoda Gerstäcker, 1863). Consequently, the Hunsrück sea spiders are important in understanding the deep evolutionary history of Pycnogonida, yet they remain poorly characterised, impacting upon attempts to establish a time-calibrated phylogeny of sea spiders.

**Methods:** Here, we investigated previously described and new material representing four of the five Hunsrück pycnogonids: *Flagellopantopus blocki* Poschmann & Dunlop, 2006; *Palaeoisopus problematicus* Broili, 1928; *Palaeopantopus maucheri* Broili, 1929; and *Pentapantopus vogteli* Kühl, Poschmann & Rust, 2013; as well as a few unidentified specimens. Using X-ray microtomography and Reflectance Transformation Imaging, we describe new fossils, provide evidence for newly revealed anatomical features, and interpret these data in comparison to extant species. We also reinterpret the previously published illustration of the (probably lost) holotype of *Palaeothea devonica* Bergström, Stürmer & Winter, 1980.

**Results:** We provide the first detailed description of the cephalic appendages of *Palaeoisopus problematicus* and revise the interpretation of the organisation of its ocular tubercle. Furthermore, we provide new insights into the structure of the legs and the proboscis of *Palaeopantopus maucheri*, the first description of the body of *Flagellopantopus blocki* and describe a new specimen of *Pentapantopus vogteli*, demonstrating that it had eight legs, in contrast to previous interpretations. We argue that, contrary to previous suggestions, *Palaeothea devonica* probably had a different body plan from extant pantopods. We discuss the ecological traits of the Hunsrück pycnogonids based on their morphological adaptations, and conclude that there is no compelling evidence of Pantopoda in the Devonian. Through comparative interpretation of the legs as well as general morphology, we can divide the Hunsrück pycnogonids into two morphological groups, while Pantopoda constitutes a third morphological group.

# INTRODUCTION

Sea spiders (Arthropoda: Pycnogonida) are enigmatic arthropods, in terms of their extant diversity (*Sabroux, Hassanin & Corbari, 2019*; *Brenneis et al., 2020*), biology (*Brenneis et al., 2017*, *2023*; *Lehmann, Heß & Melzer, 2017*; *Brenneis & Wagner, 2023*), phylogeny (*Arabi et al., 2010*; *Sabroux et al., 2017*; *Ballesteros et al., 2021*; *Sabroux, Corbari & Hassanin, 2023*) and their ecology (*Dietz et al., 2018*). The same is true of their fossil record which is perhaps best described as "few and far between" (Table 1). Only 11 to 13 species (two are still regarded as ambiguous) have been described, representing over 500 Myr (million years) of evolutionary history (*Bergström, Stürmer & Winter, 1980*; *Waloszek & Dunlop, 2002*; *Poschmann & Dunlop, 2006*; *Charbonnier, Vannier & Riou, 2007*; *Kühl, Poschmann & Rust, 2013*; *Rudkin et al., 2013*; *Sabroux et al., 2019*). There are long fossil hiatuses between the Devonian and Jurassic, and between the Jurassic and the present (see a review of known fossils in *Sabroux et al., 2019*). The description of the known fossils is also challenging. The few researchers who have conducted research on pycnogonid fossils have produced high quality work, but studies are typically of limited means (being primarily based on light microscopy and X-rays) and this has, in turn, limited interpretation.

Sea spiders are integral to understanding the early evolution of arthropods since they branch near the roots of Arthropoda and Chelicerata (*Martin et al., 2010*; *Rehm et al., 2011*; *Lozano-Fernandez et al., 2016*; *Ballesteros & Sharma, 2019*; *Ballesteros et al., 2021*; *Sabroux, Corbari & Hassanin, 2023*). Since extant pycnogonid species (which are united in the order Pantopoda; *Hedgpeth, 1954*; *Bergström, Stürmer & Winter, 1980*; *Bamber, 2007*) are highly derived and hardly comparable to other arthropods, fossil pycnogonids have the potential to inform on the origin of their unique bodyplan and its transformation from the primitive arthropod condition. Recent years have seen limited work on fossil sea spiders: four fossils species have been described in the last decade (*Kühl, Poschmann & Rust, 2013*; *Rudkin et al., 2013*; *Sabroux et al., 2019*), no comprehensive restudy has been published for more than 40 years (*Bergström, Stürmer & Winter, 1980*) and there are only two short reviews of their fossil record (*Bamber, 2007*; *Sabroux et al., 2019*).

Recently, *Ballesteros et al. (2021)* used the fossil record of sea spiders to undertake the first molecular clock analysis for Pycnogonida, using Jurassic fossils as well as the Silurian *Haliestes dasos Siveter et al., 2004* for calibration. The taxonomic assignment of the fossils used as calibration points has been called into question (*Sabroux, Corbari & Hassanin, 2023*), highlighting the need for a comprehensive review of the sea spider fossil record. To this end, the sea spiders of La Voulte-sur-Rhône have been redescribed, providing new observations and interpretations using X-ray microtomography and Reflectance Transformation Imaging (RTI; *Sabroux et al., 2023*). New material of the exquisite *Haliestes dasos* from the Herefordshire Lagerstätte was also recently reported (*Siveter et al., 2023*), excluding it from the crown-group Pantopoda (*Wolfe et al., 2016*; *contra*

**Table 1 List of sea spider fossil species and their current classification.** Asterisks (*) indicate the species for which we could access specimens for this study. See *Sabroux et al. (2019)* for an alternative representation and additional information.

| Locality | Species | Current classification (see *Bamber, El Nagar & Arango, 2024*) | Main references |
|---|---|---|---|
| Orsten (Upper Cambrian) | *Cambropycnogon klausmuelleri Waloszek & Dunlop, 2002* | Possibly Pycnogonida Latreille, 1810 (larval stage?) | *Waloszek & Dunlop (2002)* |
| Manitoba (Upper Ordovician) | *Palaeomarachne granulata Rudkin et al., 2013* | Possibly Pycnogonida Latreille, 1810 (moult?) | *Rudkin et al. (2013)* |
| Herefordshire (Lower Silurian) | *Haliestes dasos Siveter et al., 2004* | Nectopantopoda *Bamber, 2007* | *Siveter et al. (2004, 2023)* |
| Hunsrück Slate (Lower Devonian) | *\*Flagellopantopus blocki Poschmann & Dunlop, 2006* | Pantopoda Gerstäcker, 1863 *incertae sedis* | *Poschmann & Dunlop (2006)* |
| | *\*Palaeoisopus problematicus Broili, 1928* | Palaeoisopoda *Hedgpeth, 1978* | *Bergström, Stürmer & Winter (1980)* |
| | *\*Palaeopantopus maucheri Broili, 1929* | Palaeopantopoda *Broili, 1930* | *Bergström, Stürmer & Winter (1980)* |
| | *Palaeothea devonica Bergström, Stürmer & Winter, 1980* | Pantopoda *incertae sedis* | *Bergström, Stürmer & Winter (1980)* |
| | *\*Pentapantopus vogteli Kühl, Poschmann & Rust, 2013* | Pycnogonida Latreille, 1810 | *Kühl, Poschmann & Rust (2013)* |
| La Voulte-sur-Rhône (Middle Jurassic) | *Colossopantopodus boissinensis Charbonnier, Vannier & Riou, 2007* | Pantopoda Gerstäcker, 1863 | *Charbonnier, Vannier & Riou (2007), Sabroux et al. (2023)* |
| | *Palaeoendeis elmii Charbonnier, Vannier & Riou, 2007* | Pantopoda Gerstäcker, 1863 | *Charbonnier, Vannier & Riou (2007), Sabroux et al. (2023)* |
| | *Palaeopycnogonides gracilis Charbonnier, Vannier & Riou, 2007* | Pantopoda Gerstäcker, 1863 | *Charbonnier, Vannier & Riou (2007), Sabroux et al. (2023)* |
| Solnhofen (Upper Jurassic) | *Colossopantopodus nanus Sabroux et al., 2019* | Pantopoda Gerstäcker, 1863 | *Sabroux et al. (2019)* |
| | *?Eurycyde golem Sabroux et al., 2019* | Pantopoda Gerstäcker, 1863 | *Sabroux et al. (2019)* |

*Ballesteros et al., 2021*). Following on from these works, we now review what is probably the most remarkable, and in any case the most diverse sea spider fossil fauna known: the fossil pycnogonids of the Hunsrück Slate, Germany (*Broili, 1928*, *1929*, *1930*, *1932*; *Bergström, Stürmer & Winter, 1980*; *Poschmann & Dunlop, 2006*; *Kühl, Poschmann & Rust, 2013*).

Five species have been recorded from the Emsian (c.a. 400 Ma) Hunsrück Slate: *Palaeoisopus problematicus Broili, 1928*; *Palaeopantopus maucheri Broili, 1929*; *Palaeothea devonica Bergström, Stürmer & Winter, 1980*; *Flagellopantopus blocki Poschmann & Dunlop, 2006*; and *Pentapantopus vogteli Kühl, Poschmann & Rust, 2013*. *Palaeoisopus problematicus* was the first sea spider fossil to be described, initially interpreted as an isopod (hence its name) by *Broili (1928)* who mistook the antero-posterior axis, *i.e.* interpreting the animal's abdomen as a proboscis, and its chelifores as an abdomen. It was only later that *Broili (1932)* recognized *P. problematicus* to be a pycnogonid–a surprising intuition, given that he still did not restore the proper antero-posterior axis of the fossil. This taxonomic assignment was met with scepticism (*e.g.*, *Hedgpeth, 1954*), but when the

antero-posterior axis was revised correctly (*Dubinin, 1957*; *Lehmann, 1959*), the pycnogonid affinity of *P. problematicus* was widely accepted (*e.g.*, *Hedgpeth, 1978*). *P. problematicus* presents the typical features of sea spiders (*i.e.*, the three pairs of cephalic appendages, the proboscis, and the ocular tubercle), but the deviations it shows from the body plan of modern Pycnogonida are striking: its legs were flattened, with a variable number of podomeres depending on the body segments, a long, segmented abdomen that bore a terminal telson, and enigmatic ring-like structures associated with the base of its legs.

*Palaeopantopus maucheri* was discovered at about the same time as *P. problematicus*, but its identification as a sea spider was much less problematic (*Broili, 1929*): the fossil presents the typical eight cylindrical legs of sea spiders, cephalic appendages, and a characteristic sea spider-like outline; a proboscis was later identified (*Bergström, Stürmer & Winter, 1980*). Nevertheless, this fossil also shows several unique features, such as the apparent absence of a cephalic segment, a long, segmented abdomen, and similar ring-like structures at its leg bases as in *P. problematicus*. Discovered much later were *Palaeothea devonica*, which has been regarded as the oldest possible evidence of Pantopoda in the fossil record, the bizarre *Flagellopantopus blocki* with its long flagellum, and *Pentapantopus vogteli*, a purported 10-legged sea spider (*Bergström, Stürmer & Winter, 1980*; *Poschmann & Dunlop, 2006*; *Kühl, Poschmann & Rust, 2013*). As detailed below, the holotype and only specimen known of *Palaeothea devonica* is probably lost.

The sea spiders of the Hunsrück Slate are thus remarkably different from extant sea spiders, *i.e.*, pantopods. In particular, the abdomen of *P. problematicus*, *P. maucheri* and *F. blocki* was clearly developed, segmented and sometimes possessed a conspicuous telson, while a reduced, unsegmented abdomen lacking a telson is arguably the most solid synapomorphy for Pantopoda (*Bergström, Stürmer & Winter, 1980*; *Sabroux et al., 2019*, *2023*). This differentiates Hunsrück–and Palaeozoic pycnogonids in general–from the sea spiders of the Jurassic, which are all typical pantopods (*Charbonnier, Vannier & Riou, 2007*; *Sabroux et al., 2019*, *2023*). However, it has been suggested that *F. blocki*, *P. devonica*, *P. maucheri*, or *P. vogteli* are related to Pantopoda, relying on some common features of each of these fossils with extant pycnogonids (*e.g.*, structure of the chelifores, of the abdomen or of the legs, polymerous species) or based on phylogenetic analyses (*Bergström, Stürmer & Winter, 1980*; *Siveter et al., 2004*; *Poschmann & Dunlop, 2006*; *Bamber, 2007*; *Kühl, Poschmann & Rust, 2013*). If at least one of these fossils could be confidently assigned to crown-group Pantopoda, this would bear significantly on the calibration of the sea spiders' evolution to time, providing the oldest calibration point(s) for the minimal age of the group (*Sabroux et al., 2023*). If not, the Hunsrück pycnogonids remain of considerable importance in resolving the taxonomy of fossil sea spiders, which is poorly developed. *P. problematicus* and *P. maucheri* were assigned to two orders, Palaeoisopoda and Palaeopantopoda (*Broili, 1930*, *1932*; *Hedgpeth, 1978*; *Bamber, 2007*) of which they are the sole representatives. Similarly, the Silurian *Haliestes dasos* was assigned to a third order, Nectopantopoda (*Bamber, 2007*), while others have not been assigned to any suprageneric group (Table 1). Deciphering how these fossils are related to each other, to pantopods, and to other arthropods, will provide a clearer understanding of the early evolution of sea
spiders during the Palaeozoic and, ultimately, could help to resolve where Pycnogonida fit into Arthropoda phylogeny. This work presents a new perspective of the Hunsrück sea spiders morphology using X-ray microtomography and RTI.

## GEOLOGICAL SETTING AND PALAEOENVIRONMENT

The Hunsrück Slate (*Bartels, Briggs & Brassel, 1998*; *Rust et al., 2016* and references therein; *Wilkin, 2023*), sometimes referred to using the German *Hunsrückschiefer*, is a group of marine deposits of the Early Devonian (Late Pragian to Early Emsian), extending South and West of Koblenz in the part of the Rhenish Massif that extends into Rhineland-Palatinate (for a map, see fig. 3 in *Bartels, Briggs & Brassel (1998)*). This area has been an active mining region from the Roman period (and probably before), through the Middle Ages, and until the end of the 20[th] century. The intensive extraction of slates brought to light a remarkable collection of fossils, among which are the famous and exquisitely preserved specimens from the Central Hunsrück Basin, which qualify the Hunsrück Slate as a Konservat-Lagerstätte. Exceptional fossil preservation is restricted to the middle Kaub Formation, which is itself subdivided in the Central Hunsrück Basin into a series of members: Bundenbach, Herrenberg, Kühstabel, Eschenbach, Bocksberg, Wingertshell and Obereschenbach (*Schindler et al., 2002*; *Bartels et al., 2002*). Because most fossils were found among the mining spoils rather than *in situ*, the exact origin of Hunsrück fossils is often unknown: most of those presented in this study, if not all, are from the region of the Central Hunsrück Basin in the vicinities of Bundenbach and Germünden, and their preservation is typical of the middle Kaub Formation (see Material S1 for details).

The deposits of the middle Kaub Formation are assigned to the Early Emsian (*Schindler et al., 2002*; *Kaufmann et al., 2005*; *De Baets et al., 2013*). They correspond to a shallow marine environment, along the coastline of the Old Red Sandstone Continent, probably just below wave base and not deeper than 100 m, well oxygenated, and inhabited by a large diversity of organisms (more than 260 animal species have been identified), including sponges, arthropods, molluscs, bryozoans, annelids, echinoderms and fishes (*Bartels, Briggs & Brassel, 1998*; *Rust et al., 2016*). Continental erosion formed mud and sand that were transported by rivers into the Central Hunsrück Basin. Mudflows were frequent, facilitating the rapid autochthonous or parautochthonous burial of a few living organisms and resulting in their exceptional preservation. Part of this material was pyritized. Accumulating deposits of mudstones encased these specimens, and the sediments were subject to low-grade metamorphism during the Variscan orogeny (*Wagner & Boyce, 2006*), forming the modern slates.

## MATERIALS AND METHODS

### Studied material

We accessed and studied in detail 46 slabs preserving 63 fossil specimens (listed in Material S1) representing at least four species: *Palaeoisopus problematicus* (51 specimens), *Palaeopantopus maucheri* (three specimens), *Flagellopantopus blocki* (one specimen), and *Pentapantopus vogteli* (three specimens, with possibly an additional one) and some

undetermined pycnogonid material (four specimens). We could not locate the holotype and only specimen known of *Palaeothea devonica* and as such, it could not be included in our study. Part of the fossil material described here has already been presented in previous works while some other fossils are figured and described for the first time (refer to Material S1). The fossils are preserved as flattened slates with variable levels of pyritization. They are hosted in the collections of the Museum für Naturkunde, Berlin (collection numbers MB-A-), the Institut für Geowissenschaften, Section Palaeontology, University of Bonn (collection numbers IGPB-), the Naturhistorisches Museum Mainz (collection numbers NHMMZ PWL), and the Bayerische Staatssammlung für Paläontologie und Geologie, Munich (collection numbers SNSB-BSPG-).

### X-ray microtomography

Seventeen slabs were selected for X-ray microtomography, scanned either with a Nikon XTH 225ST at the University of Bristol, a Phoenix|x-ray v|tome|xs at the Rheinische Friedrich-Wilhelms-Universität Bonn, or a Yxlon FF85 Modular at the Museum für Naturkunde in Berlin. The current during the scan ranged from 70 to 300 μA, voltage from 30 to 215 kV and exposure time from 1 to 2 s. A tungsten reflection target and copper and tin filters were used in some cases. The voxel sizes obtained ranges from 10 to 126 μm depending primarily on the size of the slab hosting the specimen (see Material S2 for detailed parameters for each scan). Often the poor contrast between the matrix and specimens (where not pyritised) or the resulting streak artefacts (where pyritised) precluded segmentation. As such, we instead present microtomography data using two-dimensional maximum intensity projections over a few slices. The software ORS Dragonfly (build 2021.3.0.1087, Montreal, Canada) was used to reconstruct these projections.

### Reflectance transformation imaging

Reflectance transformation imaging (*e.g.*, *Béthoux, Llamosi & Toussaint, 2016*; *Decombeix et al., 2021*; *Sabroux et al., 2023*) was used to image the surface of the fossils. This was performed for 46 slabs (60 specimens), using an RTI-dome, *i.e.*, a hemispheric rig placed over the fossil with an automated lighting series of LEDs evenly spaced around its concave surface. Here, we used a 32-cm diameter light dome equipped with 52 LEDs distributed over three rings. Light orientation was recorded on each image by a reflective metal ball. Photos were taken with a Nikon D850 mounted with NIKKOR 40, 60 and 105 mm lenses or a Canon EOS 700D digital camera, with Canon 50 mm and Canon EF 100 mm macro lenses (see Material S3 for details).

Images were computed with the software RTIbuilder (Cultural Heritage Imaging, San Francisco, CA, USA) using the Highlight Based (HSH Filter) operation sequence. The resulting models were then visualised using the RTIviewer software (Cultural Heritage Imaging, San Francisco, CA, USA). RTIviewer allows the user to vary the direction of illumination at will. "Specular enhancement" allows the contrast between illuminated and shadowed surfaces of the item to be enhanced by estimating the normal for each pixel and using this to render surfaces using synthetic specular highlights. Finally, "normals

visualization" enables visualisation of all normals at once with false colours depending on the position of the light source that most strongly illuminates a given part of the item.

## RESULTS

### Systematic palaeontology

The synonymy lists were written following the recommendations of *Matthews (1973)*. The terminal claw of appendages is counted here as one podomere, and podomere morphological terminology adopted is the same as in *Sabroux et al. (2023)* and *Siveter et al. (2023)*.

### Class Pycnogonida Latreille, 1810

*Palaeoisopus problematicus* Broili, 1928
Figures 1–19.

v* 1928 *Palaeoisopus problematicus*: Broili, plate 1.

1932 *Palaeoisopus problematicus*: Broili, pp 45-54, figs. 1-5.

vp 1932 *Palaeoisopus problematicus*: Helfer, pp 70, 71, figs. 53, 55 (*Helfer, 1932*).

vp 1933 *Palaeoisopus problematicus*: Broili, pp 33-46, plates 1-5 (*Broili, 1933*).

*1944 Palaeoisopus problematicus*: Størmer, p. 146 [text], fig. 29a (*Størmer, 1944*).

*1954 Palaeoisopus problematicus*: Hedgpeth, pp 197-199 [text], 201 [text], 202 [text], fig. 3.

*1955b Palaeoisopus problematicus*: Hedgpeth, pp P171-P173 [text], fig. 123 (*Hedgpeth, 1955b*).

*1957 Palaeoisopus problematicus*: Dubinin, p. 881, fig. 1A-B.

*1958 Palaeoisopus problematicus*: Tiegs & Manton, p. 321 [text], fig. 18a (*Tiegs & Manton, 1958*).

vp 1959 *Palaeoisopus problematicus*: Lehmann, pp 98-101, tabs 10, 11.

*1978 Palaeoisopus problematicus*: Hedgpeth, pp 23-25 [text], 30 [text], 33 [text], figs 1, 2B (*Hedgpeth, 1955a*).

vp 1980 *Palaeoisopus problematicus*: Bergström et al., pp 10-31, 32 [text], 33 [text], 47-49 [text], figs. 1-23, 34 [affinities].

v 1998 *Palaeoisopus problematicus*: Bartels et al., pp 153-155, figs. 130-132.

v 2012a *Palaeoisopus problematicus*: Kühl et al., p. 70 [text], figs. 74, 75.

v 2012b *Palaeoisopus problematicus*: Kühl et al. p. 74 [text], figs. 74, 75. (cop *Kühl et al., 2012a*, *2012b*).

2017 *Palaeoisopus problematicus*: Südkamp, pp 77–79, figs. 118–120.

**Examined material: HOLOTYPE:** SNSB-BSPG 1928 VII 11: 1 juvenile–**OTHERS:** IGPB-AR-339: 1 sp., -AR-340: 1 sp., -M-142: 1 sp., -HS206:1 sp., -HS207:1 sp., -HS456: 1 sp., -HS457: 1 sp., -HS582: 2 spp., -HS636: 1 sp., -HS660: 1 sp., -HS694: 1 sp., -HS942: 1 juvenile, -HS1039: 3 spp.,– MB-A-46: 1 sp., -47: 1 sp., -288: 1 sp., -313: 1 sp., -3969-1: 1 sp. – NHMMZ PWL 1986/3: 1 juvenile, 1992/178-LS: 1 sp. 1994/54-LS: 1 juvenile, 1 sp., 1994/55-LS: 1 sp., 1994/56-LS: 1 sp., 1994/133-LS: 1 sp., 1995/17-LS: 1 sp., 1995/35-LS: 4 spp., 1996/18-LS: 1 sp., 1997/44-45-LS: 3 spp., 1998/122-LS: 2 spp, 1998/155-LS: 1 sp., 2000/46-LS: 1 juvenile, 2003/272-LS: 2 spp., 1 juvenile, 2008/141-LS: 1 sp., 2013/8-LS: 1 sp. – SNSB-BSPG 1932 I 63: 1 sp., 1932 I 67: 1 sp., 1967 I 306: 1 sp., 2021 IV 1: 1 sp.

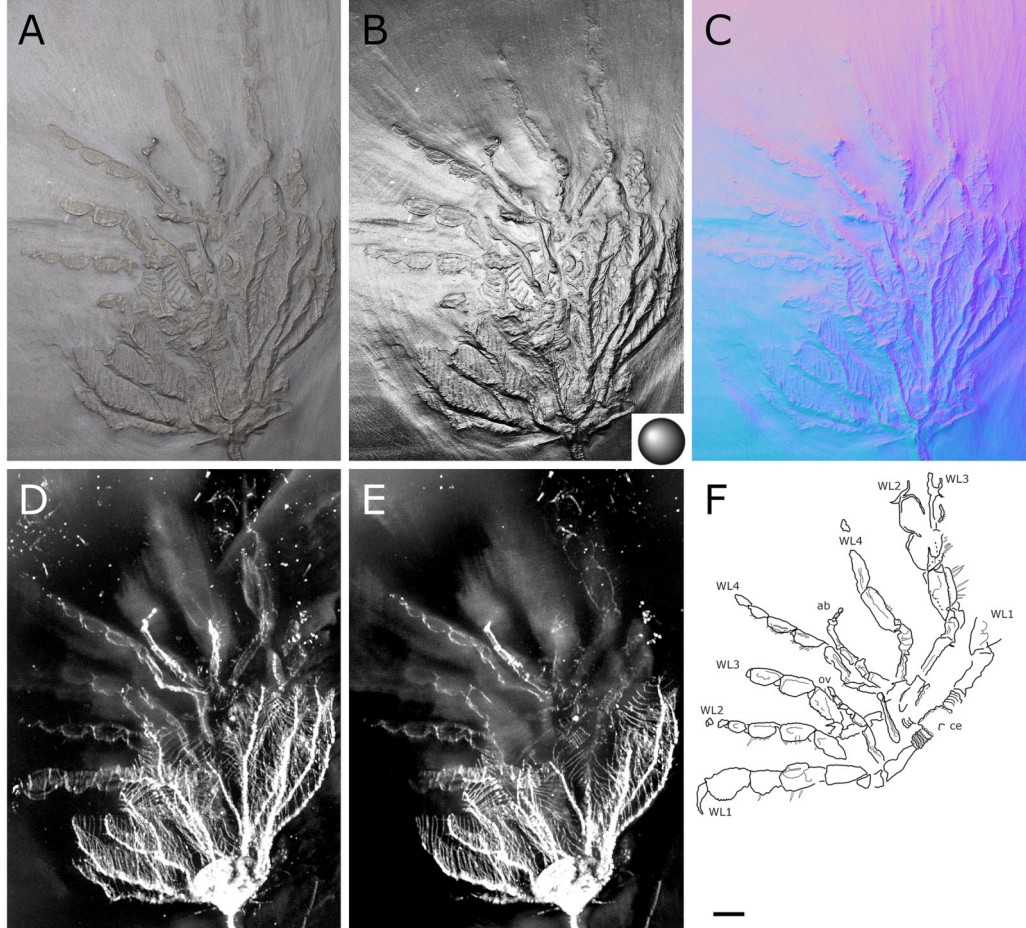

**Figure 1** *Palaeiosopus problematicus*, holotype SNSB-BSPG 1928 VII 11. (A) Standard view. (B) Specular enhancement (direction of the light is indicated by the hemisphere on the bottom right, see Material S3 for details). (C) Normals visualization. (D, E) Maximum intensity views of X-ray microtomography. (F) Interpretative drawing. *ab*: abdomen, *ce*: cephalon, *ov*: oviger. *WL1-4*: walking legs 1–4. Scale bar 5 mm.               

**Diagnosis.** Body robust. Proboscis broad, about as long as the distance between the cephalic anterior margin and the posterior margin of the second trunk segment. Dorsal surface of the second and third trunk segments with tubercles disposed in a V, distal margin of all trunk segments thickened. Abdomen long, five-segmented, carrying an anus on the ventroproximal part of the ultimate segment, distal telson lanceolate. Chelifores massive. Palps with nine articles, the proximalmost podomere (coxa 1) divided in three rings, the distalmost ring individualised from the two proximalmost rings, sixth podomere (tibia) carrying a dorsal spur, no claw. Ovigers' proximalmost podomere (coxa 1) divided into three rings, penultimate article with robust spine on distal margin, ultimate article a claw. Four pairs of legs consisting of nine (first pair of legs) or 10 (second to fourth pairs of legs) podomeres including a subchelate, terminal main claw; coxa 1 divided into rings; distal leg podomeres (patella to propodus) laterally flattened, second to fourth legs with pairs of long ventral setae.

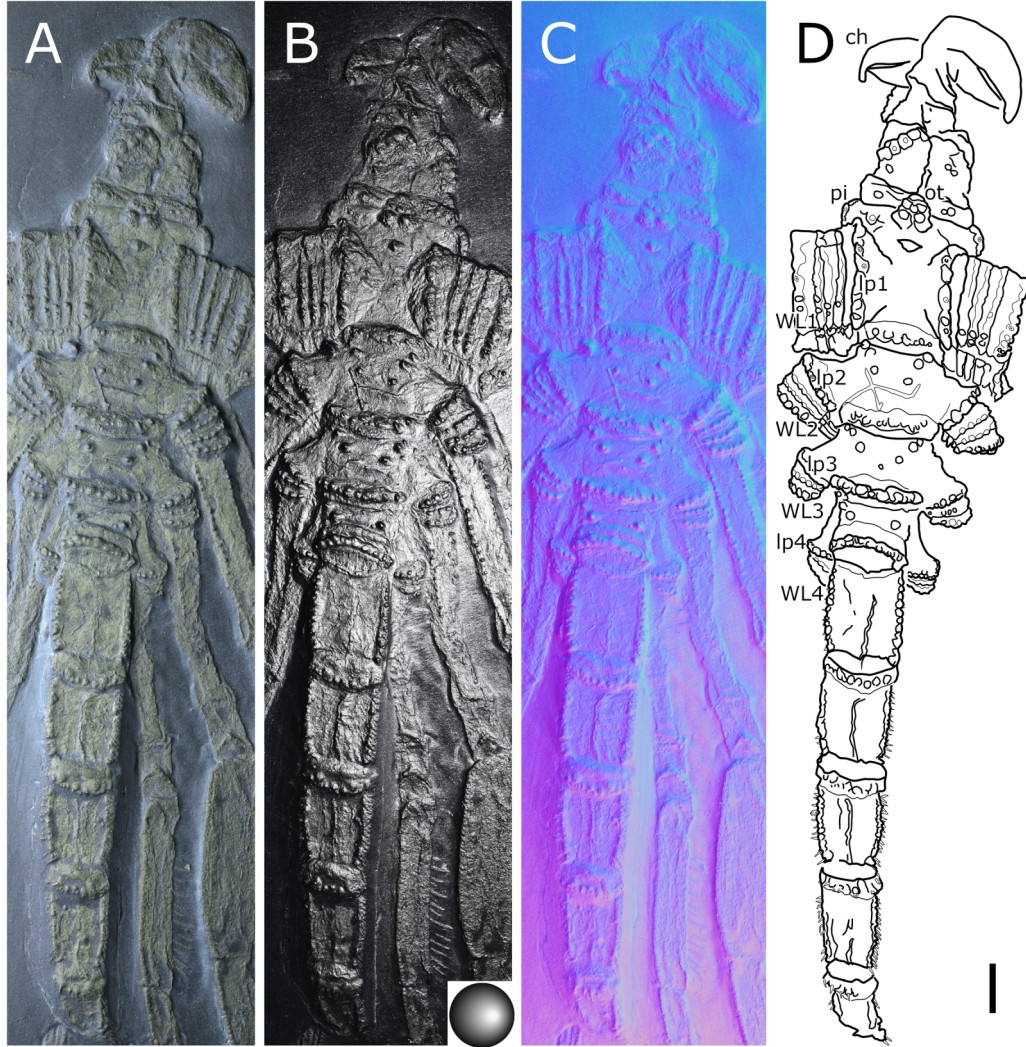

**Figure 2** *Palaeoisopus problematicus*, body of specimen NHMMZ PWL 1994/133-LS. (A) Standard view. (B) Specular enhancement (direction of the light is indicated by the hemisphere on the bottom right, see Material S3 for details). (C) Normals visualization. (D) Interpretative drawing. *ch*: chelifore, *lp1-4*: lateral processes 1–4, *ot*: ocular tubercle, *pi*: palp insertion, *WL1-4*: walking legs 1–4. Scale bar 5 mm.

**Description and interpretation.** Members of this species are very large for pycnogonids, reaching up to 400 mm in legspan, and are among the best-known fossils in the Hunsrück Slate; more than 80 specimens are known. The investigations of *Bergström, Stürmer & Winter (1980)* detailed the general morphology of *Palaeoisopus problematicus*, with a complete description of the trunk, the abdomen and the legs. We accessed 51 specimens, including not yet published material (Material S1), to compare and revise the descriptions of *Bergström, Stürmer & Winter (1980*; based on 57 specimens) in the light of new evidence.

**The body** of *P. problematicus* is composed of four trunk and five abdominal segments (Figs. 1, 2). The cephalon is fused with the first trunk segment as in modern pycnogonids.

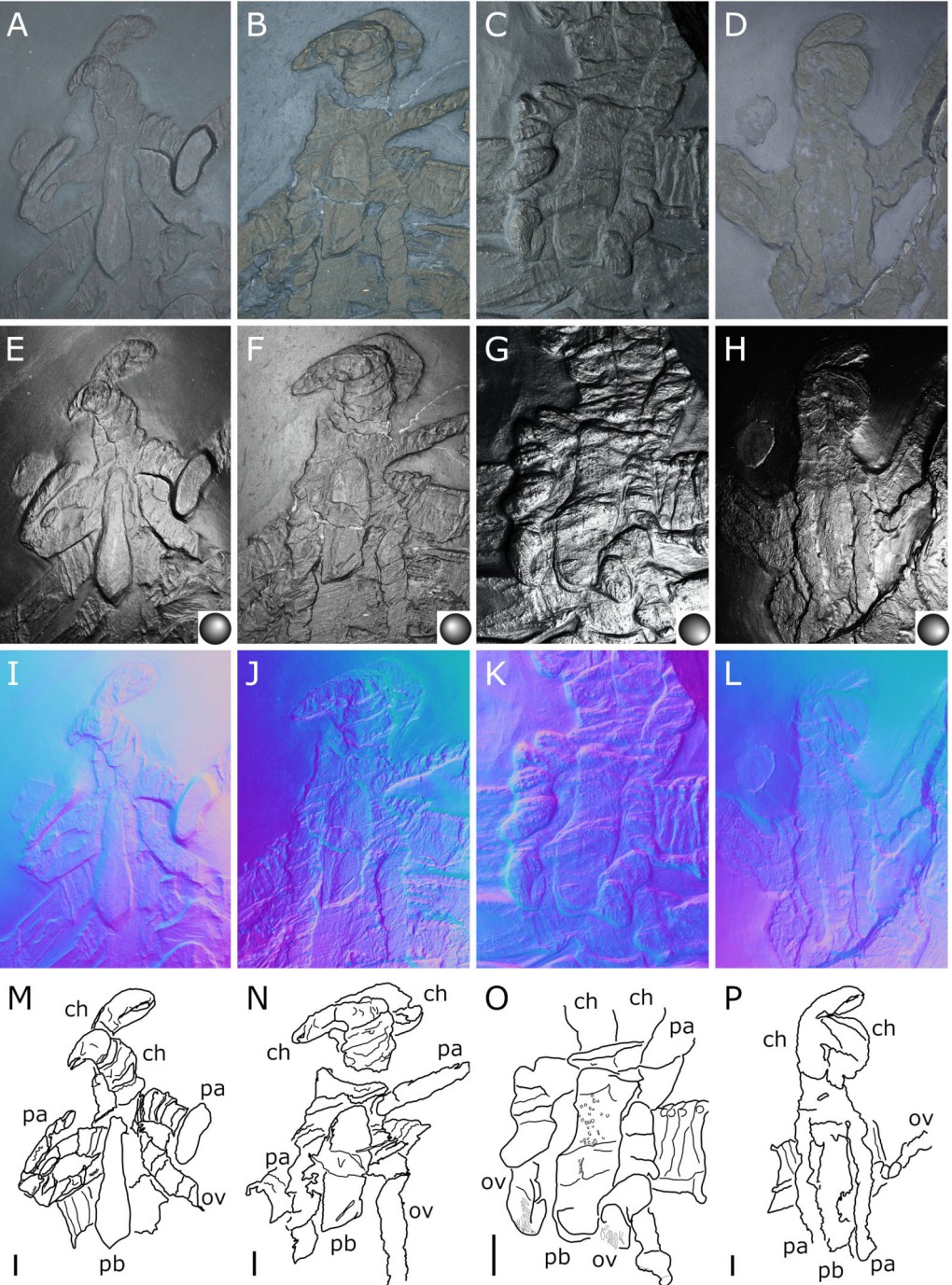

**Figure 3 *Palaeoisopus problematicus*, cephalon and proboscis in ventral view.** Specimens IGPB-HS207 (A, E, I, M), IGPB-HS660 (B, F, J, N), SNSB-BSPG 1932 I 67 (C, G, K, O) and NHMMZ PWL 1997/44-45-LS (D, H, L, P). (A–D) Standard view. (E–H) Specular enhancement (direction of the light is indicated by the hemisphere on the bottom right, see Material S3 for details). (I–L) Normals visualization. (M–P) Interpretative drawings. *ch*: chelifore, *pa*: palp, *pb*: proboscis, *ov*: oviger. Scale bars 5 mm.

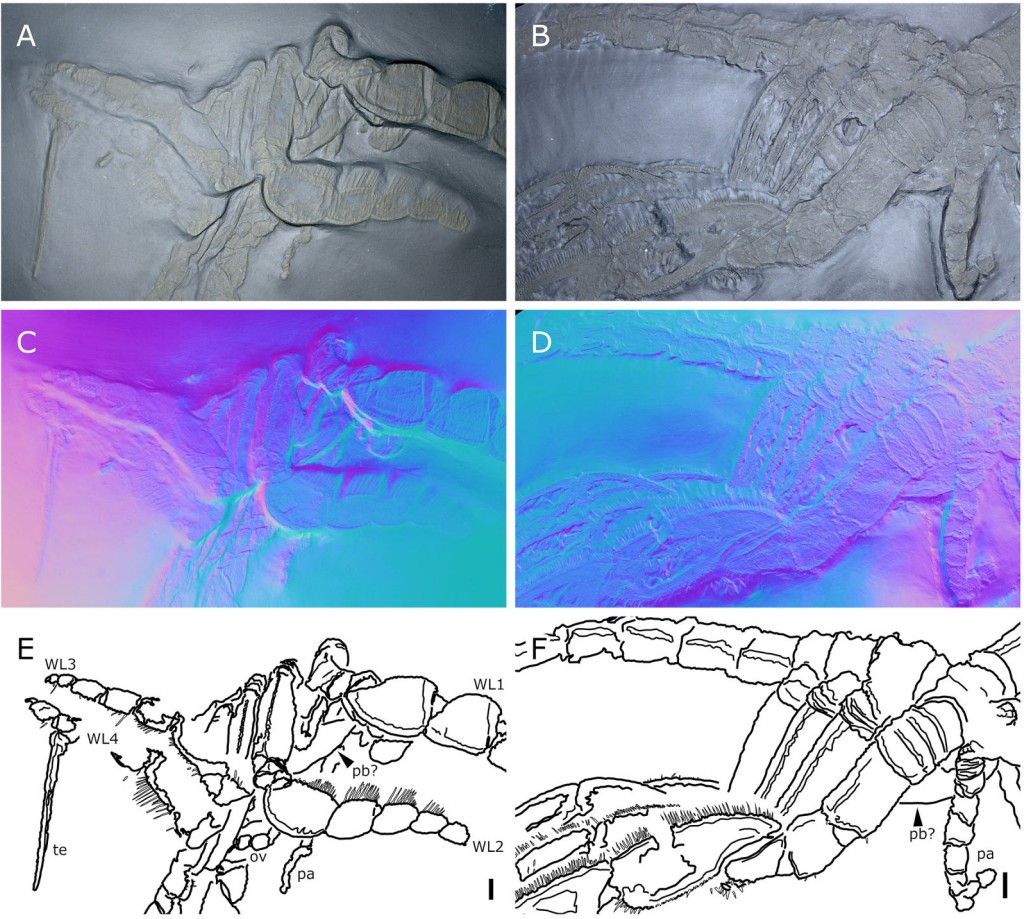

**Figure 4** *Palaeoisopus problematicus*, **lateral view of the body where the proboscis is potentially visible.** Specimens IGPB-HS457 (A, C, E) and NHMMZ PWL 1995/35-LS (B, D, F). (A, B) Standard view. (C, D) Normals visualization. (E, F) Interpretative drawings. *ch*: chelifore, *pa*: palp, *pb*: proboscis, *ov*: oviger, *te*: telson, *WL1-4*: walking legs 1–4. Scale bars 5 mm.

The fusion of the cephalon and the first trunk segment confers a roughly square shape. The first trunk segment carries the lateral processes to which the first pair of walking legs (WL1) articulates (Fig. 2D). These processes are approximately cylindrical in shape, with lateral thickening and ornamentation composed of rounded tubercles (Fig. 2D). Anteriorly and ventrally to the lateral processes, the cephalon also carries lateral extensions to which the palps articulate (Fig. 2D), their distal border broadened. Ovigers insert ventrally, at about the same level as the lateral processes of the first trunk segment (Fig. 3). The anterior margin of the cephalon on which the chelifores articulate also presents an ornamented thickening. The posterior margin of the cephalon is thickened and ornamented with tubercles.

**The proboscis** is visible on some ventrally preserved specimens (Fig. 3), the tip of it being posteriorly directed. Some specimens preserved laterally (Fig. 4) suggest that the proboscis is folded ventrally (its tip directed posteriorly), though the proboscis cannot be

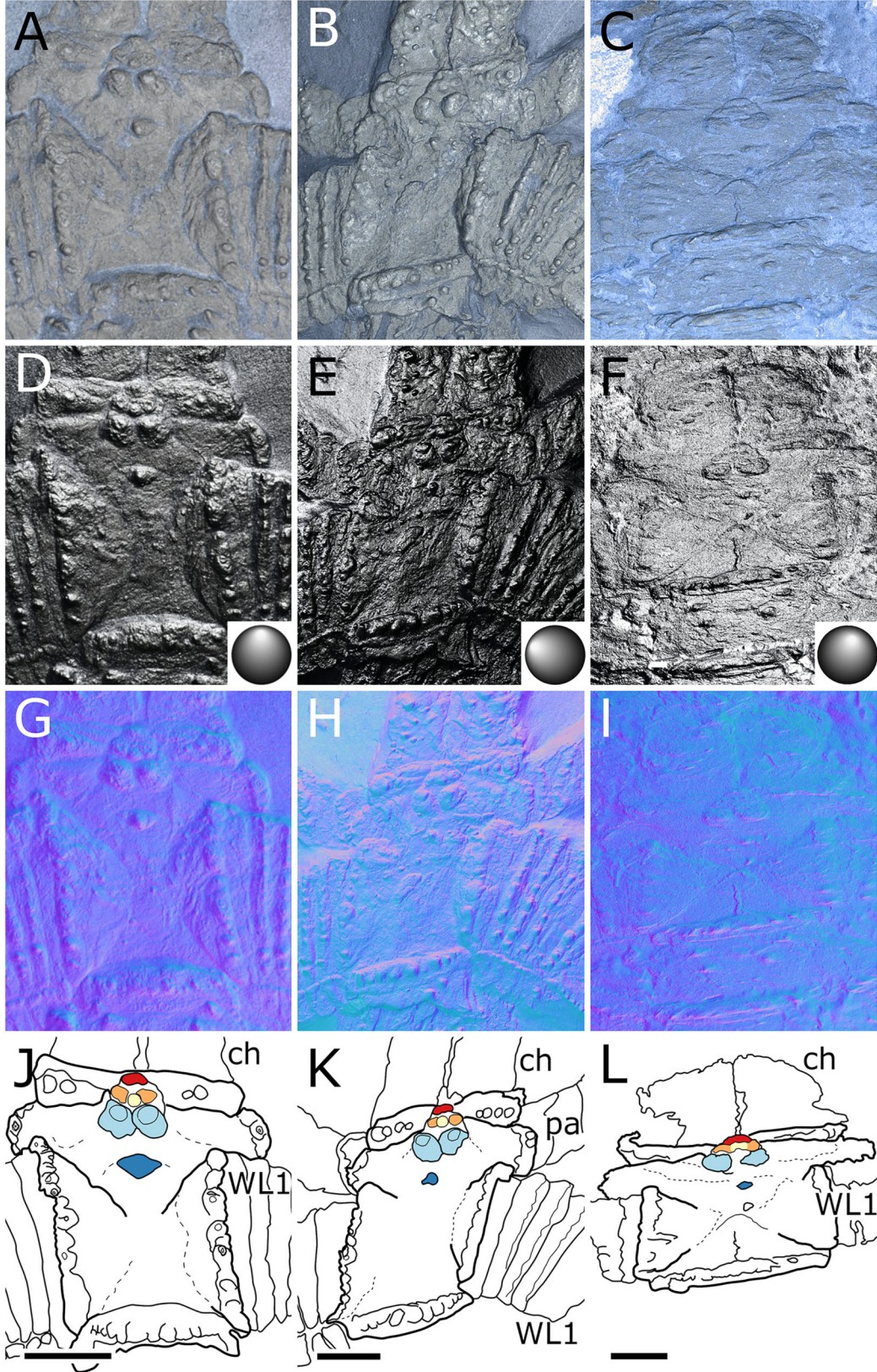

**Figure 5** *Palaeoisopus problematicus*, **cephalon and first trunk segment.** Specimens NHMMZ PWL 1994/133-LS (A, D, G, J), NHMMZ PWL 2008/141-LS (B, E, H, K) and IGPB-AR-340 (C, F, I, L). (A–C)

**Figure 5** (continued)
Standard view. (D-E Specular enhancement (direction of the light is indicated by the hemisphere on the bottom right, see Material S3 for details). (D–I) Normals visualization. (J–L) Interpretative drawings; features of the ocular tubercle discussed in the text are coloured (compare also with Fig. 6), putative lateral sense organs are coloured in orange. *ch*: chelifores, *pa*: palp, *WL1*: walking leg 1. Scale bars 5 mm.

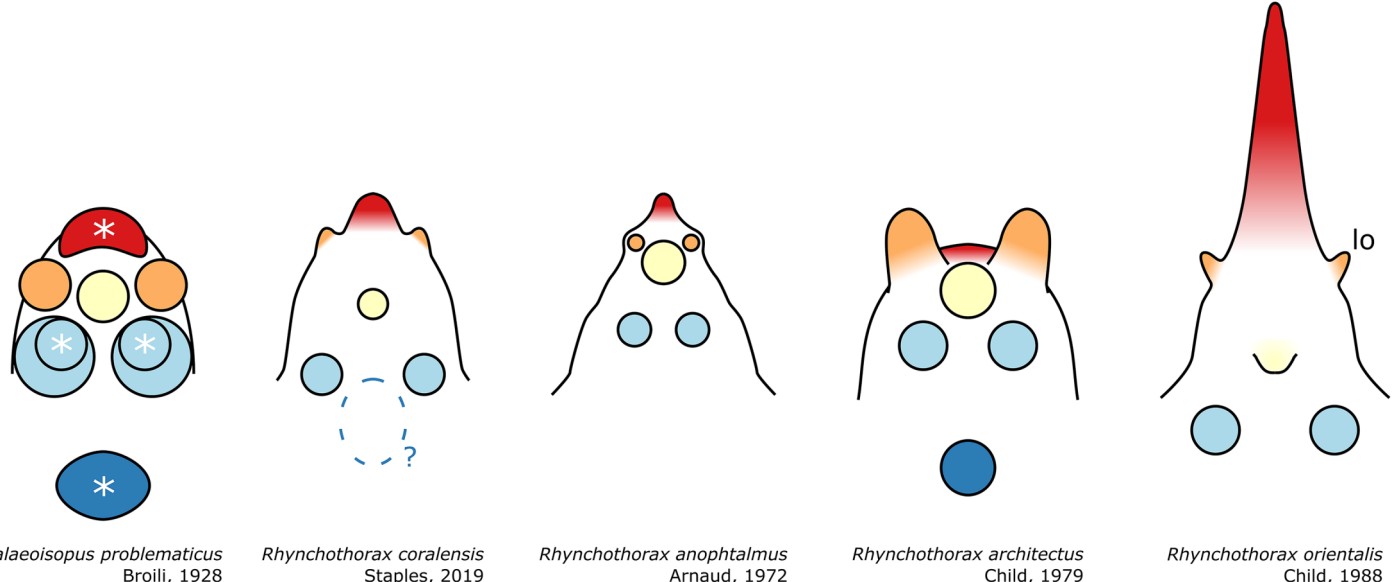

| *Palaeoisopus problematicus* | *Rhynchothorax coralensis* | *Rhynchothorax anophtalmus* | *Rhynchothorax architectus* | *Rhynchothorax orientalis* |
| --- | --- | --- | --- | --- |
| Broili, 1928 | Staples, 2019 | Arnaud, 1972 | Child, 1979 | Child, 1988 |

**Figure 6** Diagram of the organisation of the ocular tubercle ornamenation of *Palaeoisopus problematicus* as well as a few representatives of the modern *Rhynchothorax.* Colours indicate putative homologies and correspond with fig. 5. In the case of the posteriormost dorsal tubercle of *R. coralensis*, there is no clear evidence of ornamentation, but *Staples (2019)* represents a low elevation of the tergite around this position which could correspond. Asterisks (*) indicate the tubercles that were interpreted as eyes by *Bergström, Stürmer & Winter (1980)*. *lo*: lateral sense organ (coloured in orange). *Rhynchothorax* ocular tubercles diagrams are based on the illustrations of *Arnaud (1974)*, *Child (1979, 1988)* and *Staples (2019)*.

unambiguously identified in these specimens. If correct, this position is potentially a resting position and the proboscis was mobile, as seen in modern species presenting the same orientation of the proboscis (such as many modern Ascorhynchidae Hoek, 1881). The proboscis is tubular in shape overall, without clear proximal or distal narrowing. According to *Bergström, Stürmer & Winter (1980)* the dorsal surface of the proboscis is divided into three fields, suggesting that it is composed of three antimeres like extant species: this observation was made based on a single specimen that we could not examine.

**The ocular tubercle** region (Fig. 5) is positioned near the anterior margin of the cephalon. It has six rounded features (Figs. 5, 6). The anteriormost is roughly reniform, and there are two positioned postero-laterally to the first, surrounding a third central one. Posterior to these, are two further lateral rounded features with a broader base. These six features are concentrated in a region on the cephalon. More posteriorly, a seventh feature is also visible. *Bergström, Stürmer & Winter (1980)* proposed that this posteriormost feature is a posterior ocellus; these authors also interpreted the anterior, reniform features as an anterior eye, and the posteriormost pair of features as laterally positioned eyes (Fig. 6). If correct, this

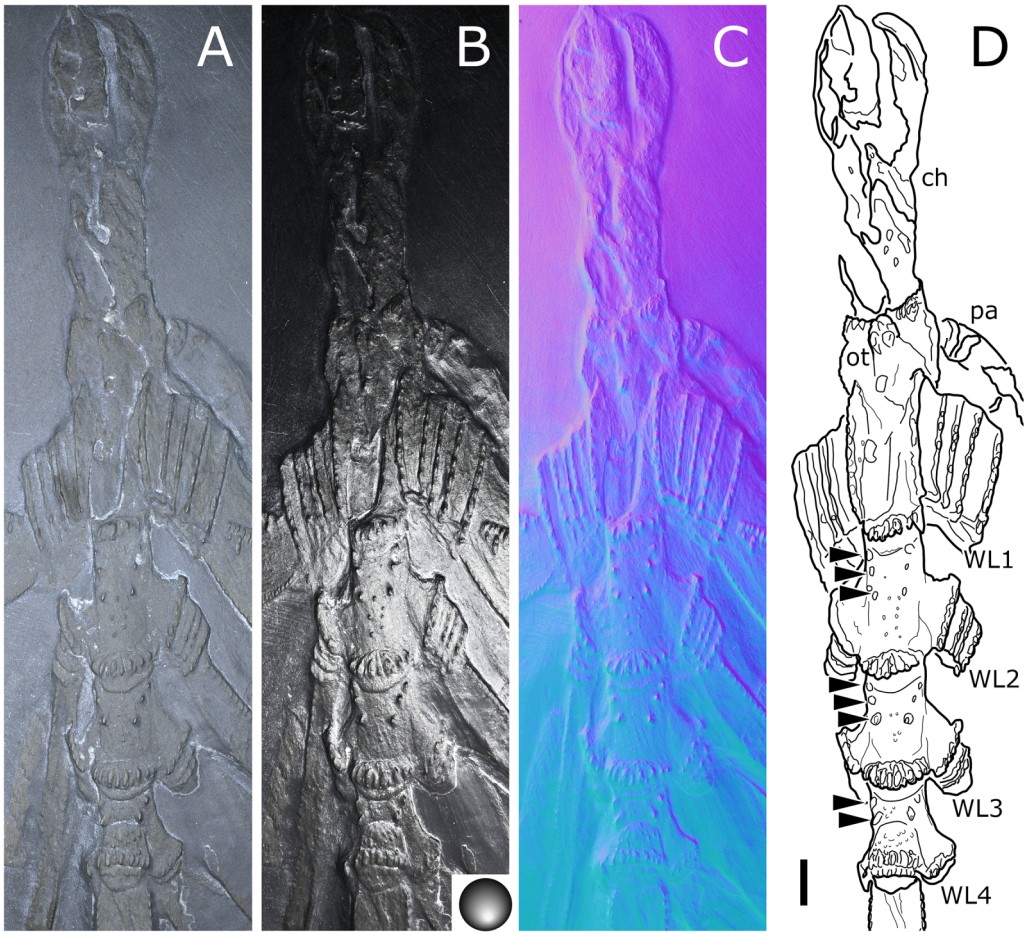

**Figure 7** *Palaeoisopus problematicus*, **trunk and cephalon of the specimen IGPB-HS206.** (A) Standard view. (B) Specular enhancement (direction of the light is indicated by the hemisphere on the bottom right, see Material S3 for details). (C) Normals visualization. (D) Interpretative drawing. Black arrowheads point out the dorsal ornamentation of the trunk segments. *ch*: chelifore, *ot*: ocular tubercle, *pa*: palp, *WL1-4*: walking legs 1–4. Scale bar 5 mm.           

arrangement would be very different from extant species that instead present anterior and posterior eye pairs, and this interpretation does not explain the identity of the three features directly posterior to the "anterior eye". It seems unlikely that any of these represent any eye at all, since eyes of modern sea spiders generally do not protrude from the cuticle, or only as a low convex lens (*Brenneis, 2022*) which is unlikely to preserve as markedly as the structures here following the dorsoventral compression typical of Hunsrück fossils. Many species in the extant genus *Rhynchothorax* Costa, 1861 (family Rhynchothoracidae Thompson, 1909) possess a disposition of more or less elongated tubercles very similar to the features seen in *P. problematicus* (Fig. 6; some of them can be absent in some species). We suggest that the features of *P. problematicus* are tubercles, homologous to those in *Rhynchothorax*. The two tubercles postero-lateral to the anteriormost correspond in position to the two lateral sense organs (*sensu Lehmann, Heß & Melzer, 2017*). These sensory structures have been identified in all sea spider families except for Pycnogonidae Wilson, 1878 (*Brenneis, 2022*). We cannot address whether these

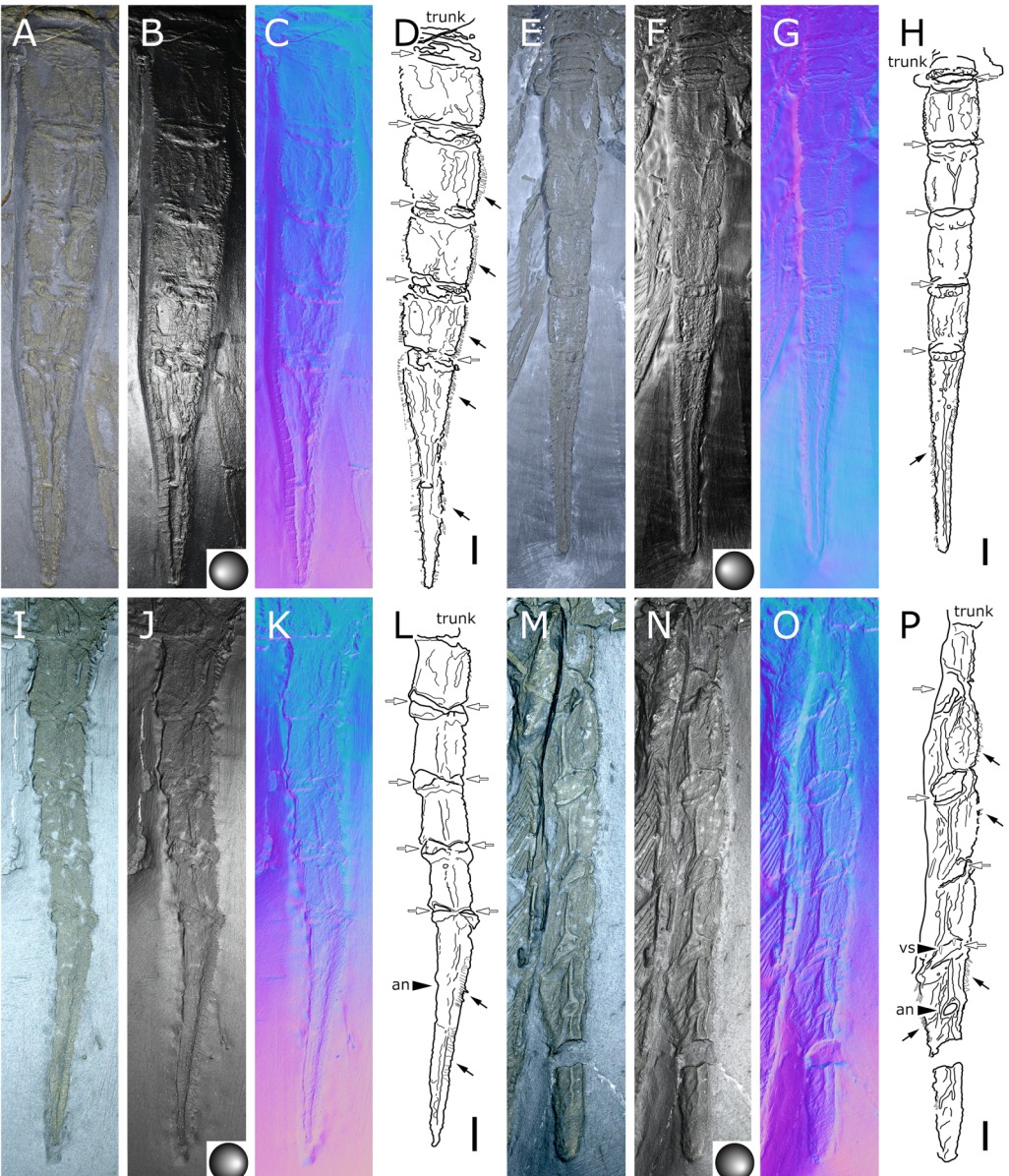

**Figure 8** *Palaeoisopus problematicus,* **abdomen.** Specimens MB-A-46 (A–D, dorsal view), NHMMZ PWL 1998/122-LS (E–H, dorsal view), IGPB-HS1039 (I–L, lateral view), IGPB-HS660 (M-P, ventral view). (A, E, I, M) Standard view. (B, F, J, N) Specular enhancement (direction of the light is indicated by the hemisphere on the bottom right, see Material S3 for details). (C, G, K, O) Normals visualization. (D, H, L, P) Interpretative drawings, black arrows point out some of the setae along the abdominal lateral margins, white arrows point out the arthrodial membranes of the abdomen. *an*: anus, *vs.*: pair of ventral setae. Scale bars 5 mm.

were already functional as lateral sense organs (indeed, their exact function remains unknown even in modern species; *Lehmann, Heß & Melzer, 2017*).

**Trunk** third and fourth segments have two to three pairs of large tubercles arranged in a V tapering posteriorly (Figs. 2, 7). These are followed posteriorly by thinner tubercles which are more variable in disposition and/or preservation. The fourth trunk segment carries two

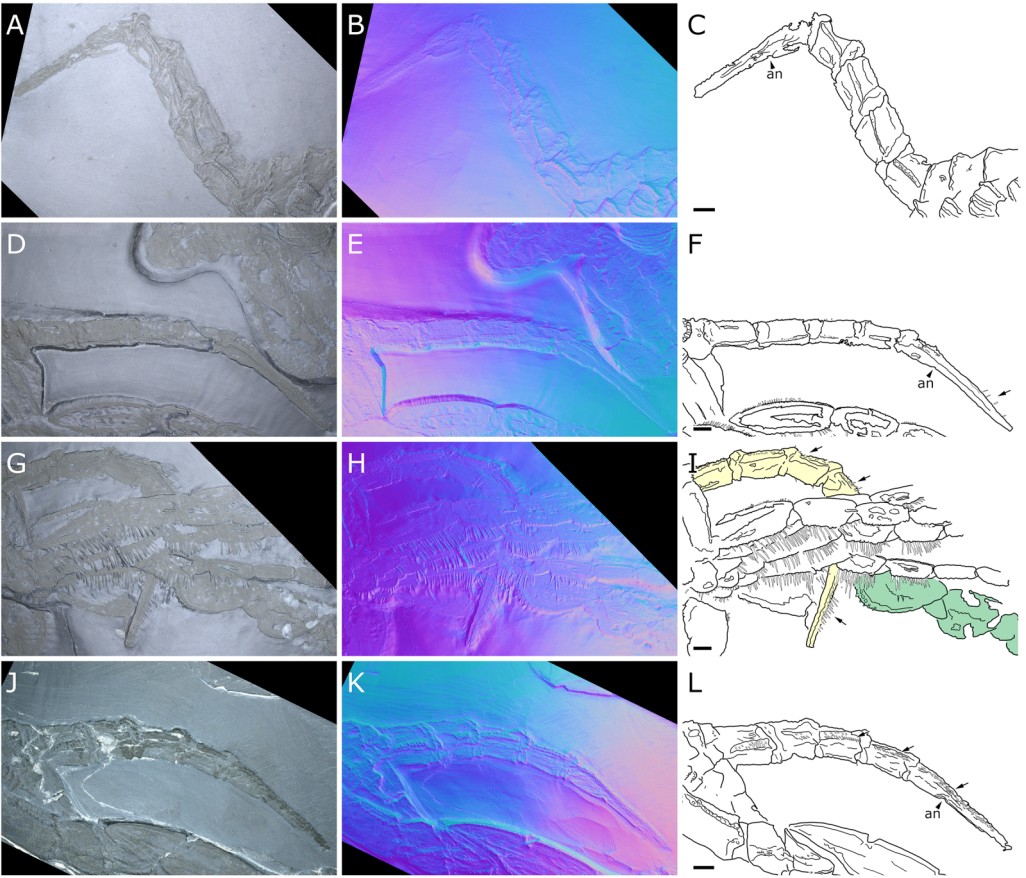

**Figure 9** *Palaeoisopus problematicus*, **evidence for the mobility range of the abdomen (in lateral view).** Specimens NHMMZ PWL 1994/56-LS (A–C), NHMMZ PWL 1995/35-LS specimens 2 (D–F) and 3 (G–I), and IGPB-HS694 (J–L). (A, D, G, J) Standard view. (B, E, H, K) Normals visualization. (C, F, I, L) Interpretative drawings (in I, the abdomen was coloured in yellow for better visibility, and the WL1 of another specimen is coloured in green), black arrows point out some of the setae along the abdominal lateral margins. *an*: anus. Scale bars 5 mm. 

pairs of dorsal tubercles, and many thinner tubercles may be present more posteriorly. The distal margin of each trunk segment is thickened and presents additional ornamentation composed of about 10 tubercles along its length. The lateral processes are very short on the first trunk segment and are longer posteriorly. Their orientation is almost orthogonal to the trunk axis on the first trunk segment, and increasingly diagonal posteriorly, on the fourth trunk segment the axis of the lateral processes is almost parallel to the trunk. The distal margin of the lateral processes is thickened and ornamented with tubercles.

**The abdomen** (Figs. 8, 9) is attached on a posterior thickening of the fourth trunk segment. It carries four rectangular segments that are increasingly elongated distally, and a terminal fifth, lanceolate, very elongated segment (about 40% of the total abdomen length). The fifth abdominal segment carries a ventrally positioned anus at approximately a quarter of the total segment length (Figs. 8I–8P). For *Bergström, Stürmer & Winter (1980)*, the anus indicates the posterior end of the fifth and ultimate abdominal segment, so that

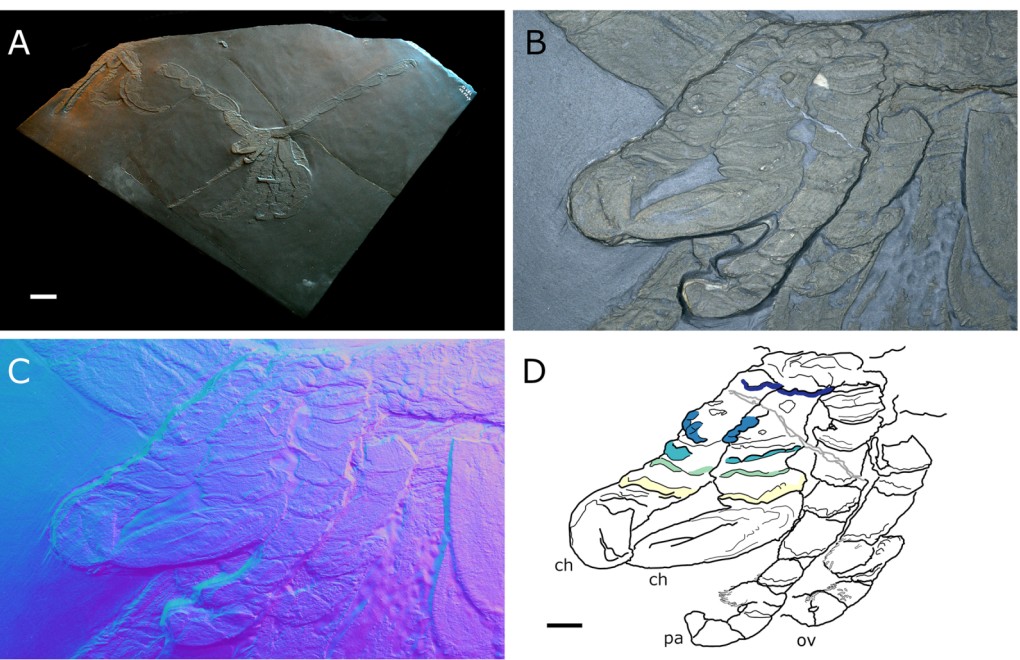

**Figure 10** *Palaeoisopus problematicus,* **specimen IGPB-HS582.** (A) General view of the fossil. (B–D) Zoom on the cephalon region: (B) Standard view. (C) Normals visualization. (D) Interpretative drawing. Margins and lines (see text) are highlighted with different colours according to their position. *ch:* chelifore, *pa:* palp, *ov:* oviger. Scale bars: A: 20 mm, D: 5 mm (B, C same scale as D).

*Palaeoisopus problematicus* possesses five true abdominal segments, with the ultimate one fused with the post-anal telson; but it was suggested alternatively (*e.g.,* *Vilpoux & Waloszeck, 2003*; *Dunlop & Lamsdell, 2017*) that the whole fifth segment actually corresponds to a telson, so that *P. problematicus* only has four abdominal segments, and uniquely presents an anus carried ventrally on the telson. At least the four distal abdominal segments (and probably also the first abdominal segment) carry short setae along their lateral margin. Thickened ridges are present medioventrally and mediodorsally. Dorsally each trunk segment has an anterior thickened margin. This thickened anterior margin may be present ventrally as well, as observed in specimen 1 of the slab IGPB-HS1039 (Figs. 8I–8L), although it is not visible in specimen IGPB-HS660 (Figs. 8M–8P). Both ventral and dorsal sides show a free arthrodial membrane (see Fig. 8). It is larger ventrally, giving the abdomen great ventral mobility (Figs. 9, 10A); dorsally, the arthrodial membrane is smaller, and provided limited dorsal mobility to the abdomen (Figs. 9D–9F), except at the abdomen's base where dorsal mobility seems important (Figs. 9A–9C). Each abdominal segment has setae along its lateral margins, which in some cases (Figs. 9G–9I) appear to be relatively long. They are arranged in at least one, possibly several, rows. Low numbers of additional setae can sporadically be found on the ventral or dorsal posterior margins of abdominal segments (Figs. 8M–8P, 9G–9L). In specimen IGPB-HS660 (Figs. 8M–8P) there is clear evidence of a pair of short setae on the ventroposterior margin of the fourth abdominal segment.

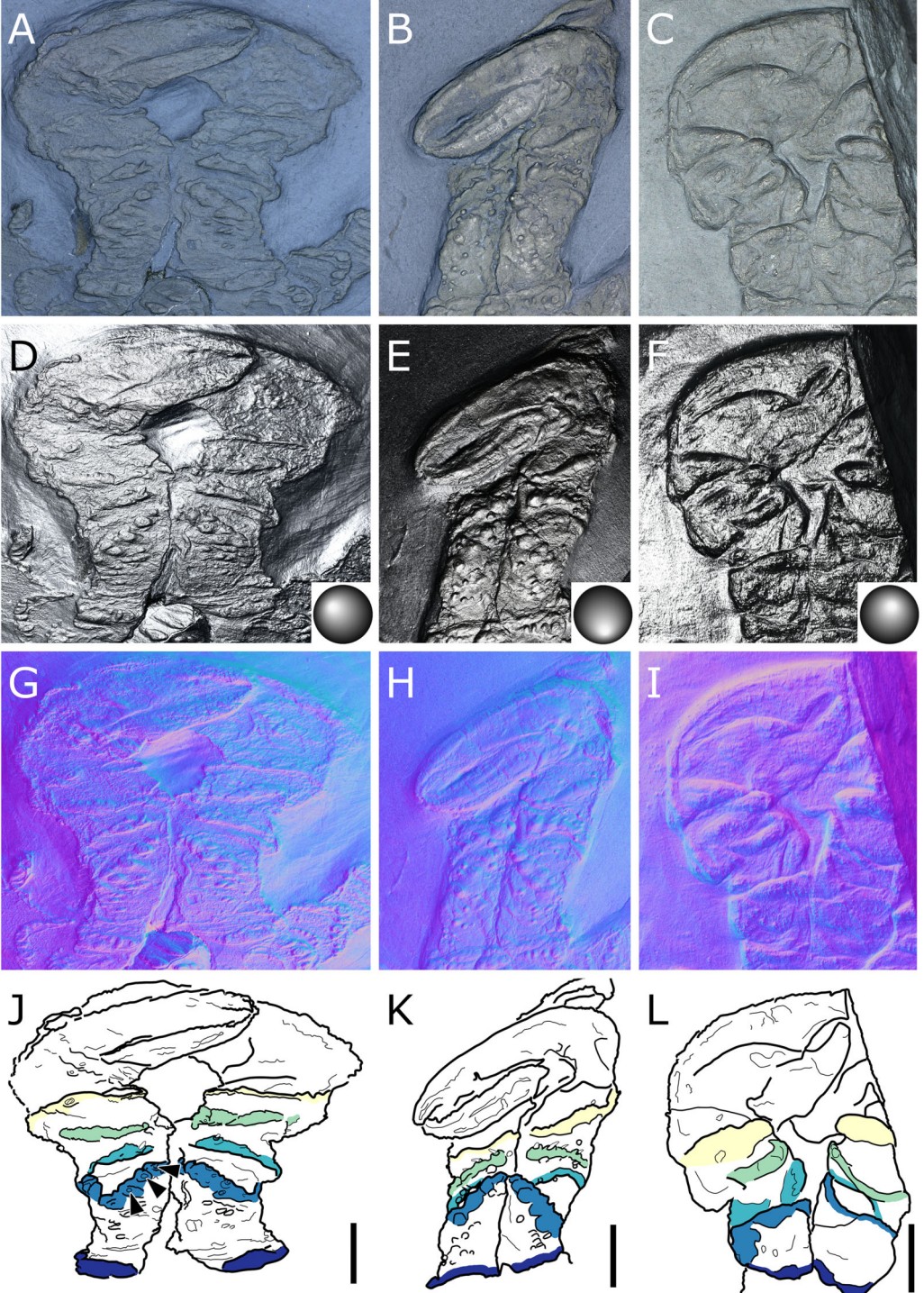

**Figure 11** *Palaeoisopus problematicus,* **chelifores.** Specimens NHMMZ PWL 1996/18-LS (A, D, G, J), NHMMZ PWL 2008/141-LS (B, E, H, K) and SNSB-BSPG 1932 I 67 (C, F, I, L). (A–C) Standard view. (D–F) Specular enhancement (direction of the light is indicated by the hemisphere on the bottom right, see Material S3 for details). (G–I) Normals visualization. (J–L) Interpretative drawings. Black arrowheads in J indicate some of the potential setae insertions. Margins and lines (see text) are highlighted with different colours according to their position. Scale bars 5 mm.

**The chelifores** (Figs. 10, 11) have prominent chelae. Despite their often-superb preservation, we are puzzled by their segmentation. *Bergström, Stürmer & Winter (1980)* counted (with some hesitation) five podomeres, three of which are scapes: this differs significantly from what is observed in any modern pycnogonid, which generally have three podomeres (including one scape), although sometimes one, two or four (with one or two scapes) are present, but never five. From the material we examined, we could not confidently identify five podomeres; four (including two scapes) is the minimal number and seems as likely, if not likelier, than five. The robustness of this interpretation is, however, limited by the ambiguity of the ornamentations of the scapes: it is often very difficult to determine whether ornamentation demarks the margin of a segment, or is in more of a median position (see highlighted margins and lines in Figs. 10 and 11). The proximal margin of the chelifores' first podomere has a ridge ornamented by tubercles, roughly perpendicular to the segment axis; more distally, is another, diagonal ridge (the distal margin of the first scape?) that is ornamented with large tubercles, some of which have marks of possible setal insertions (Fig. 11J). This is followed by another ridge (the proximal ridge of the second scape?) roughly perpendicular to the axis of the purported second scape. Still more distally, is another ridge diagonal to the axis of the appendage. This may either represent a median ridge (two scapes) or the joint between the second and the third scapes (three scapes). It is followed distally by one final ridge–the distal margin of the ultimate scape. Because it is not possible to distinguish proximal and distal margins of the purported joints between the purported second and third scapes, and because it is more conservative (in light of chelifore morphology of extant pycnogonids), we favour the two-scapes hypothesis (Fig. 14A).

**The palps** (Figs. 10, 12) are preserved laterally and generally directed anteriorly. They are composed of two proximal rings (Figs. 10, 12D, 12E) which together form the proximalmost podomere following the interpretation of *Siveter et al. (2023)* for walking legs. The second podomere possesses an angular projection at its distal margin (Figs. 12A, 12B, 12D). The third and fifth podomeres are about 1.5 times to twice as long as the others. The seventh podomere typically has a dorsal spur, its cuticle sometimes appearing coarse (Fig. 12G) or bearing short setae (Fig. 10). There is no distal claw. The second to fifth podomeres have a thickened distal margin. In total, we count a total of 10 podomeres in the palp of *P. problematicus*. This number differs from the typical (extant) sea spider bodyplan, which has nine podomeres, or a reduction from this fundamental pattern (*Sabroux et al., 2023*). Even *Haliestes dasos* has nine palpal podomeres, excluding the terminal claw (*Siveter et al., 2023*). Typically, in modern species, the palps have a first short coxa 1, followed by a longer coxa 2 (*Sabroux et al., 2023*). Since the ovigers of *P. problematicus* have three coxal rings (see below), and palps only two, we hypothesize that the second podomere of the palp of *P. problematicus* could be the distalmost ring of coxa 1, that is individualised. We therefore propose to apply the morphological terminology of *Sabroux et al. (2023)* and *Siveter et al. (2023)* as presented in Figs. 14B, 14D. Whether this additional podomere was mobile does not seem to be addressable here.

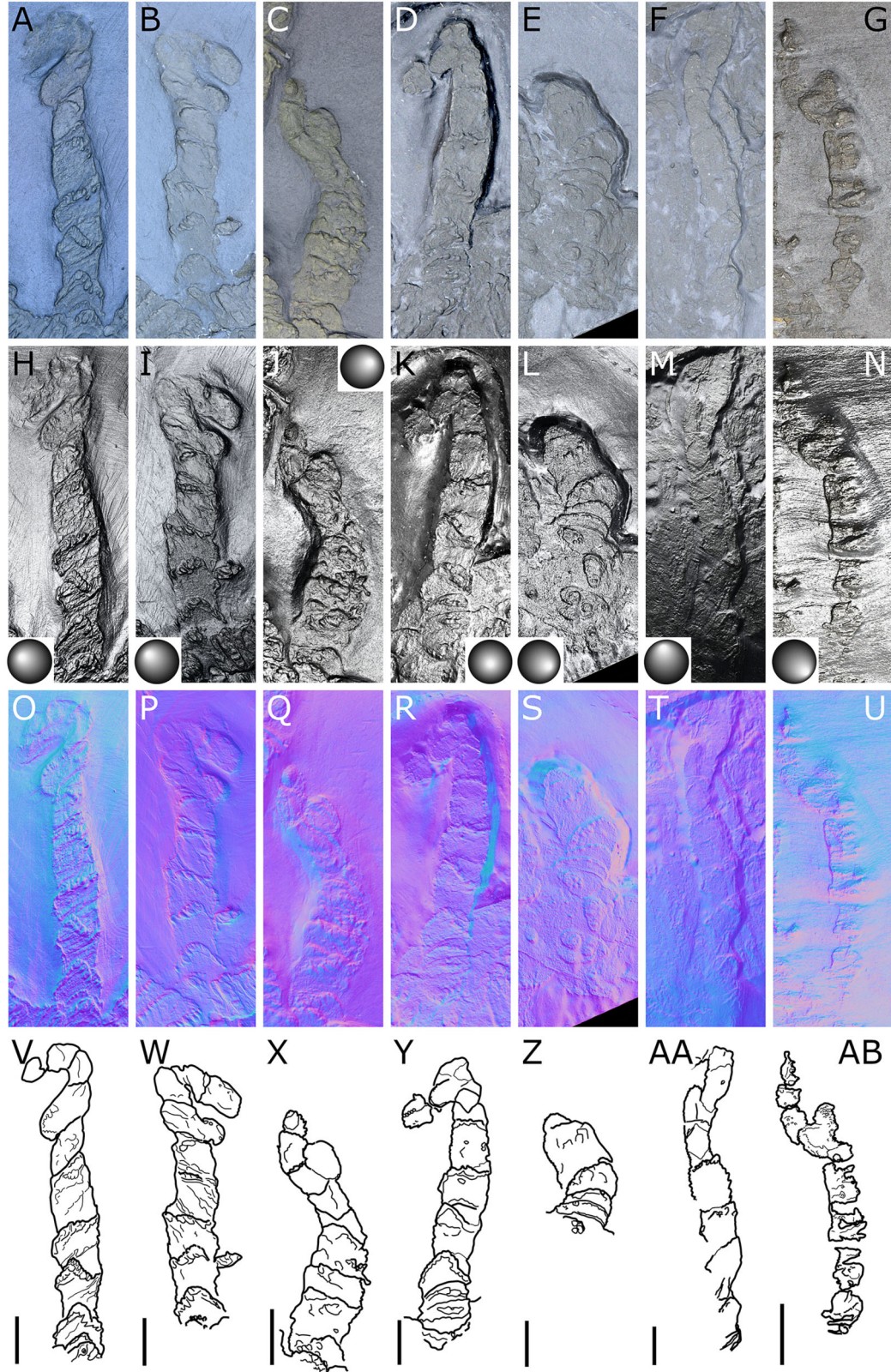

**Figure 12 *Palaeoisopus problematicus*, palps.** Specimens NHMMZ PWL 1996/18-LS (right palp: A, H, O, V; left palp: B, I, P, W), NHMMZ PWL2008/141-LS (left palp: C, J, Q, X), NHMMZ PWL 1995/35-LS

**Figure 12 (continued)**
specimens 1 (right palp: D, K, R, Y) and 2 (right palp: E, L, S, Z), NHMMZ PWL 1997/44-45-LS (right palp: F, M, T, AA), SNSB-BSPG 1932 I 63 (left palp: G, N, U, AB). (A–G) Standard view. (H-N) Specular enhancement (direction of the light is indicated by the hemisphere on the bottom right, bottom left or top right; see Material S3 for details). (O–U) Normals visualization. (V–AB) Interpretative drawings. Scale bars 5 mm.                

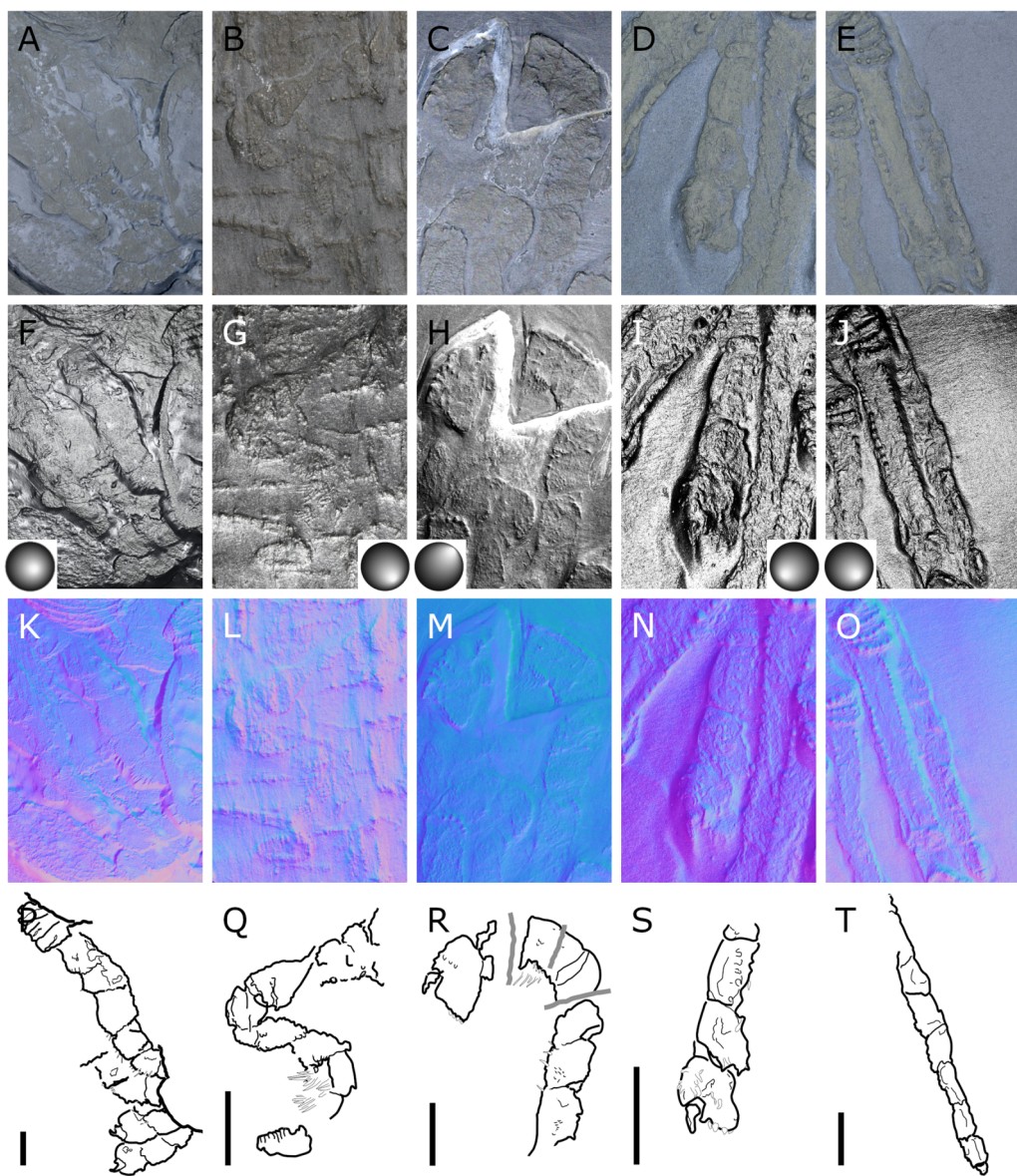

**Figure 13 *Palaeoisopus problematicus,* ovigers.** Specimens NHMMZ PWL 1995/35-LS specimen 3 (unidentified oviger: A, F, K, P), SNSB-BSPG 1932 I 63 (unidentified oviger: B, G, L, Q), NHMMZ PWL 1994/56-LS (unidentified oviger: C, H, M, R) and NHMMZ PWL 1994/133-LS (left oviger: D, I, N, S; right oviger: E, J, O, T). (A–E) Standard view. (F–J) Specular enhancement (direction of the light is indicated by the hemisphere on the bottom right or bottom left; see Material S3 for details). (K–O) Normals visualization. (P–T) Interpretative drawings. Scale bars 5 mm.

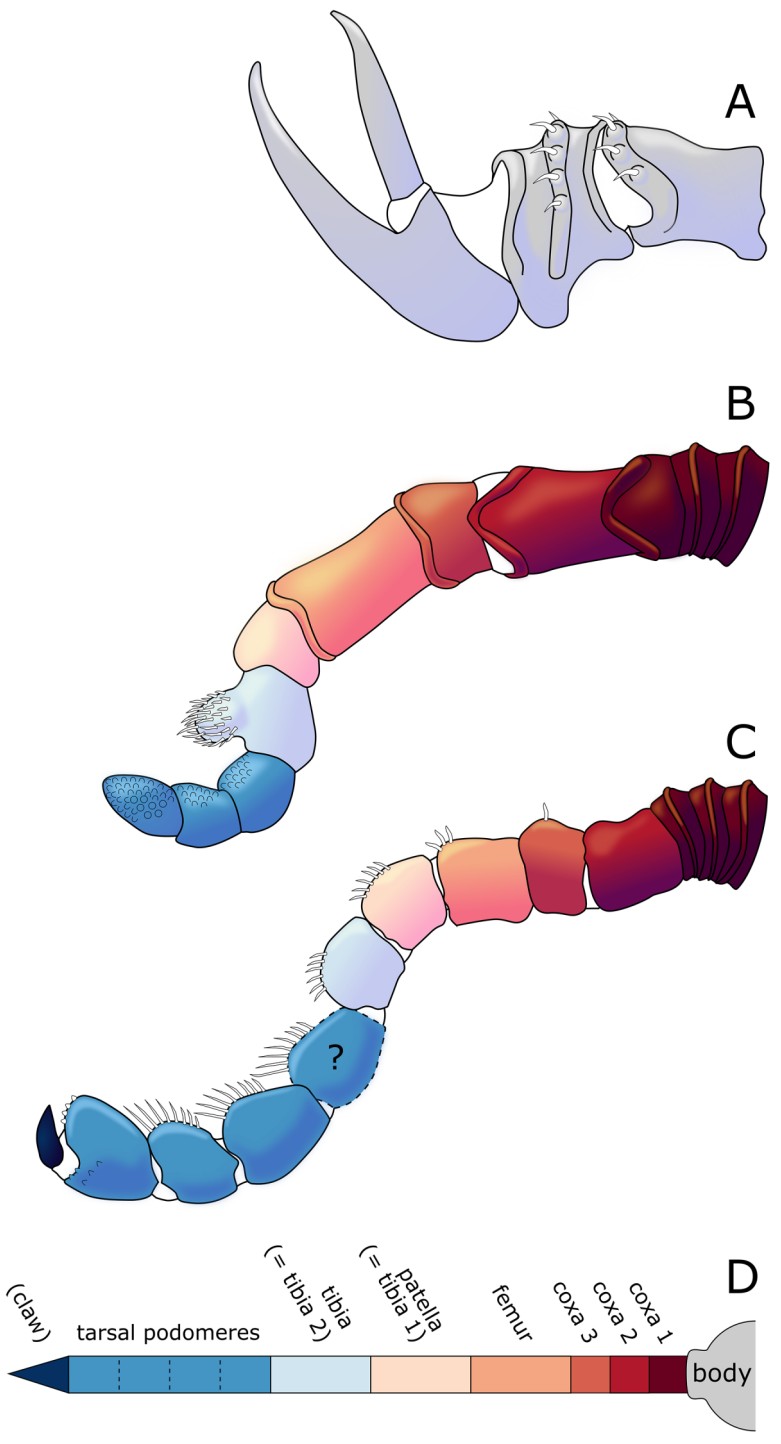

**Figure 14 Diagram of the interpreted structure of the cephalic appendages of *Palaeoiospus problematicus*.** (A) Chelifore. (B) Palp. (C) Oviger. (D) Colour code for putative podomeres homology in palps and ovigers, following the nomenclature of *Sabroux et al. (2023)* and *Siveter et al. (2023)* (modified from *Sabroux et al., 2023* and *Siveter et al., 2023*).

**The ovigers** (Figs. 10, 13) are positioned ventrally, as in extant species. Ovigers are never completely preserved in any studied specimen, making it difficult to count podomeres. The limbs bear a distal claw (Figs. 13P, 13R, 13S). In specimen 3 of NHMMZ PWL 1995/35-LS (Figs. 13A, 13F, 13K, 13P), nine podomeres plus a podomere composed of three proximal rings (the coxa 1) can be counted; a part of the limb (from the seventh proximal podomere) is hidden under a leg. In the hidden part of the oviger, there is room for at least one more podomere, suggesting at least 11 podomeres (in agreement with *Bergström, Stürmer & Winter, 1980*). Also, the fourth podomere appears slightly larger than more distal ones. In the right oviger of specimen NHMMZ PWL 1994/133-LS (Figs. 13E, 13J, 13O, 13T), a similarly longer podomere is visible as the eighth most distal podomere. This supports the 11 podomeres hypothesis (fig. 14C). Most extant families (Ammotheidaec Dohrn, 1881; Ascorhynchidae Hoek, 1881; Callipallenidae Hilton, 1942; Colossendeidae Jarzynsky, 1879; Nymphonidae Wilson, 1878; Pallenopsidae Fry, 1978; Rhynchothoracidae Thompson, 1909; see, for example *Sabroux et al., 2023*) also have 11 podomeres, which has been suggested as the plesiomorphic state among Pantopoda (*Bamber, 2007*). We consider the count herein uncertain, however, due to the absence of any completely preserved and uninterrupted oviger. There is evidence of setae on the purported eighth podomere (Figs. 13C, 13H, 13M, 13R), on the fifth and sixth podomeres (Figs. 13A, 13F, 13K, 13P), and probably on some of the tarsal podomeres (Figs. 13B, 13G, 13L, 13Q). The distal margin of the last tarsal podomere (purported tenth podomere) forms a sort of vertical lamella ornamented with spines in at least some specimens (Figs. 13R, 13S), although this is not apparent in all specimens (Figs. 13A, 13F, 13K, 13P). Ovigers are one of the appendages in which sexual dimorphism, when present, is the most marked morphologically (*e.g.*, *Sabroux et al., 2023*), and this could be also the case here, although a simple difference in preservation seems as likely.

**The first pair of walking legs** (WL1; Fig. 15) has four proximal rings (see also Figs. 2, 7) interpreted as coxa 1 by *Siveter et al. (2023)*. These are followed by eight plain podomeres: the next podomere, coxa 2, is roughly trapezoidal, enlarging distally. It carries at least two ornamented ridges extending proximo-distally. Their exact position is difficult to interpret due to the flattening of the fossil that deformed the legs, but these are likely to be dorsal and anterior in position. The distal margin of coxa 2 is thickened. Coxa 3 is short, roughly cylindrical, its distal margin thickened markedly with a dorsal notch probably enabling the levator-depressor swing of the femur and patella. The femur is short, roughly triangular, its broader side ventral, marking a strong geniculation in the legs, and its distal margin thickened. The patella, tibia, tarsus, and propodus are strongly flattened laterally, as made clearly visible in specimens that have their legs preserved dorsally (Figs. 15L–15Q). The patella is characterised among these podomeres by its tapering proximal extremity. The ventral margin of the patella, tibia and tarsus distally presents a cluster of short setae near the insertion of the arthrodial membrane of the following podomere. Dorsally, these podomeres carry a (single?) row of short setae set on small tubercles. The propodus has at least two rows of denticles on its distal lamellar margin (Figs. 15I–15K). The claw is subchelate. In modern pycnogonids, the distal claw of legs (generally referred to as the

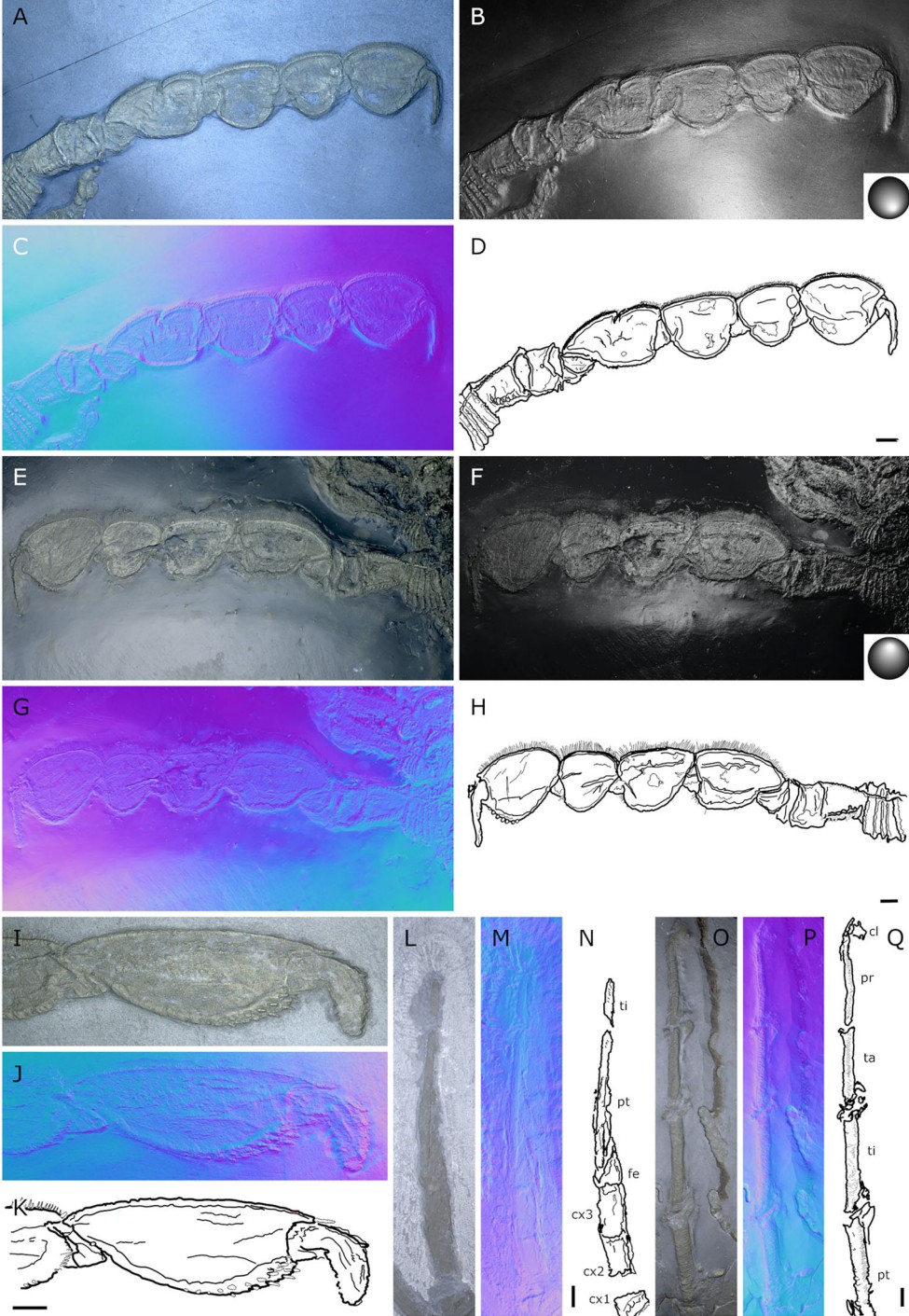

**Figure 15 *Palaeoisopus problematicus*, first walking pair of legs (WL1).** Specimens NHMMZ PWL 2008/141-LS (left WL1, lateral view: A–D), IGPB-HS636 (left WL1, lateral view: E–H), terminal podomeres of IGPB-HS582 (left WL1, lateral view: I–K; note the ventral spines), IGPB-AR-340 (left WL1, dorsal view: L–N), MB-A-46 (left WL1, dorsal view: O–Q). (A, E, I, L, O) Standard view. (B, F) Specular enhancement (direction of the light is indicated by the hemisphere on the bottom right or bottom left; see Material S3 for details). (C, G, J, M, P) Normals visualization. (D, H, K, N, Q) Interpretative drawings. *cx1-3*: coxa 1-3, *cl*: claw, *fe*: femur, *pt*: patella, *pr*: propodus, *ta*: tarsus, *ti*: tibia. Scale bars 5 mm.

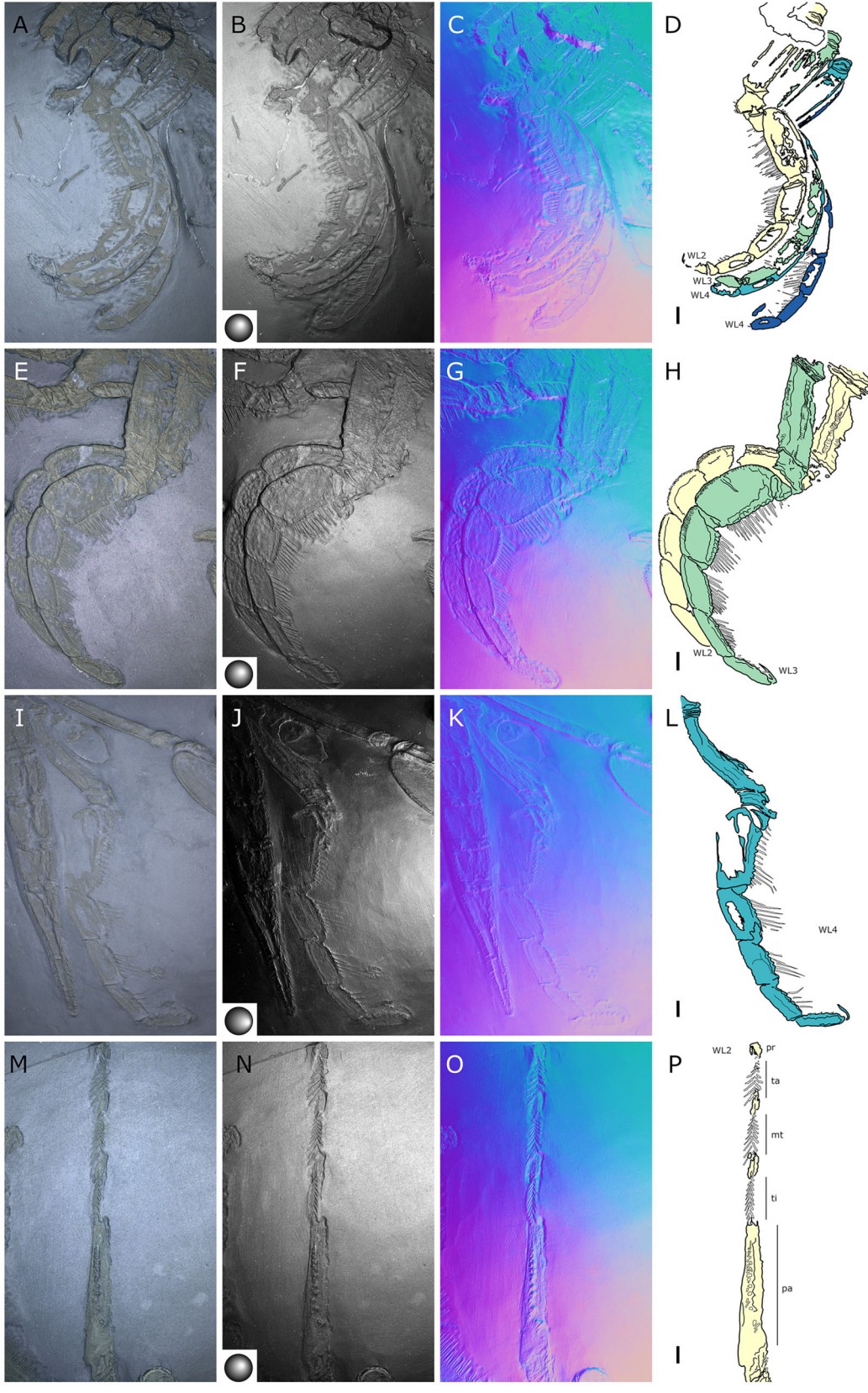

**Figure 16** *Palaeoisopus problematicus*, **second to fourth walking pairs of legs (WL2-4).** Specimens IGPB-HS1039 (A–D), IGPB-HS582 (specimen 2, right WL: E–H), MB.A.46 (right WL: IL) and IGPB-

**Figure 16** (continued)
HS582 (specimen 1, right WL: M–P). (A, E, I, M) Standard view. (B, F, J, N) Specular enhancement (direction of the light is indicated by the hemisphere on the bottom left, see Material S3 for details). (C, G, K, O) Normals visualization. (D, H, L, P) Interpretative drawings. *WL2-4*: walking legs 2–4. Scale bars 5 mm.               

main claw) is movable, so it is very much possible that the distal claw of *P. problematicus* WL1 was able to close on the distal indented margin of the propodus, making it a raptorial and/or cutting appendage.

**Walking legs two to four** (WL2-4; Fig. 16) differ from WL1 in presenting an additional podomere compared to WL1, as seen in *Haliestes dasos* (this additional podomere is referred to as a metatibia; *Siveter et al., 2023*). We propose that these two "additional" podomeres are homologous, and therefore, we follow the same morphological terminology. The coxa 1 presents three (WL2) or two (WL3-4) rings; it is shorter than broad. Coxa 2 is distinctly more elongated than in WL1, about 2.5 to 4.5 times as long as broad. It is ornamented like WL1, including rounded tubercles and short setae. Coxa 3 and the femur are short, about as long as broad, and the femur is triangular as seen in WL1. The patella is much more elongated than that of WL1: about twice as long as broad. The tibia, metatibia, tarsus, and propodus are all longer than broad. Podomeres from the patella to the tarsus presents two rows of long ventral setae (Figs. 16M–16P), as well as short dorsal setae. The propodus also carries setae, but it is unclear whether these are in one or several rows (Figs. 16E–16H). The terminal claw is subchelate.

**Juveniles:** six specimens of much smaller size are regarded as juveniles of this species, including the holotype (Figs. 1, 17, 18). They are often found in association with crinoids (*Hapalocrinus frechi* Jäkel, 1895 and *Parisangulocrinus minax* (Schmidt, 1934); see details in Material S1). Besides their smaller size, they differ from the adults in having a larger propodus on WL1, and the WLs are ornamented with only a few ventral setae rather than a dense range of ventral setae like in adults. Ventral setae of WL1 are also much longer than in adults and are disposed in a loose range along the length of WL1 rather than in a distal bouquet.

**Locality and age.** All fossils originate from the middle Kaub Formation in the Central Hunsrück Basin (consistent with their preservation), in the vicinities of Bundenbach and Germünden (see details in Material S1). Specimens IGPB-HS206, -HS207, -HS456, -HS457 and NHMMZ PWL 1992/178-LS are more precisely attributed to the Eschenbach quarry, IGPB-HS582, -HS636, -HS660, -HS694, NHMMZ PWL 2003/272-LS and 2008/141-LS to the Eschenbach-Bocksberg quarry, IGPB-M-142 to the Schmiedenberg quarry, and IGPB-HS942, -HS1039 and NHMMZ PWL 1998/155-LS to the Obereschenbach quarry. This dates the specimens to the Early Emsian.

*Palaeopantopus maucheri Broili, 1929*
Figures 20–24
v* 1929 *Palaeopantopus maucheri*: Broili, pp 272-274, 277, fig. 6, plate 5.

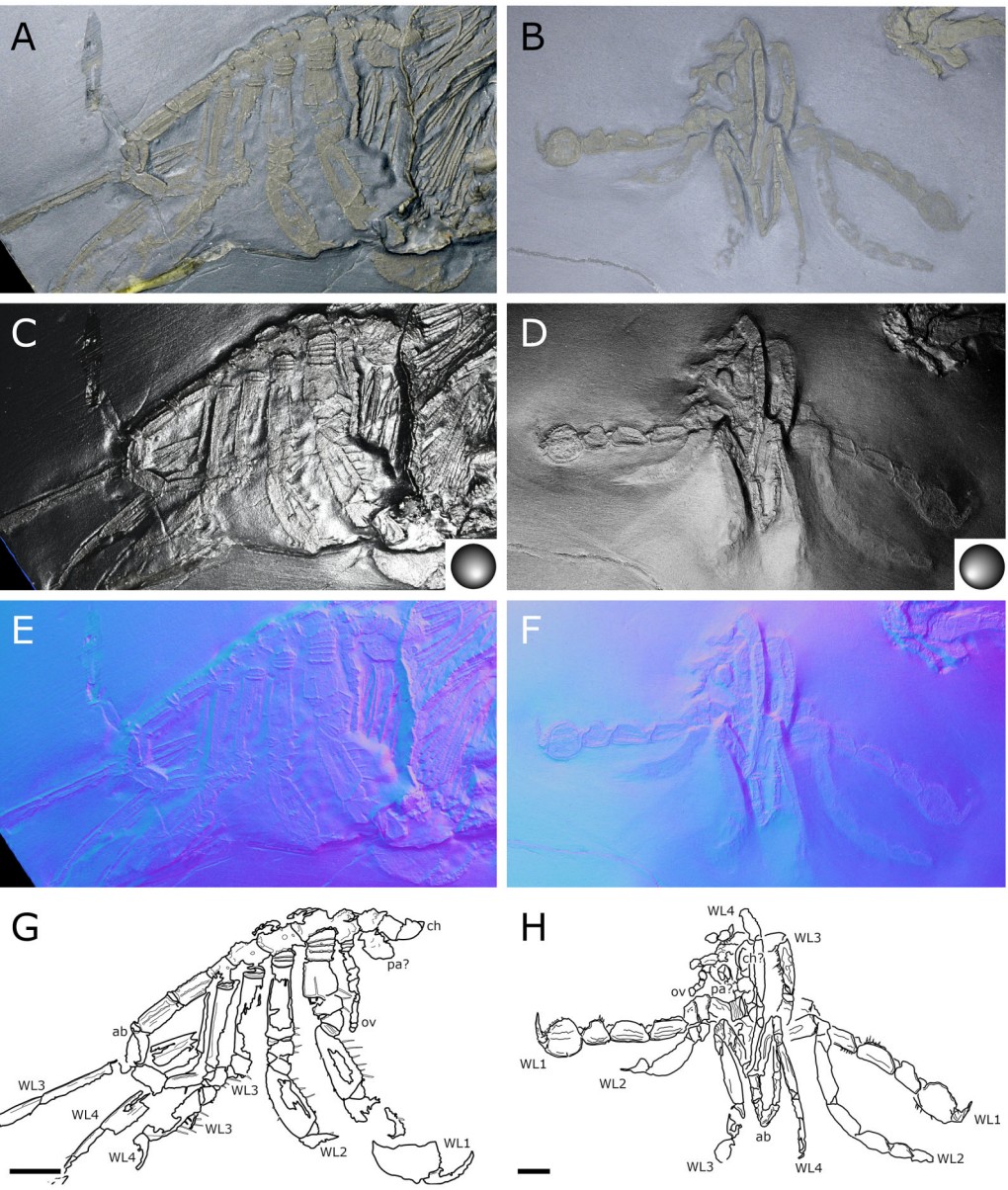

**Figure 17 Juvenile instars of *Palaeoisopus problematicus* (see also following illustration).** Specimens IGPB-HS942 specimen 1 (A, C, E, G) and NHMMZ PWL 2003/272-LS specimen 1 (B, D, F, H). (A, B) Standard view. (C, D) Specular enhancement (direction of the light is indicated by the hemisphere on the bottom right, see Material S3 for details). (E, F) Normals visualization. (G, H) Interpretative drawings. *ab*: abdomen, *ch*: chelifore, *ov*: oviger, *pa*: palp, *WL1-4*: walking legs 1–4. Scale bars 5 mm.

v 1930 *Palaeopantopus maucheri*: Broili, pp 209-213, figure.
*v 1932 Palaeopantopus maucheri*: Helfer, fig. 54.
*1954 Palaeopantopus maucheri*: Hedgpeth, pp 197 [text], 198 [text], 202 [text], 203 [text], fig. 4.
*1955a Palaeopantopus maucheri*: Hedgpeth, pp P169 [text], P170 [text], fig. 122.
*1957 Palaeopantopus maucheri*: Dubinin, p. 881, fig. 1Г, Д.

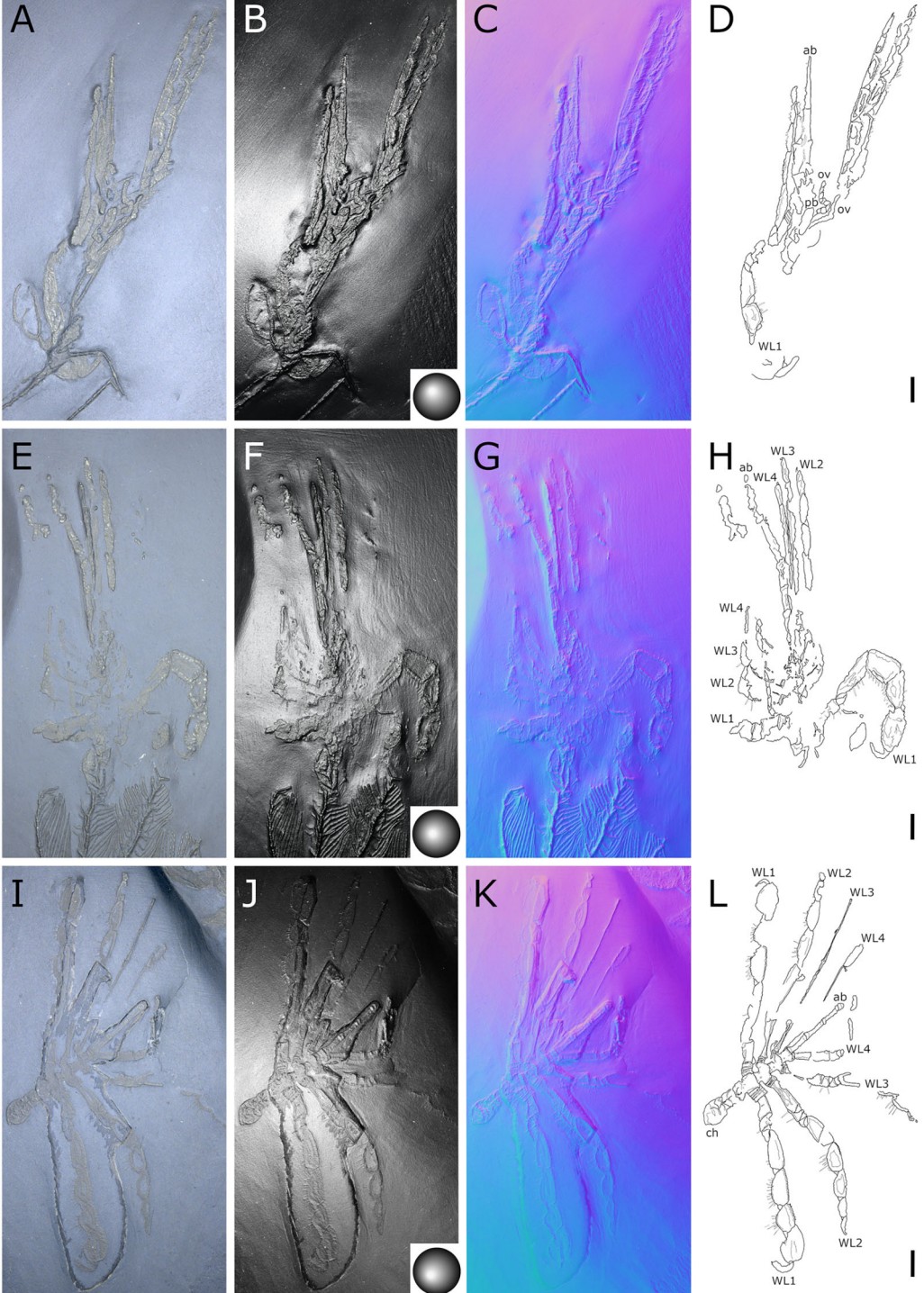

**Figure 18 Juvenile instars of *Palaeoisopus problematicus*.** Specimens NHMMZ PWL 2000/46 (A–D), NHMMZ PWL 1986/3 (E–H) and NHMMZ PWL 1994/54-LS specimen 1 (I–L). (A, E, I) Standard view. (B, F, J) Specular enhancement (direction of the light is indicated by the hemisphere on the bottom left, see Material S3 for details). (C, G, K) Normals visualization. (D, H, L) Interpretative drawings. *ab*: abdomen, *ch*: chelifore, *ov*: oviger, *pb*: proboscis, *WL1-4*: walking legs 1–4. Scale bars 5 mm.

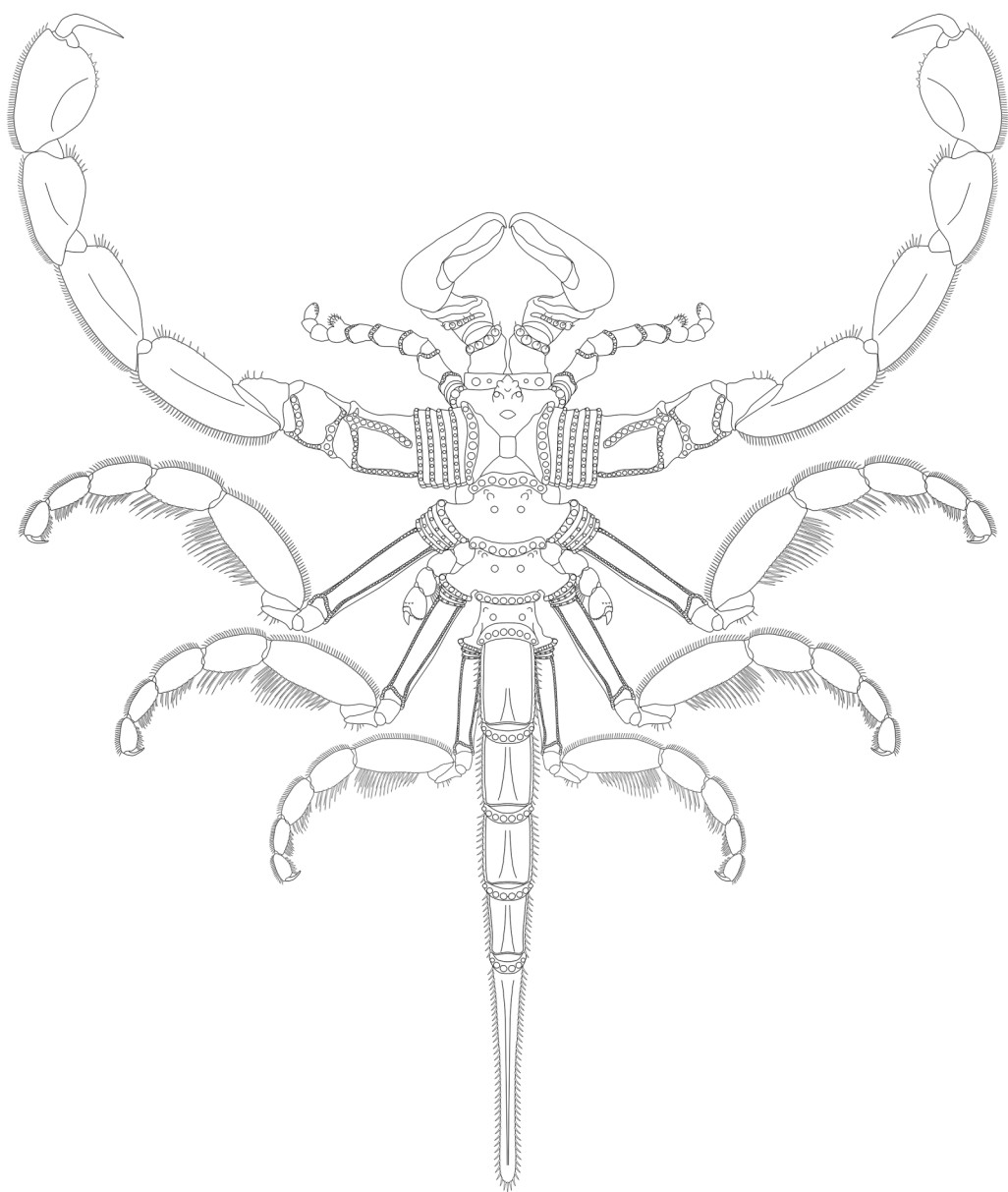

**Figure 19 *Palaeoisopus problematicus* reconstruction based on observed material, in dorsal view.** The number of podomeres of the chelifore scapes is speculative (see related section).

*1958 Palaeopantopus maucheri*: Tiegs & Manton, p. 321 [text].

*1978 Palaeopantopus maucheri*: Hedgpeth, pp 25 [text], 32 [text], fig. 3.

vp *1980 Palaeopantopus maucheri*: Bergström et al., pp 33-41, 46-48 [text], figs. 24-29.

*1998 Palaeopantopus maucheri*: Bartels et al., pp 155 [text; pro parte], 156 [text] [non fig. 133].

v *2017 Palaeopantopus maucheri*: Südkamp, pp 78, 79, fig. 121.

**Examined material. HOLOTYPE.** SNSB-BSPG 1929 V 3: 1 sp. **– OTHER.** SNSB-BSPG 1930 I 501: 1 sp., MB-A-45: 1 sp.

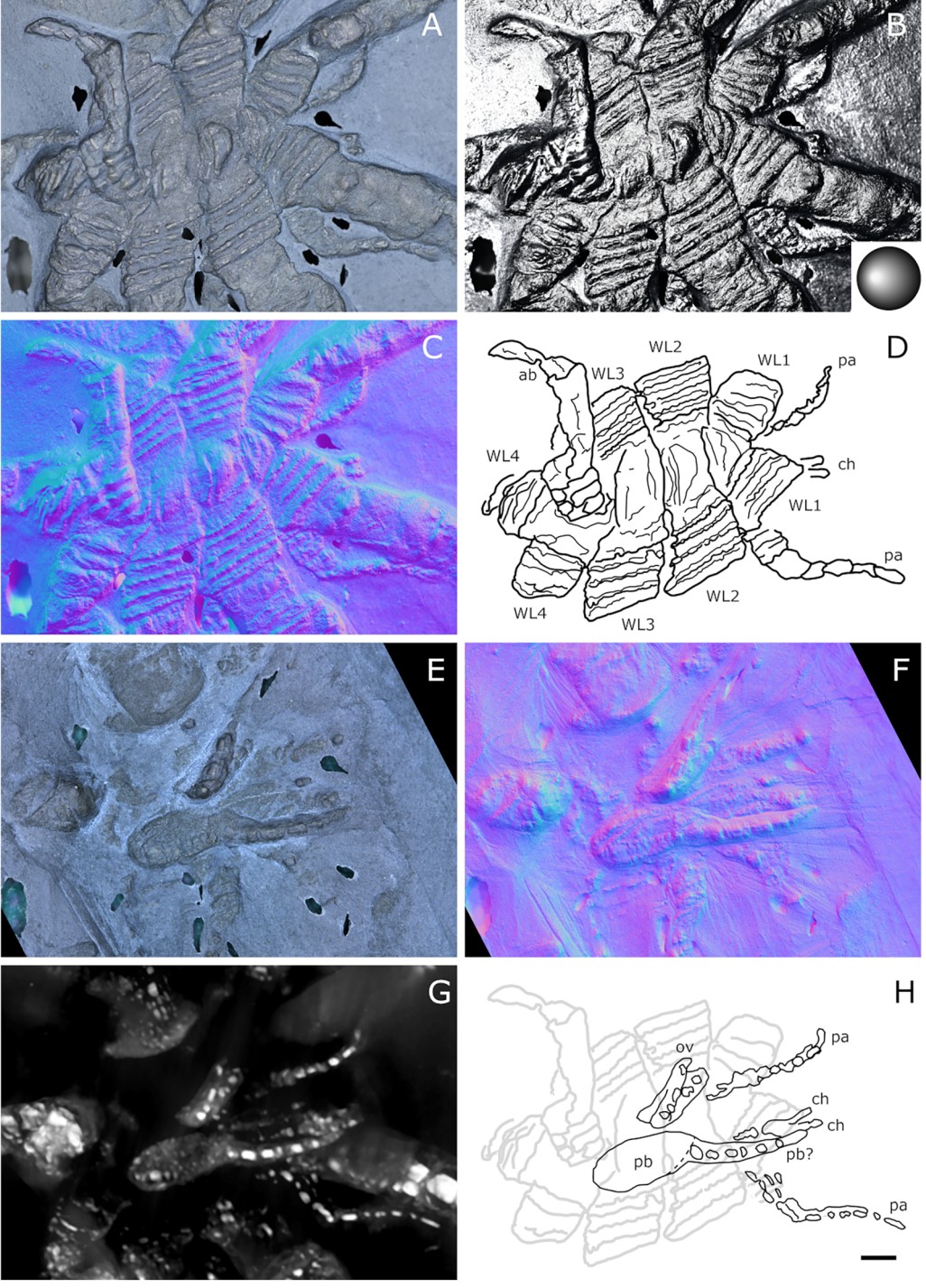

**Figure 20** *Palaeopantopus maucheri*, **holotype SNSB-BSPG 1929 V 3.** Dorsal (A–D) and ventral (E, F) views of the body. (A, E) Standard view. (B) Specular enhancement (direction of the light is indicated by the hemisphere on the bottom right, see Material S3 for details). (C, F) Normals visualization. (D) Interpretative drawing based of the fossil's surface. (G) Maximum intensity views of X-ray micro-tomography. (H) Interpretative drawing of structure revealed with X-ray, overlapped with surface structure (in grey). Pictures (E and F) were mirrored to align with (A–D, G and H). *ab*: abdomen, *ch*: chelifore, *pa*: palp, *pb*: proboscis, *ov*: oviger, *WL1-4*: walking leg 1–4. Scale bar: 2 mm.

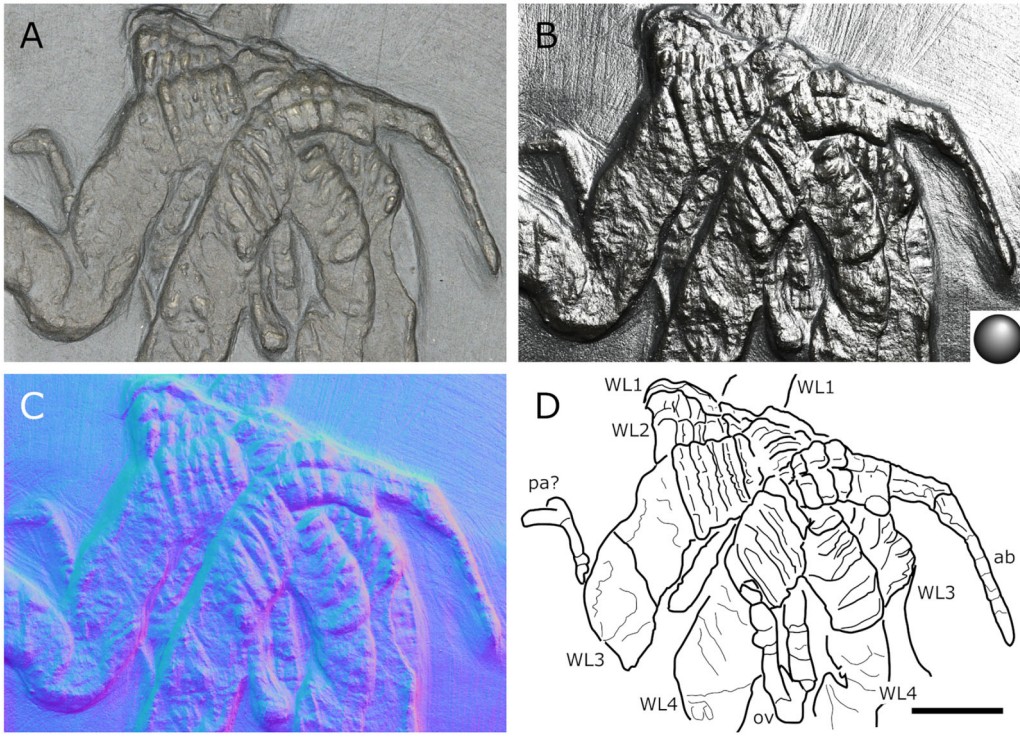

**Figure 21** *Palaeopantopus maucheri*, **specimen SNSB-BSPG 1930 I 501, posterior view of the body.**
(A) Standard view. (B) Specular enhancement (direction of the light is indicated by the hemisphere on the bottom right, see Material S3 for details). (C) Normals visualization. (D) Interpretative drawing based of the fossil's surface. *ab*: abdomen, *pa*: palp, *ov*: oviger, *WL1-4*: walking leg 1–4. Scale bar: 5 mm.

**Diagnosis.** Body robust, with lateral processes touching. Cephalon inconspicuous dorsally. Trunk unornamented. Abdomen at least three-segmented, with basal ring-like structures. Proboscis ventral. Chelifores, palps and ovigers present. Legs cylindrical in section, composed of at least 10 podomeres including a terminal claw, first podomere (coxa 1) divided into rings, second podomere (coxa 2) short, podomeres 4 to 8 elongate, joint between podomeres 4 and 5 immovable.

**Description and interpretation.** We accessed the three specimens described by *Broili (1929*: Fig. 20; *1930*: Fig. 21) and *Bergström, Stürmer & Winter (1980*: Fig. 22) and are aware of no additional material for this species. We propose that the specimen identified by *Bartels, Briggs & Brassel (1998)*, IGPD-HS437, does not belong to *Palaeopantopus maucheri* (see the dedicated paragraph in this section). *Bergström, Stürmer & Winter (1980)* also suggested that specimen MB-A-45 (Fig. 22) could be another species, distinct from *P. maucheri*, but RTI and microtomography reveal a body plan very similar in all well-preserved structures, except for a smaller size that may indicate different sex or life stage. The combination of RTI and X-ray microtomography data allows us to go further into details with the description of this species.
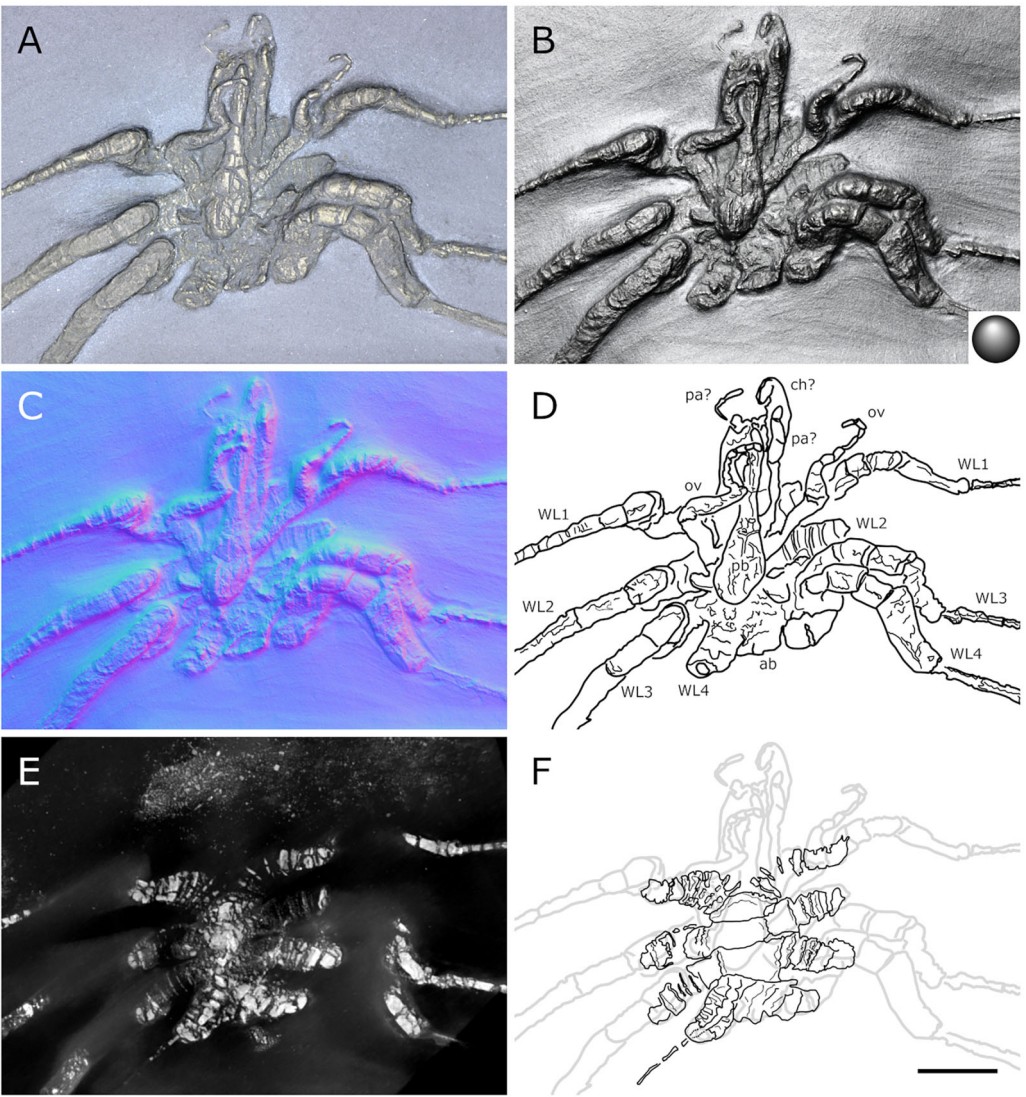

**Figure 22 *Palaeopantopus maucheri*, specimen MB-A-45, ventral view of the body.** (A) Standard view. (B) Specular enhancement (direction of the light is indicated by the hemisphere on the bottom right, see Material S3 for details). (C) Normals visualization. (D) Interpretative drawing based of the fossil's surface. (E) Maximum intensity views of X-ray microtomography. (F) Interpretative drawing of structure revealed with X-ray, overlapped with surface structure (in grey). *ab*: abdomen, *ch*: chelifore, *pa*: palp, *pb*: proboscis, *ov*: oviger, *WL1-4*: walking leg 1–4. Scale bar: 5 mm.

**The trunk** is compact, four-segmented, and broad. Lateral processes are very short and touching, their distal margin thickened (Fig. 20). The posterior margin of each trunk segment is slightly broadened. There is no evidence of a cephalon in dorsal or ventral view (Figs. 20, 22), though cephalic appendages and proboscis are present (see below).

**The abdomen** is composed of at least six elements (Figs. 20, 21), including three short, proximal elements, a median element of medium size, and two longer, distal elements. These two long distal elements are clearly identified as abdominal segments, with their

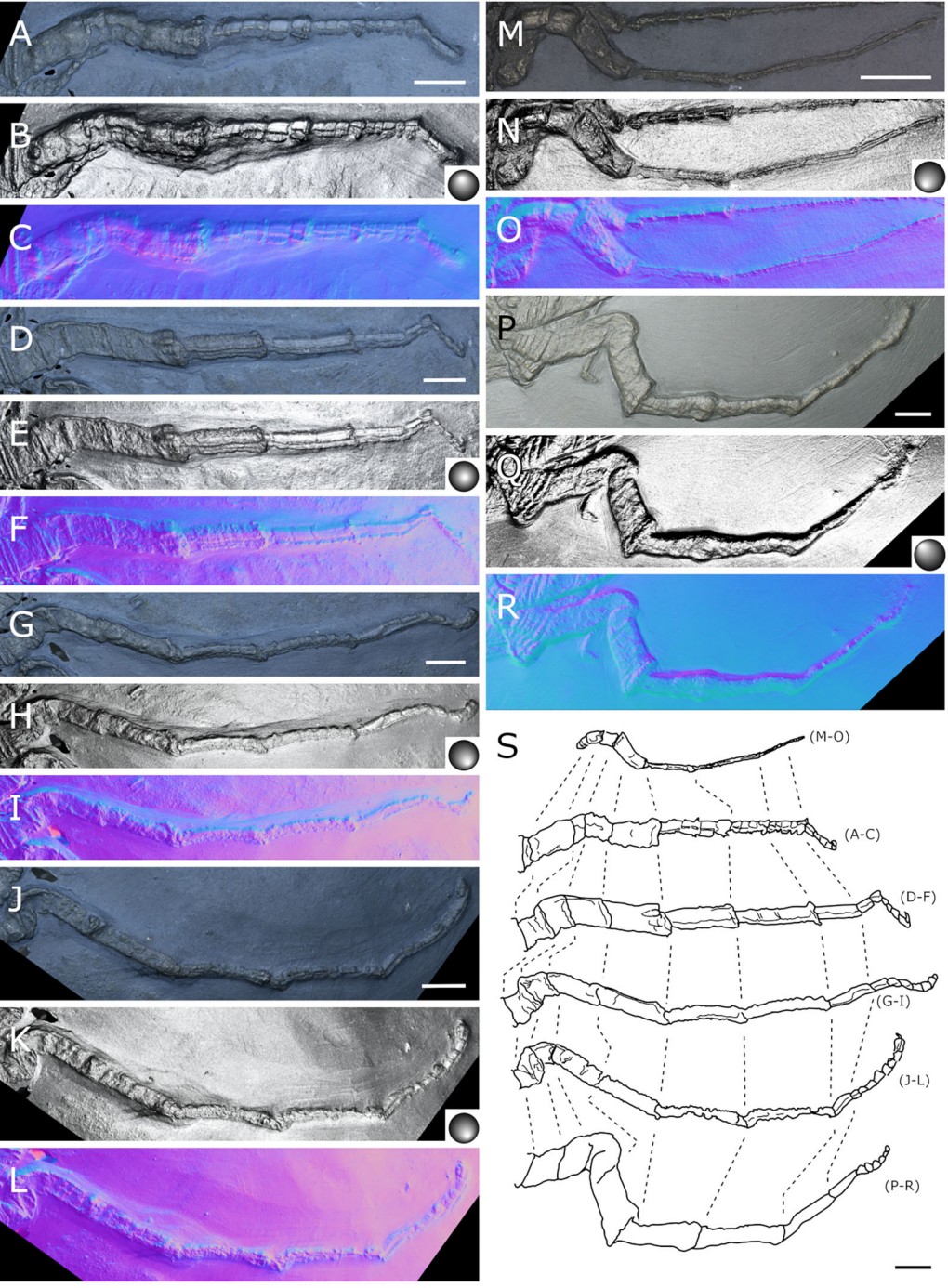

**Figure 23 *Palaeopantopus maucheri*, legs.** Holotype SNSB-BSPG 1929 right WL1 (A–C), right WL2 (D–F), right WL3 (G–I), right WL4 (J–L), specimens MB-A-45 left WL4 (M–O) and SNSB-BSPG 1930 I 501 left WL3 (P–R). (A, D, G, J, M, P) Standard view. (B, E, H, K, N, Q) Specular enhancement (direction of the light is indicated by the hemisphere on the bottom right, see Material S3 for details). (C, F, I, L, O, R) Normals visualization. (S) Interpretative drawings of all legs, put to the same scale. All scale bars 5 mm, B and C same as A; E, F same as D; H, I same as G; K, L same as J; N, O same as M; Q, R same as P.

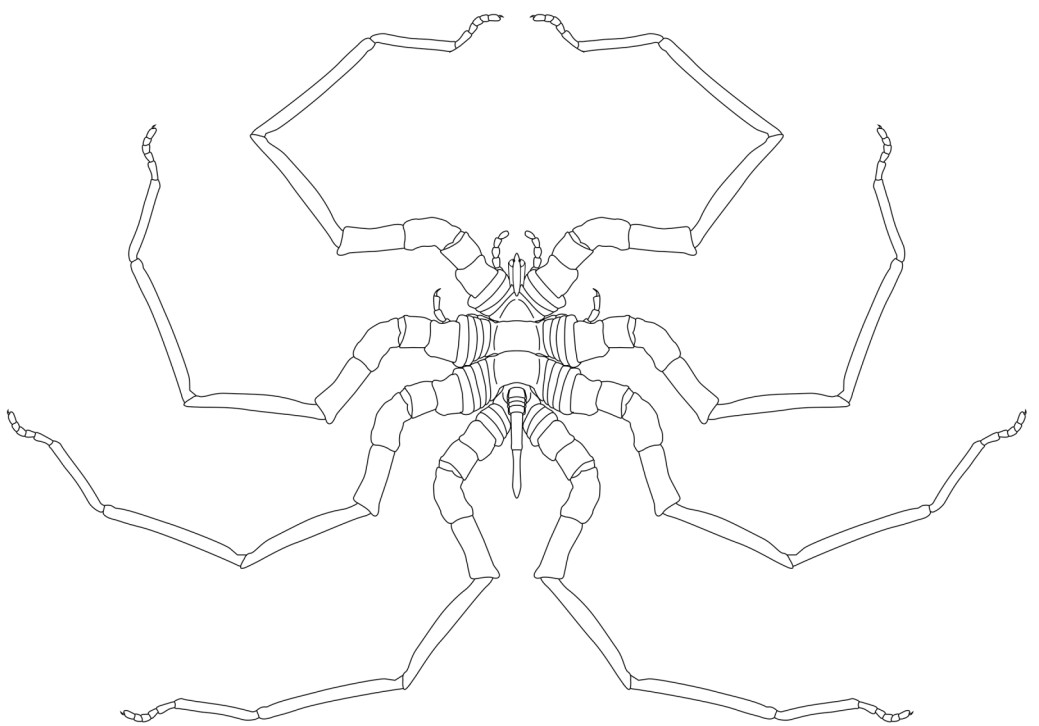

**Figure 24** ***Palaeopantopus maucheri* reconstruction based on observed material, in dorsal view.** The structure and number of podomeres in the cephalic appendages, as well as the terminal podomeres of the walking legs, are conjectural.

joint visibly bent in the two specimens where the abdomen is visible. The first three or four, proximal elements are reminiscent of the coxal rings of the walking legs (see below). It is not clear if their demarcation represents true segmentation and these four elements may correspond to one to four abdominal segments. *Bergström, Stürmer & Winter (1980)* suggested that the anus was terminal (hence regarding *P. maucheri* as a telson-less pycnogonid), based on their interpretation of internal pyritization as a trace of the digestive tract. However, this internal pyritization is also present in the palps and ovigers (Fig. 20G), while in extant pycnogonids the digestive tract never extends into these appendages (*Frankowski, Miyazaki & Brenneis, 2022*). Thus, we do not accept the internal pyritization to represent a digestive tract and so *P. maucheri* could have possessed a non-terminal anus (and thus a telson). The lanceolate tip of the abdomen, starting about after the first proximal fourth of the last abdominal segment length, could correspond in shape to an abdomen quite like that of *Palaeoisopus problematicus*. To our knowledge there is, instead, no instance of extant pycnogonids (which all show a terminal anus) with a long distal portion of the abdomen slender and lanceolate.

**The proboscis** of *P. maucheri* is challenging to delimit. There is little doubt that the roughly pyriform, rounded structure extending posteriorly on the venter (Figs. 20, 22) is part of this proboscis. However, the slender, tapering structure projecting anteriorly from it (Fig. 22) puzzled *Bergström, Stürmer & Winter (1980)* just as it puzzles us. A similar structure can also be observed with RTI on the ventral side of the holotype (Figs. 20E, 20F)

and its presence is confirmed by microtomography (Fig. 20G). *Bergström, Stürmer & Winter (1980)* ultimately interpreted this structure as a cephalic appendage, which is consistent with the internal pyritization seen in the holotype (Fig. 20G): in this case, it would be the oviger. Under this hypothesis, the proboscis is likely folded ventrally, with its distal mouth oriented posteriorly, and its base probably movable as in *P. problematicus*. However, the striking continuity of this structure with the proboscis cannot be ignored, especially in specimen MB-A-45 (Figs. 22A–22D). We suggest, as an alternative hypothesis, that this structure could also be the anterior part of the proboscis. If so, the slender, tapering projection is likely the distal part of the proboscis (and the rounded, pyriform part was proximal) as there is no evidence of any articulating petiole (such as found in *Eurycyde* Schiödte, 1857; *e.g.*, *Sabroux et al., 2019*) or anterior cephalic projection (such as found in some modern *Ascorhynchus* Sars, 1877, *e.g.*, *Clark, 1963*). Under this hypothesis, the round, broad proximal portion of the proboscis would be its base, implying that it inserts very posteriorly on the body, at the level of the third trunk segment (Figs. 20H, 22F). If correct, this condition is very different to extant species and it could perhaps be linked with the apparent disappearance of the cephalon in this species, which could in turn be linked with its ventralisation. Alternatively, perhaps the proboscis of *P. maucheri* possessed ventral ornamentation that would project posteriorly, as seen in the modern *Anoplodactylus californicus-digitatus* complex (*Arango & Maxmen, 2006*).

**Cephalic appendages** are all present. Chelifores are poorly preserved, and do not show any evidence of chelae (Figs. 20, 22). Palps are identified as such based on their anterior position, but their preservation is too poor to address their segmentation (*Bergström, Stürmer & Winter, 1980* identified at least seven podomeres) although their base clearly shows annular structures. The tip of the palps does not show any evidence of a terminal claw. The ovigers are positioned posteriorly to the palps. They present a hook distally (a strigilis?) followed by at least two broad podomeres (Fig. 22). *Bergström, Stürmer & Winter (1980)* counted nine podomeres, though we can neither confirm nor reject this assertion. The base of the ovigers is not visible.

**Walking legs** (Fig. 23) have annular structures at their base. *Bergström, Stürmer & Winter (1980)* suggested that these corresponded to the lateral processes rather than to the appendages in this species (unlike *P. problematicus*) based on the presumed absence of lateral processes. However, as indicated above, we identify the lateral processes as short but present. In addition, it seems unlikely that such similar structures could be convergent between *P. problematicus* and *P. maucheri* (as well as in *Halietes dasos* and other Hunsrück fossils as demonstrated below). We therefore interpret these rings as coxa 1, as in *P. problematicus*. Due to the potentially high number of podomeres in the legs of *P. maucheri*, we refrain from applying the morphological terminology of *Sabroux et al. (2023)* and *Siveter et al. (2023)* for WL (but see Discussion). The two podomeres following coxa 1 are relatively short (the second is slightly longer than the third in many cases) and the following fourth and fifth podomeres are always directed on the same axis so that their joint may not have been movable. These two podomeres are up to twice as long as the

proximal podomeres, the fifth longer than the fourth. The sixth podomere is narrower than the fifth, about three to five times as long as wide in the holotype. The seventh podomere is about 10% longer.

Distal podomeres are more difficult to define due to the distal fragmentation which either reflects anatomical segmentation or an artefact due to pyritization; the eighth podomere seems to be shorter, or as long as the seventh. It is followed by at least one, or possibly two, podomeres, which are then followed by a hook-like feature. This hook is variable in shape and its fragmentation suggests that it is composed of several, short podomeres, rather than made of a single curved podomere (such as the propodus of many extant species). Based on the number of fragments that can be counted, it composes at least four podomeres. However, taphonomic artefacts cannot be excluded to explain this fragmentation. These putative podomeres are followed by a terminal claw (Figs. 23A–23L, 23S).

**Locality and age**. All fossils originate from the middle Kaub Formation in the Central Hunsrück Basin, in the vicinities of Bundenbach (SNSB-BSPG 1929 V 3, 1930 I 501) and Germünden (MB-A-45; see also Material S1). This dates the specimens to the Early Emsian.

*Flagellopantopus blocki Poschmann & Dunlop, 2006*
Figures 25–27
v* 2006 *Flagellopantopus blocki*: Poschmann & Dunlop, pp 984-986, text-fig. 1.
V 2017 *Flagellopantopus blocki*: Südkamp, pp 79, 80, fig. 123.
V 2019 *Flagellopantopus blocki*: Sabroux et al., pp 1927 [text], 1929 [text], 1945 [text], Supplementary Material 3.

**Material. HOLOTYPE.** NHMMZ PWL 2004/5024-LS.

**Diagnosis.** Slender body. Proboscis pyriform (distally or totally?). Abdomen elongate, broad, four-segmented with a terminal flagelliform telson and two elongate processes on the second abdominal segment. Chelifores, palps and ovigers present, ovigers with a terminal claw. Four pairs of legs; first podomere (coxa 1) divided into rings, carrying one long dorsal process in the first pair of legs; second podomere (coxa 2) and podomeres 4 to 6 elongate; joint between podomeres 4 and 5 immovable.

**Description and interpretation.** This species described by *Poschmann & Dunlop (2006)* is, as far as we know, represented by a single specimen (Fig. 25; but see Discussion). Here, the CT-scan provides interesting insights on the body of *F. blocki*, and RTI helps us to improve our understanding of the structure of the legs relative to the accounts of *Poschmann & Dunlop (2006)* and *Sabroux et al. (2019)*.

**The trunk** segments (Figs. 25E, 25F) are individualised rather than fused. The cephalon carries a mediodorsal tubercle interpreted as the ocular tubercle (Fig. 25F), which has not been observed before. The distal margin of each segment is inflated, and ornamented with rounded tubercles, including one larger dorsomedial tubercle on each segment margin.

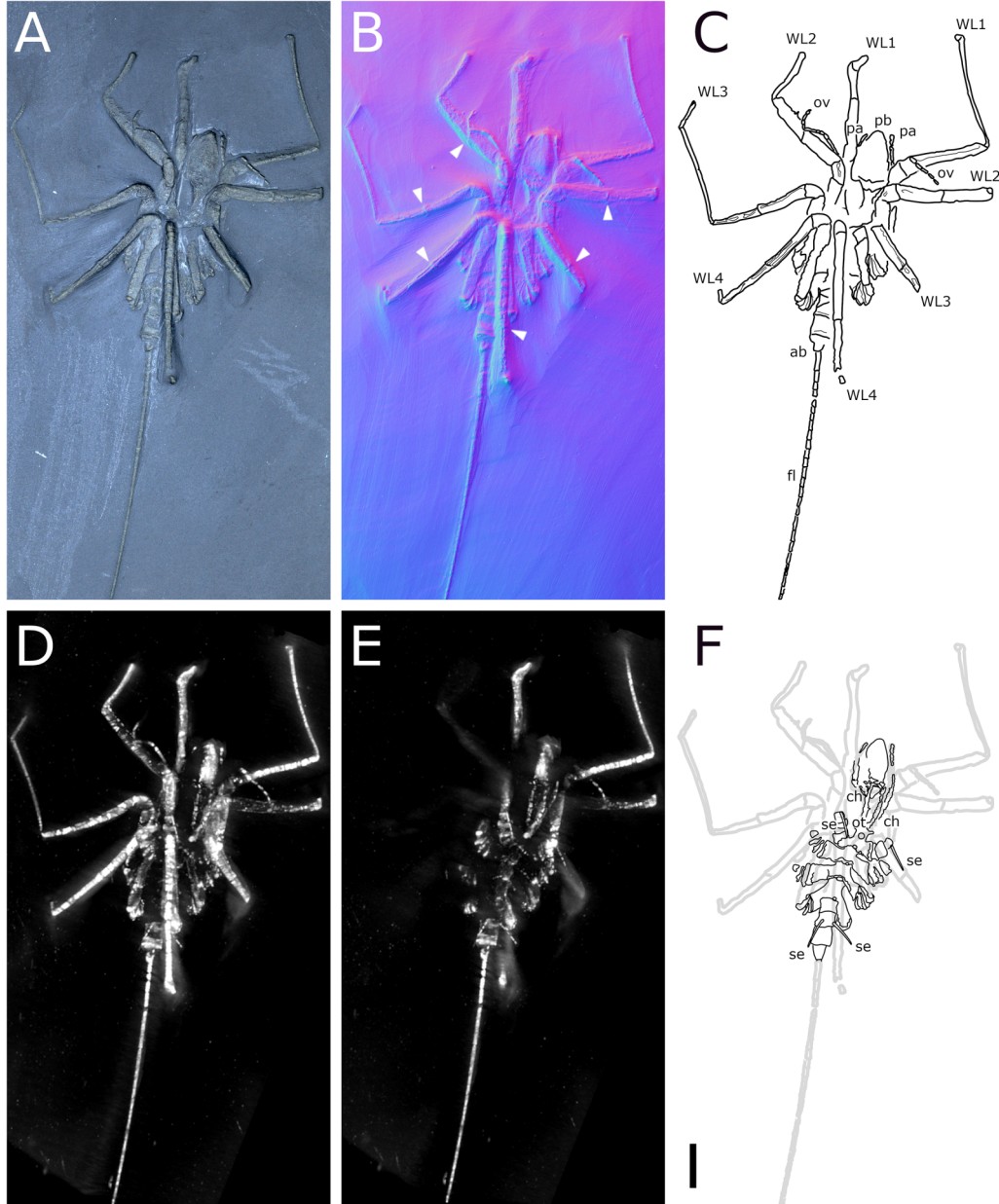

**Figure 25** *Flagellopantopus blocki*, **holotype NHMMZ PWL 2004/5024-LS, ventral view.** (A) Standard view. (B) Normals visualization, white arrowheads indicate the immobile joint. (C) Interpretative drawing based of the fossil's surface. (D, E) Maximum intensity views of Xray microtomography. (F) Interpretative drawing of structure revealed with X-ray, overlapped with surface structure (in grey). *ab*: abdomen, *ch*: chelifore, *fl*: flagellum, *ot*: ocular tubercle, *ov*: oviger, *pa*: palp, *pb*: proboscis; *se*: seta, *WL1-4*: walking leg 1–4. Scale bar: 5 mm.                                     

The first lateral processes are about as long as their basal width, the others about twice as long as the width of the base, separated by a distance shorter than their own diameter.

**The proboscis** is proximally obscured, even to X-ray imaging, but distally, it is pyriform in shape, tapering to a blunt tip. How the proboscis inserted on the cephalon is therefore not

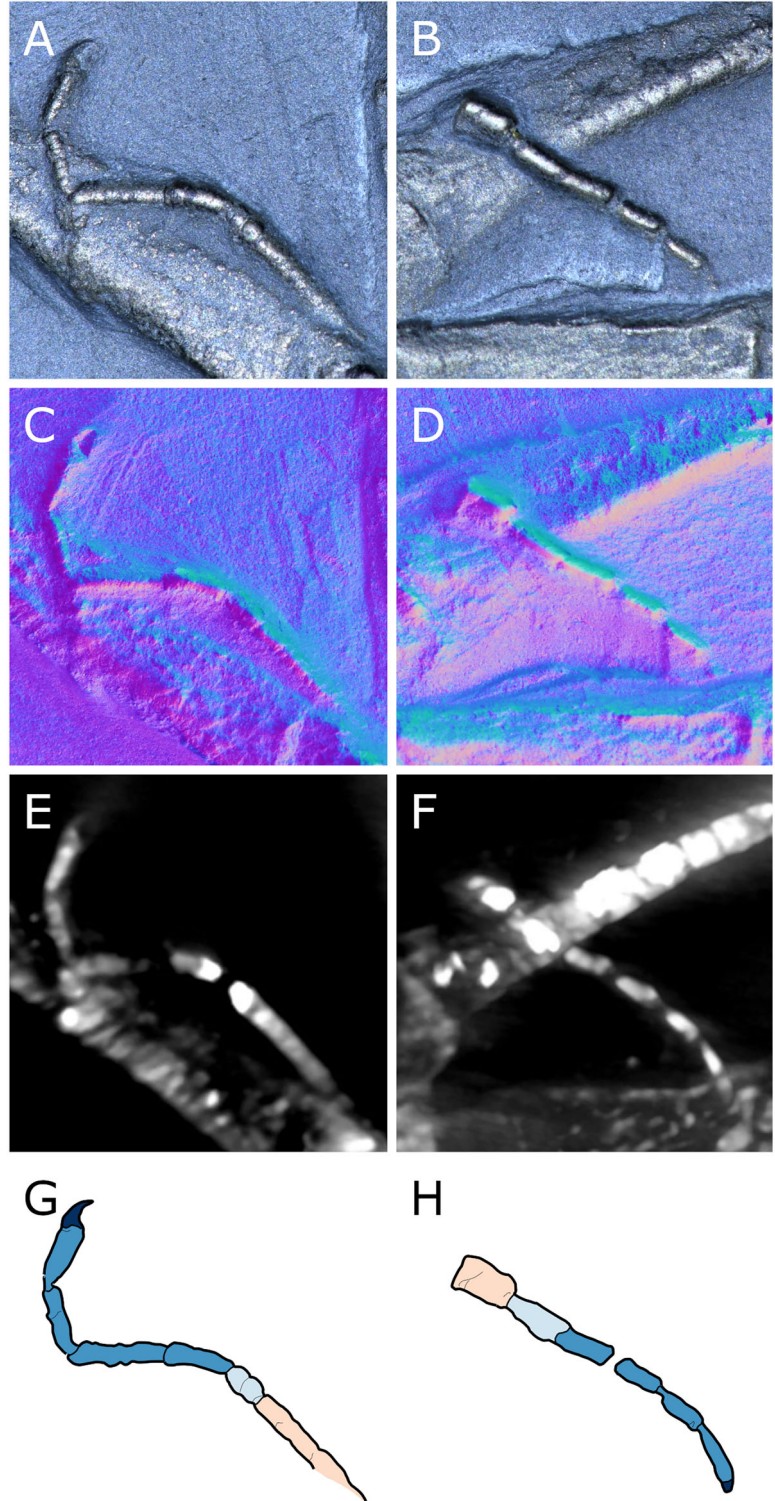

**Figure 26** *Flagellopantopus blocki*, **ovigers of holotype NHMMZ PWL 2004/5024-LS.** Right (A, C, E, G) and left (B, D, F, H) ovigers. (A, B) Standard view. (C, D) Normals visualization. (E, F) Maximum intensity views of X-ray microtomography. (G, H) Interpretative drawing, tentative podomere-per-podomere homology hypothesis indicated by colour codes (refer to Figs. 14 and 36). Scale bars 2 mm.

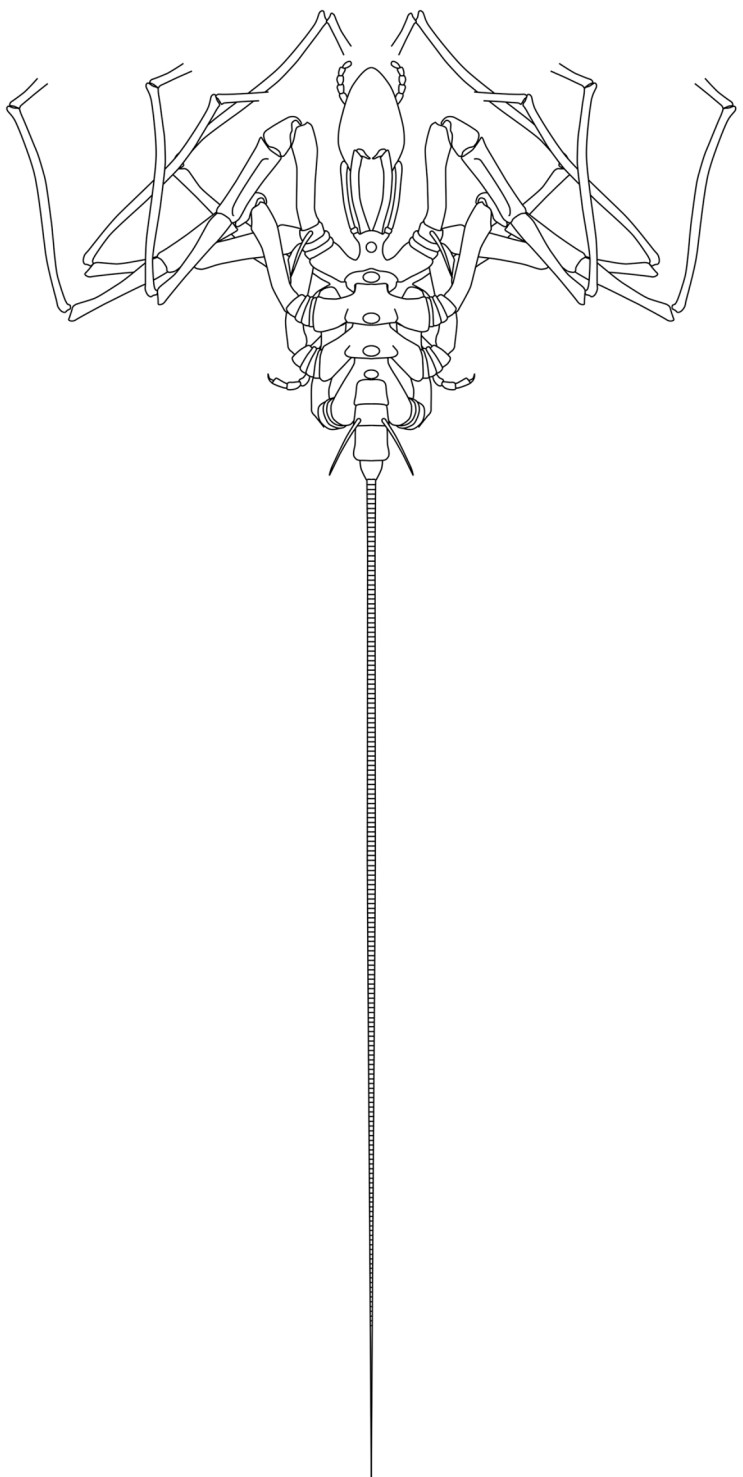

**Figure 27 *Flagellopantopus blocki*, reconstruction based on observed material, in dorsal view.** The proximal portion of the proboscis, and the structure of the chelifores and the palps are conjectural.

clear: its proximal part may be missing, or alternatively, the proboscis could have been simply detached during the decay of the specimen.

**The abdomen** is broad and segmented. It is composed of four large proximal segments about as long together as the last three trunk segments (the third abdominal segment slightly longer than the others), followed by a very long flagellum, more than three times the length of the body (from the cephalon to the abdomen). The flagellum is likely articulated, rather than rigid, as suggested by the curves along its length. Some traces of potential segmentation are also visible. The second abdominal segment carries two dorsolateral extensions near its distal margin that are likely to represent long setae.

**Cephalic appendages** are all present. Chelifores are long, with their chelae not preserved or absent. Palps reach the tip of the proboscis; their segmentation is poorly preserved. Ovigers (Fig. 26) have similarly poor preservation, although they have a strigilis (possibly four-segmented). This appears to have terminated with a small distal claw that is visible both with RTI and X-ray microtomography data.

**Walking legs** all share (Fig. 25B) the same structure as far as it is possible to identify from their preservation: the most complete leg in the holotype is the right third leg. Proximally, we suggest coxae 1 carry three rings, contra three podomeres as interpreted by *Poschmann & Dunlop (2006)*, concomitant with the previous interpretation of *Sabroux et al. (2019)*. However, RTI reveals that the remaining organisation of the leg is very different from the interpretation of these two references. The rings are followed by what we propose is a long podomere (about six times as long as wide), that was initially interpreted as two podomeres by *Sabroux et al. (2019)*. This is followed by a smaller podomere at a geniculation. The fourth and fifth podomeres were interpreted as a single segment by *Poschmann & Dunlop (2006)* and *Sabroux et al. (2019)*. RTI imaging, however, suggests that they are likely two podomeres sharing a single axis, their joint seemingly immovable (Fig. 25B). These are subequal in length, about five to six times as long as wide. The sixth podomere, slightly curved, is much slenderer and longer. The seventh podomere is probably only partially preserved. We refrain from using the morphological terminology of *Sabroux et al. (2023)* and *Siveter et al. (2023)* to assign identities to these podomeres (but see Discussion), as we did for *Palaeopantopus maucheri* with which it shares several structural resemblances. The third ring of the coxae 1 of the WL1 carries a thin projection, probably a seta.

**Locality and age**. The fossil is confidently attributable to the middle Kaub Formation of the Central Hunsrück Basin, but there is some confusion about its exact original location: according to *Poschmann & Dunlop (2006)*, it originates from the Wingertshell Member of the Obereschenbach quarry; it is instead recorded by the hosting institution as originating from the Eschenbach-Bocksberg quarry. In either case, this dates the specimen to the Early Emsian.

*Pentapantopus vogteli* Kühl, Poschmann & Rust, 2013
Figures 28–30

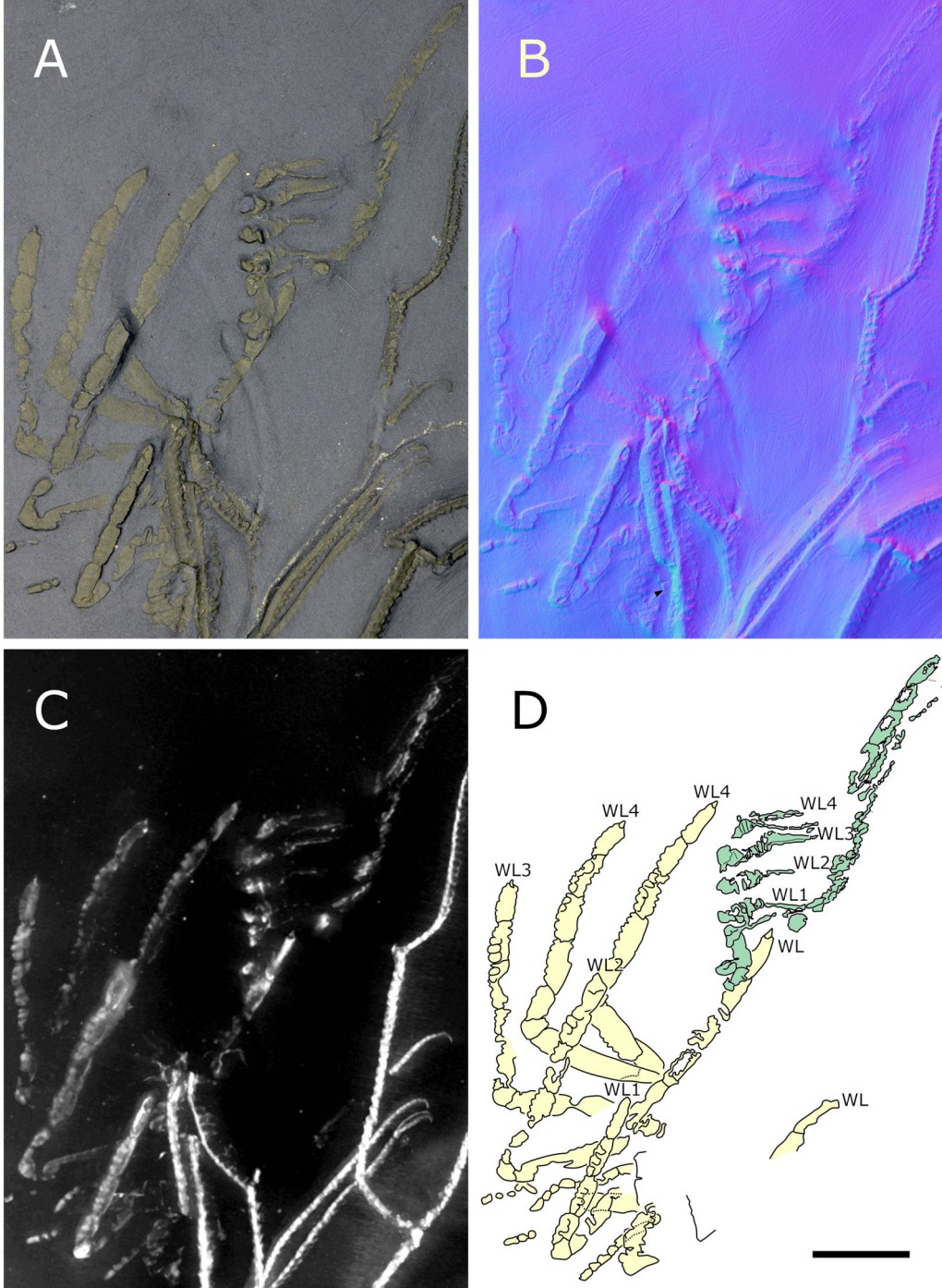

**Figure 28 *Pentapantopus vogteli* holotype and paratype NHMMZ PWL 2010/5-LS.** (A) Standard view. (B) Normals visualization. (C) Maximum intensity views of X-ray microtomography. (D) Interpretative drawing (holotype in yellow, paratype in green). *ch*: chelifore, *ov*: oviger, *pa*: palp, *pb*: proboscis, *WL1-4*: walking leg 1–4. Scale bar: 5 mm.

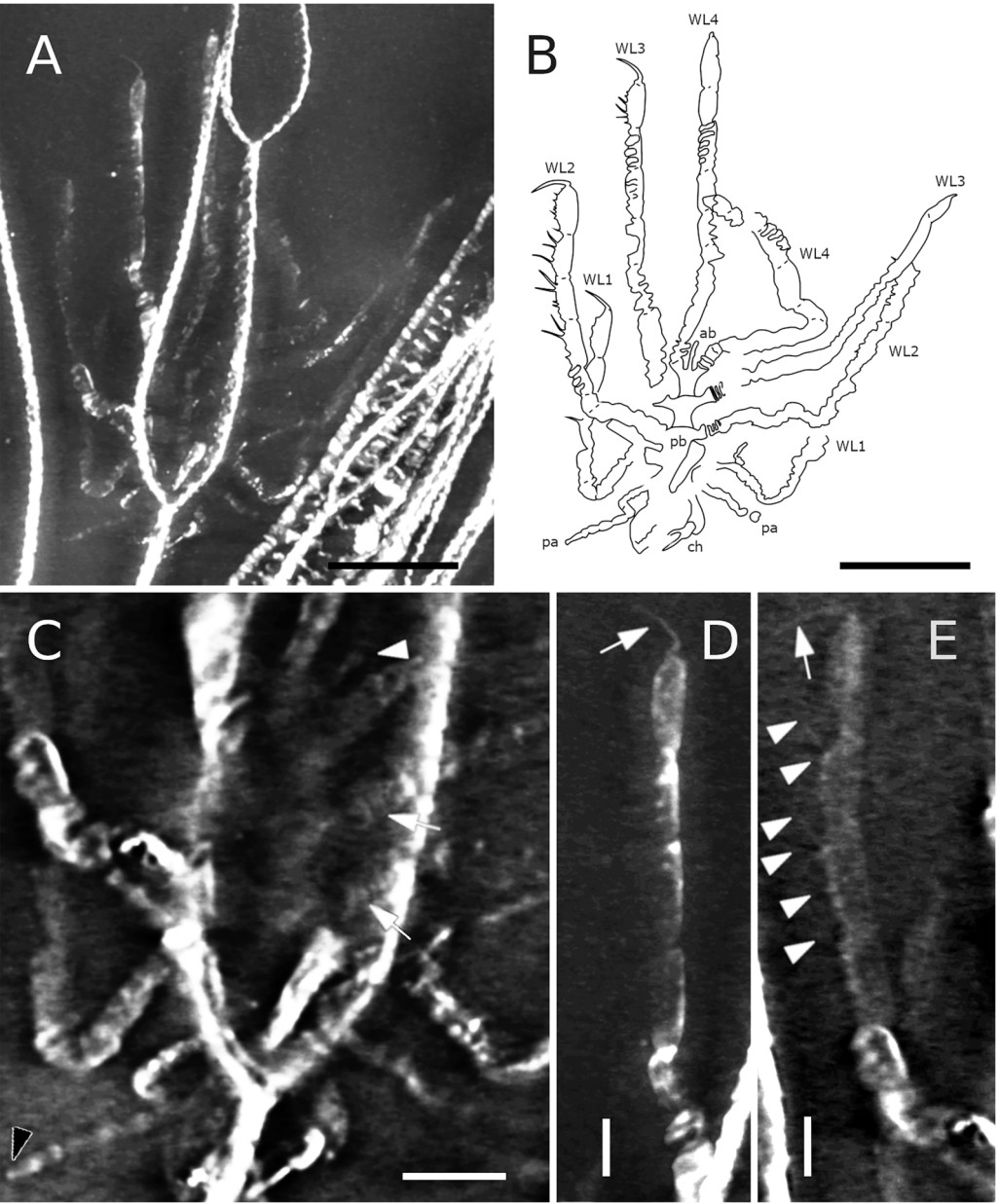

**Figure 29** *Pentapantopus vogteli*, **NHMMZ PWL 2010/5-LS, specimen 5.** (A, B) Whole specimen. (C) Close-up on the body (arrow: coxal rings, white arrowhead: abdomen, black arrowhead: palpal claw?). (D) Close-up on WL3 (arrow: terminal claw). (E) Close-up on WL2 (arrow: terminal claw, arrowhead: molariform indentations of the ventral side of WL). (A, C, D, E) Maximum intensity views of X-ray microtomography. (B) Interpretative drawing. *ab*: abdomen, *ch*: chelifore, *ov*: oviger, *pa*: palp, *pb*: proboscis, *WL1-4*: walking leg 1–4. Scale bars: 5 mm (A, B), 2 mm (C), 1 mm (D, E).

v* 2013 *Pentapantopus vogteli* : Kühl et al., pp 557, 558, figs. 1, 2.
v 2017 *Pentapantopus vogteli*: Südkamp, pp 79, 80 [pro parte], fig. 122c, d.

**Material. HOLOTYPE.** NHMMZ PWL 2004/5024-LS. – **PARATYPE.** Same slab, 1 specimen. – **OTHER.** Same slab, one specimen.

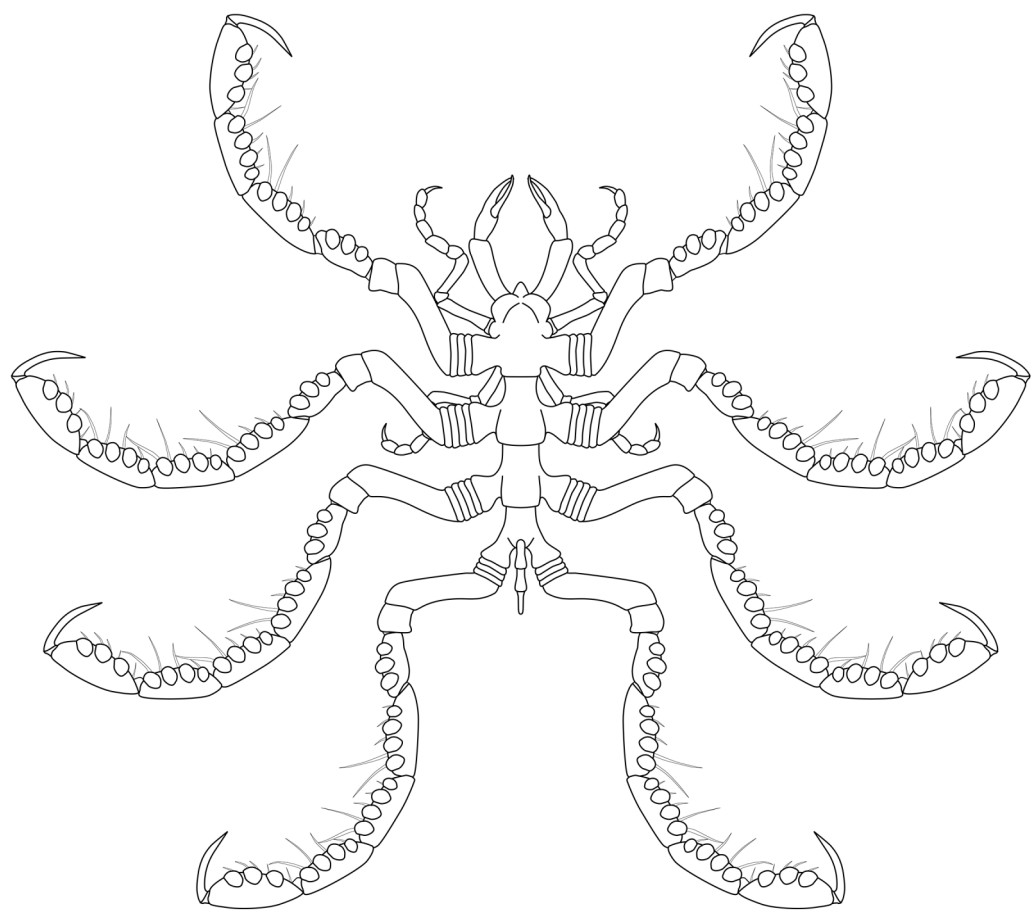

**Figure 30** *Pentanpantopus vogteli,* **reconstruction based on observed material, in dorsal view.** The articulation of the abdomen is conjectural, inspired by the morphologically similar *Haliestes dasos* (*Siveter et al., 2004*, *2023*); so is the presence of a claw at the tip of the palps, the structure of the palps and ovigers. The structure of the chelifores and the number of coxal rings are also speculative. The structure of the cephalon relies on the specimen NHMMZ PWL 2007/29-LS, which is possibly a representative of *Pentapantopus vogteli.*

**Emended diagnosis (modified from *Kühl, Poschmann & Rust, 2013*).** Slender body. Proboscis elongate. Abdomen not reaching beyond second coxa. Chelifores, palps and ovigers present. Four pairs of legs consisting of eight (first pair of legs) or nine (second to fourth pairs of legs) podomeres including a subchelate, terminal main claw; first podomere (coxa 1) divided into rings. Distal leg podomeres (femur to propodus) laterally flattened, ventrally setose, with ventral molariform indentations.

**Description and interpretation.** When first described (presented Fig. 28), *Pentapantopus vogteli* was considered to be the first 10-legged fossil pycnogonid (*Kühl, Poschmann & Rust, 2013*). While this condition is found in modern species (which are generically called polymerous, alongside 12-legged species; *Arnaud & Bamber, 1987*) it is extremely rare. In this description, we include a new specimen (Fig. 29), that was hidden amidst a crinoid (*Parisangulocrinus zeaeformis* (Follmann, 1887)) in the same slab where the holotype and paratype were found. Like *P. vogteli*, this specimen presents molariform

indentations on the ventral side of the WL (Fig. 29), and presents an overall similar leg structure, justifying its taxonomic assignment. However, it differs from the original description in that this specimen has four rather than five leg pairs, and as many trunk segments. Reinvestigating the holotype and paratype with RTI and X-ray microtomography (Fig. 28), we find no evidence of five trunk segments: the paratype (preserved laterally) exhibits four pairs of legs and the holotype (probably preserved ventrally) presents seven legs. It seems more likely that either the supposedly fourth or fifth left leg in the holotype corresponds to the fourth right leg and was folded on the other side during the burial of the specimen. As such, *P. vogteli* is eight-legged.

**The trunk** of *P. vogteli* is four-segmented, the cephalon fused with the first trunk segment. The lateral processes are about as long as wide, separated by about their diameter.

**The proboscis**, which is clearly visible in a new specimen (Fig. 29), is elongated. It is preserved folded ventrally, suggesting it is either movable or directed vertically.

**The abdomen** is present, and is relatively short compared to other Devonian species, reaching beyond the lateral processes; it is not possible to determine whether it is segmented.

**The cephalic appendages** are all present. The chelifores are composed of at least three podomeres, including at least one scape. Palps and ovigers have their segmentation poorly preserved and hardly discriminable in the holotype; the left palp of the new specimen (Fig. 29) potentially bears a terminal claw (Fig. 28C).

**The walking legs** have their coxae 1 annulated at least in WL2 and 3 (and very probably also in WL1 and 4), the number of their rings being difficult to determine (beyond a minimum of three in each). WL1 present seven other podomeres (including a terminal claw) after the ringed coxa 1 in the holotype, compared to eight in WL2-4; WL1 of the new specimen appears similar in condition, as the portion of the leg covered by WL2 is not large enough to hide a complete podomere. Given the peculiarity of this assemblage, we refrain here from naming these podomeres following the morphological terminology of *Sabroux et al. (2023)* and *Siveter et al. (2023*; but see Discussion). WL1 podomeres following coxa 1 includes, from proximal to distal: a long second podomere (about five times as long as wide); a short third podomere (about as long as wide); a short fourth podomere (about twice as long as wide) with two or three ventral molariform indentations; the next two podomeres (the fifth and sixth) present ventrally four or five molariform indentations each; the seventh podomere which also possibly presents molariform indentations; and a long terminal claw (the eighth podomere) that is incompletely preserved in the type material. This claw is subchelate, movable (as indicated by the different positions it shows in left WL1-3 of the new specimen; Fig. 29) and probably able to close on the sole of the propodus, as observed in many extant species. In WL2-4, one additional podomere relative to WL1 is present somewhere between the fifth and the eight podomeres (probably in seventh position; see Discussion). The fifth to seventh (WL1) and

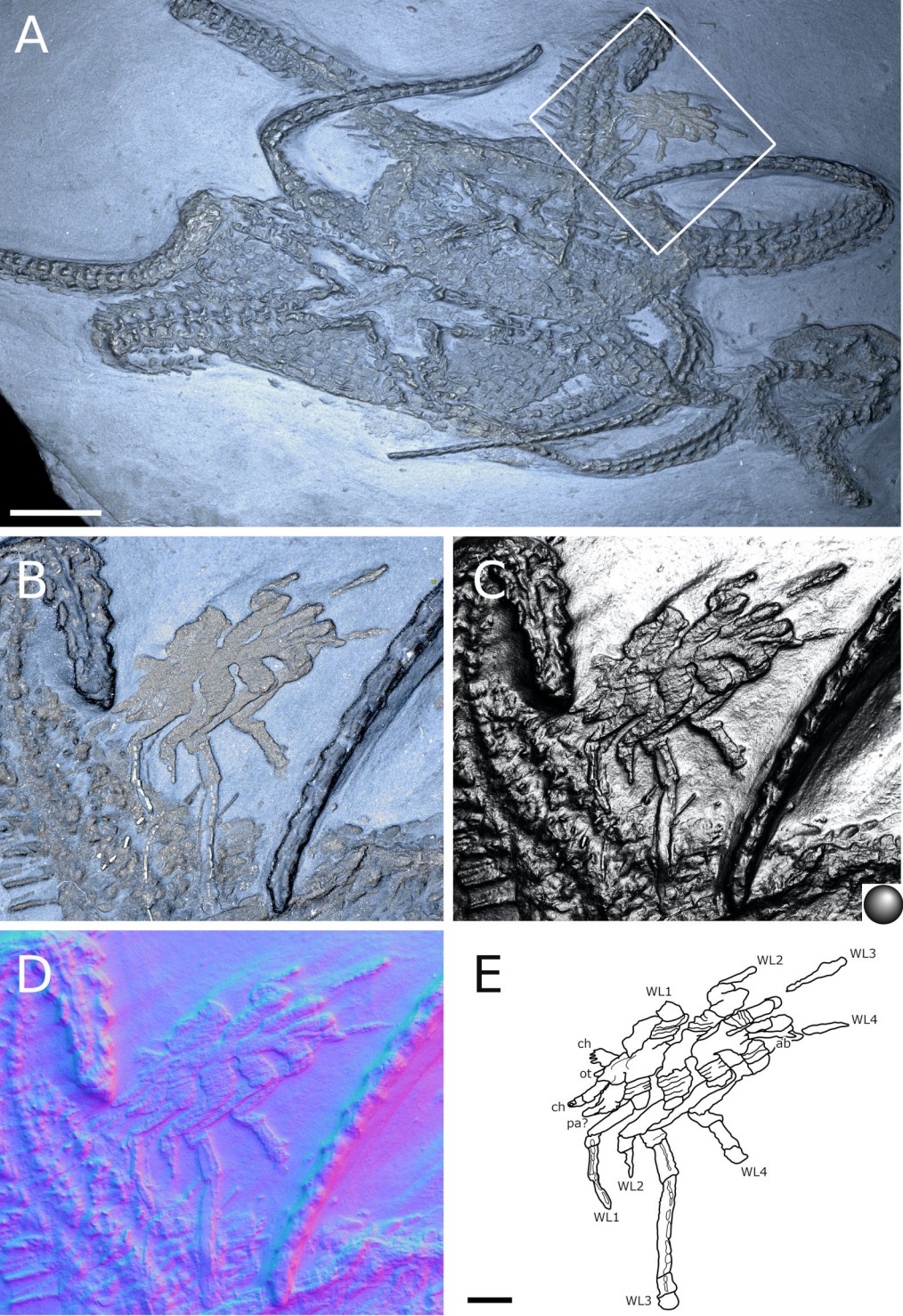

**Figure 31 Specimen NHMMZ PWL 2007/29-LS, dorsal view.** (A) Association with ophiuroid (*Bundenbachia beneckei* Stürtz, 1886). (B–E) Focus on the specimen. (A and B) Standard view. (C) Specular enhancement (direction of the light is indicated by the hemisphere on the bottom right, see Material S3 for details). (D) Normals visualization. (E) Interpretative drawing. *ab*: abdomen, *ch*: chelifore, *ot*: ocular tubercle, *ov*: oviger, *pa*: palp, *WL1-4*: walking leg 1–4. Scale bar: 10 mm (A), 2 mm (E). B, C, D same scale as E.

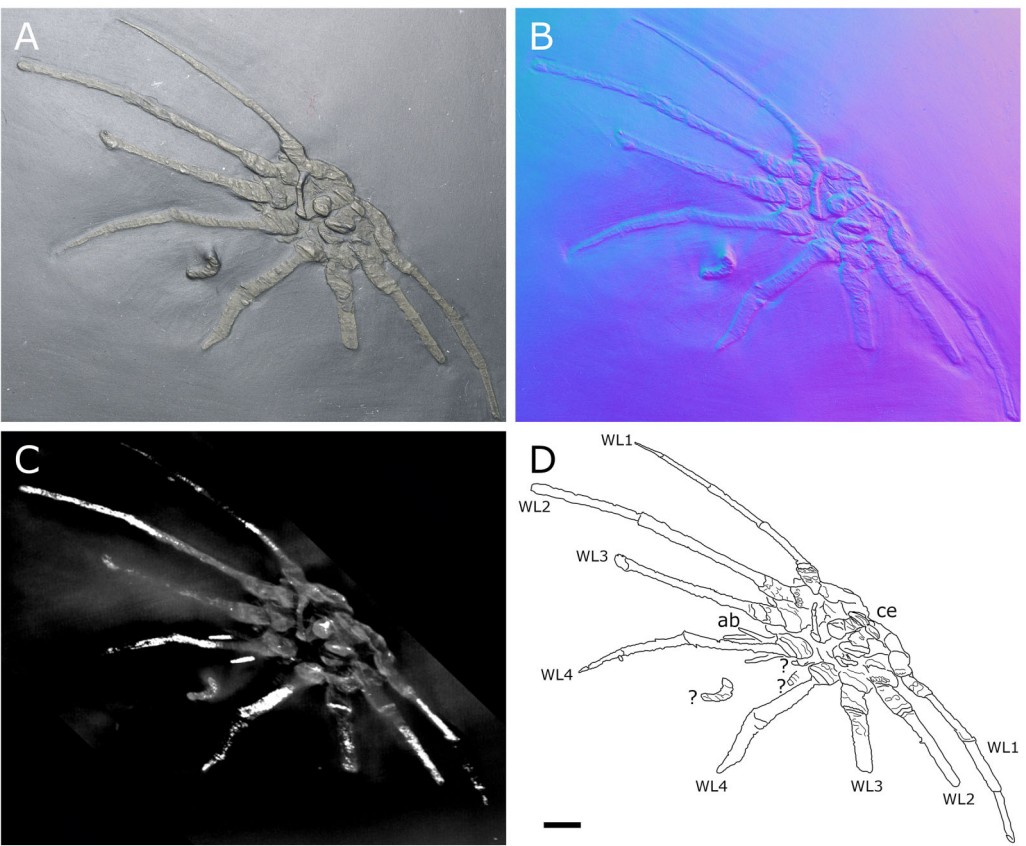

**Figure 32 Specimen IGPB-HS437, ventral view.** (A) Standard view. (B) Normals visualization. (C) Maximum intensity views of X-ray microtomography. (D) Interpretative drawing. *ab*: abdomen, *ce*: cephalon, *WL1-4*: walking leg 1–4. Scale bar: 5 mm.

fifth to eighth (WL2-4) podomeres present sparsely distributed long setae on the ventral surface that seem to occur in pairs (Fig. 29E).

**Locality and age.** The fossil is from the Central Hunsrück Basin, middle Kaub Formation, but here again there is some discrepancy between the literature and the data recorded by the hosting institution: it is either from the Wingertshell Member of the Obereschenbach quarry (*Kühl, Poschmann & Rust, 2013*); or from the Eschenbach-Bocksberg quarry (hosting institution data). In either case, the specimens date to the Early Emsian.

*Pentapantopus* ? *vogteli* ? *Kühl, Poschmann & Rust, 2013*
Figure 31

**Material.** NHMMZ PWL 2007/29-LS: 1 sp.

**Description and interpretation.** This pycnogonid is associated with two specimens of *Bundenbachia beneckei* Stürtz, 1886 (Fig. 31A). Its legs are not fully preserved and so it is not possible to address the presence of molariform indentations on the ventral margin of legs, as seen in the new *Pentapantopus vogteli* specimen (Fig. 29). Nevertheless, the two specimens have a very similar trunk outline. The number of annulations at the base of legs can be counted: four in WL1, five in WL2, four or five in WL3, and possibly three in WL4.

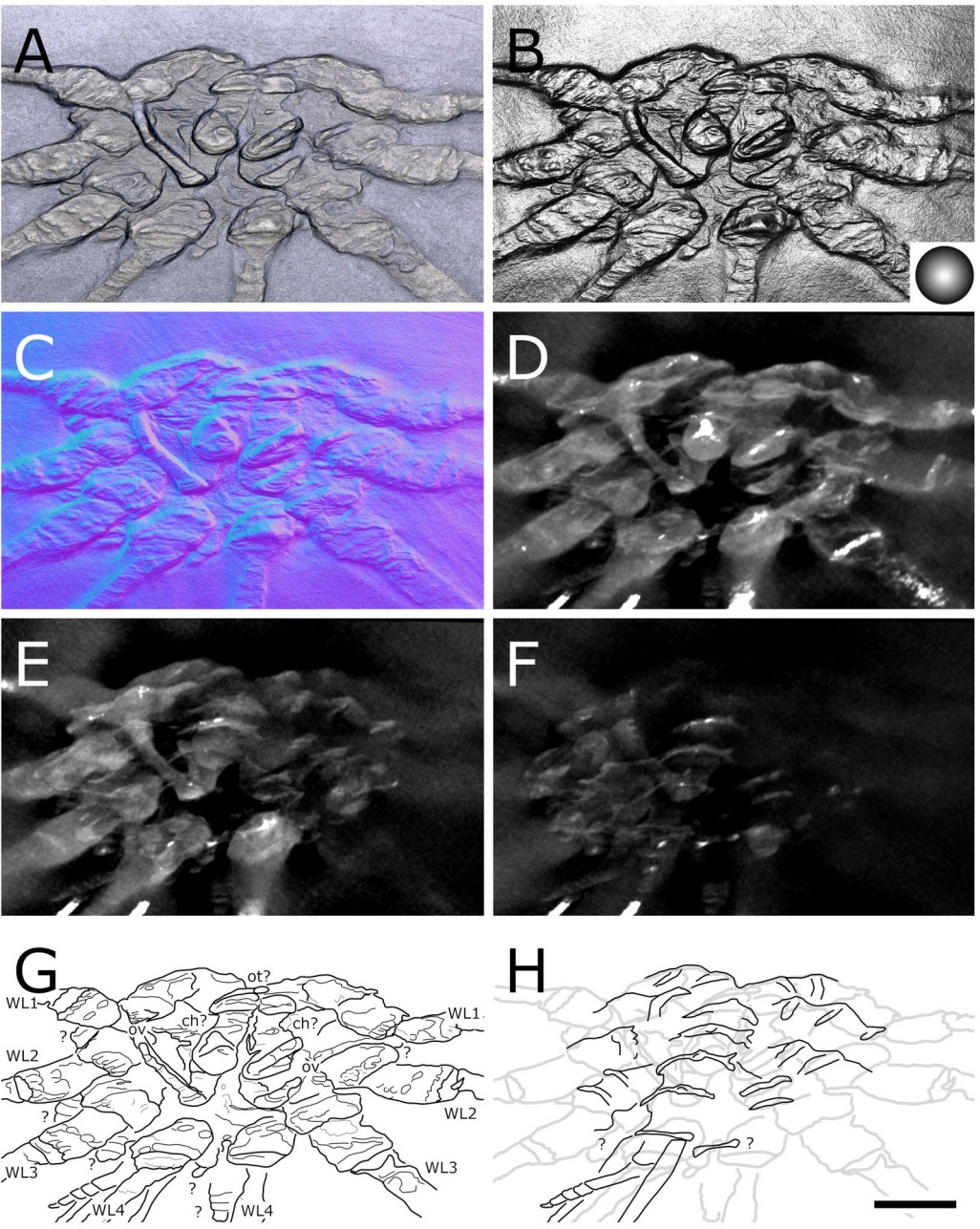

**Figure 33 Specimen IGPB-HS437, ventral view of the body.** (A) Standard view. (B) Specular enhancement (direction of the light is indicated by the hemisphere on the bottom right, see Material S3 for details). (C) Normals visualization. (D–F) Maximum intensity views of X-ray microtomography. (G) Interpretative drawing based of the fossil's surface. (H) Interpretative drawing of structure revealed with Xray, overlapped with surface structure (in grey). *ch*: chelifore, *ot*: ocular tubercle, *ov*: oviger, *WL1-4*: walking leg 1–4. Scale bar: 5 mm.

This morphology is compatible with *P. vogteli*. Distal to the rings is a short structure that we identify as a potential second podomere; but this could equally be an additional disarticulated ring, or arthrodial membrane. The cephalon is remarkably well preserved, bearing a conspicuous ocular tubercle pointing anteriorly, in a very similar fashion to *Haliestes dasos*. Chelifores are present, possibly as well as palps. The segmentation of the

abdomen is difficult to determine, its size and shape being very much like *P. vogteli* and *H. dasos*.

**Locality and age.** The fossil is recorded from Bundenbach in the Central Hunsrück Basin, and its preservation is typical of the middle Kaub Formation. This suggests the specimen dates to the Early Emsian.

Pycnogonida gen. sp.
Figures 32, 33
v. 1998 *Palaeopantopus maucheri*: Bartels et al., p. 155 [text; pro parte], fig. 133.

**Material.** IGPD-HS437: 1 sp.

**Description and interpretation.** This specimen (Fig. 32) was originally identified as the fourth representative of *Palaeopantopus maucheri* to be found (*Bartels, Briggs & Brassel, 1998*), but the two lateral projections inserted on the second abdominal segment suggest otherwise. While this specimen undoubtedly contrasts with other species described from Hunsrück (except, perhaps, *Palaeothea devonica*; see Discussion), its preservation does not allow for a sufficient list of diagnostic characters and, thus, for the description of a new species.

**The trunk** (Fig. 33) is four-segmented. The cephalon potentially has a low, rounded ocular tubercle. The posterior margins of the trunk segments widen, each possibly bearing a mediodorsal tubercle. The lateral processes are present, with a thickened distal margin.

**The proboscis** does not seem to be preserved.

**The abdomen** is multisegmented, with the two proximal-most segments being broad, about as long as wide, their posterior margin widening. The third segment is long (at least twice as long as broad), followed by up to three small segments about as long as wide. The second abdominal segment carries laterally two long processes, possibly setae.

**Cephalic appendages:** the chelifores are inserted anteriorly on the cephalon, but their number of podomeres is unclear. The chelae, if correctly identified, are comparatively large. The palps are either absent or unpreserved. The ovigers are positioned ventrally, with at least one posterior geniculation marking a joint, typically like the femoro-patellar geniculation of extant species (*Sabroux et al., 2023*).

**The four pairs of walking legs** consist of at least seven podomeres: three short ones, and four longer ones. The ultimate podomere visible in the right WL1 is either a terminal claw or it carries one. Proximally, we cannot be sure whether the specimen has coxa l rings like in other Devonian fossils.

**Locality and age.** The fossil is recorded from the Eschenbach quarry in the Central Hunsrück Basin. Its preservation is typical of the middle Kaub Formation. This suggests the specimen dates to the Early Emsian.

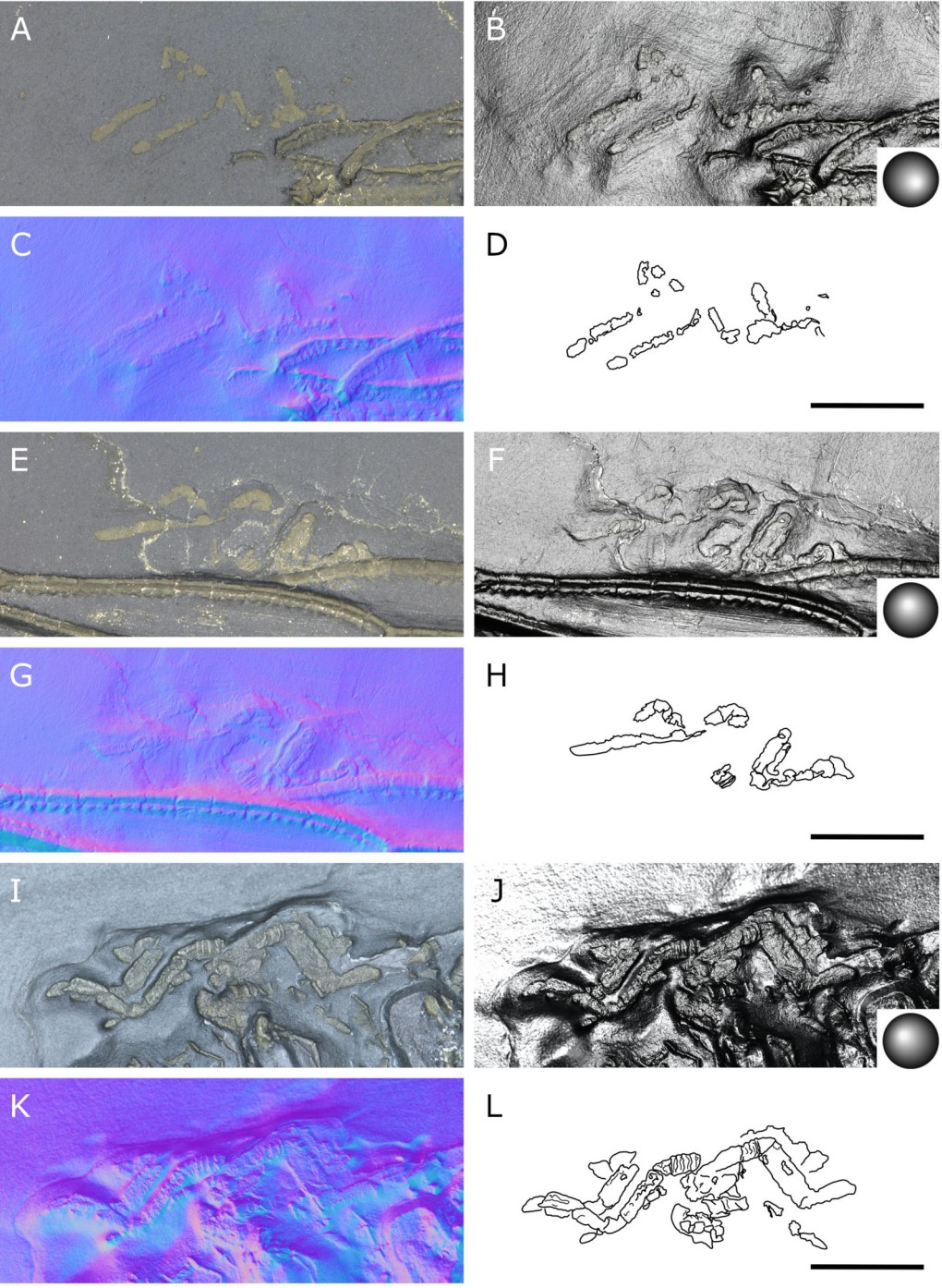

**Figure 34 Undetermined pycnogonids.** NHMMZ PWL 2010/5-LS (specimen 3: A–D: specimen 4: E–H) and IGPB-HS942 specimen 2 (I–L). (A, E, I) Standard view. (B, F, J) Specular enhancement (direction of the light is indicated by the hemisphere on the bottom right, see Material S3 for details). (C, G, K) Normals visualization. (D, H, L) Interpretative drawing. Scale bars 5 mm.

Undetermined pycnogonids
Figure 34
v 2013 "undetermined pycnogonid": Kühl et al., p. 7, fig. 4.
v. 2017 *Pentapantopus vogteli*: Südkamp, pp 79, 80 [pro parte], fig. 122a, b.

**Material.** IGPB-HS942: 1 sp., NHMMZ PWL 2010/5-LS: 2 spp.

Three specimens with the typical outline of a sea spider are found associated with crinoids (*Parisangulocrinus zeaeformis* (Follmann, 1887); and *Hapalocrinus frechi* Jäkel, 1895; see details in Material S1), but cannot be confidently assigned to any known taxon, and are not complete enough to be described as a new species. Two of them were already described by *Kühl, Poschmann & Rust (2013)*. There is not any solid evidence to refer these three specimens to the same species, except a general similar outline and size. Two of the three specimens present clear evidence of coxal rings (Figs. 34E–34L). The rest of the preservation is insufficient to allow any taxonomic assignment or species description.

**Locality and age.** All fossils originate from the middle Kaub Formation in the Central Hunsrück Basin, consistent with their preservation, and can therefore be attributed to the Early Emsian. Specimens in NHMMZ PWL 2004/5024-LS either originate more specifically from the Wingertshell Member of the Obereschenbach quarry (according to *Kühl, Poschmann & Rust, 2013*), or from the Eschenbach-Bocksberg quarry (according to the hosting institution records).

## DISCUSSION

### Ecology of the Devonian pycnogonids

Most of the fossils of the middle Kaub Formation were benthic organisms that could not escape the entrapping mudflows (*Rust et al., 2016*). Pelagic species, although present, are rarer in these layers of exceptional preservation, and were probably transported and rapidly buried (*De Baets et al., 2013*). Therefore, pycnogonids found in Hunsrück were probably benthic, consistent with extant species. *Palaeopantopus maucheri*, *Flagellopantopus blocki*, and specimen IGPD-HS437 share cylindrical legs with extant taxa, suggesting they were walking on the seabed (note that this does not exclude the fossils being able to occasionally "swim" in the water column; *e.g.*, *Morgan, 1972*). *Palaeoisopus problematicus* and *Pentapantopus vogteli*, with their flattened legs and long ventral setae, were probably able to swim actively.

Adults of *P. problematicus* are found in great abundance (*Südkamp, 2017*), but juveniles are much rarer. While the possibility of a preservational bias toward larger specimens cannot be excluded, or that big and small specimens simply represent two different species with different frequencies, other scenarios can be considered. We note that four out of the six juveniles we examined were found associated with crinoids (Figs. 1, 17A, 17C, 17E, 17G, 18A–18H and Material S1). These associations were previously interpreted as indicating a life association (*e.g.*, *Bergström, Stürmer & Winter, 1980*), but given that this almost never happens with adults this would have been limited to juveniles (the sole association of an adult *P. problematicus* of which we are aware is that of specimen

NHMMZ PWL 2013/8-LS with the hexactinellid sponge *Retifungus rugens* Rietschel, 1970). Alternatively, these juveniles might have occasionally rested (or fed?) amidst crinoids' arms, and those doing so–and which could not swim away rapidly–were those trapped by mudflows. This explains both the rarity of juveniles and their relative frequency among crinoids. It implies that they were not typically living among the benthic species commonly trapped in the Central Hunsrück Basin, but possibly just above in the water column, or in shallower seabed communities (*Bartels et al., 2002*). Such a scenario could also be proposed for *P. vogteli* which is always found in association with echinoderms (Figs. 28, 29), although the species might simply have been as rare as *Palaeopantopus maucheri* and *Flagellopantopus blocki*, which are unambiguously living on the seabed.

## Was *Palaeothea devonica* a pantopod?

We could not locate the holotype of *Palaeothea devonica*. This specimen belonged to a private collection when described by *Bergström, Stürmer & Winter (1980)* and it may have been sold, or lost (C. Bartels, 2020, personal communication). To date, it is the sole specimen attributed to this species with confidence. Given their size, it is possible that the small unidentified sea spiders on the slab NHMMZ PWL 2010/5-LS observed by *Kühl, Poschmann & Rust (2013*; Figs. 34A–34H), or the smallest specimen of the slab IGPB-HS942 (Figs. 34I–34L), are related to this species. However, all of these specimens are only fragmentarily preserved. The holotype of *P. devonica* was never prepared (*Otto, 1999*) and was only studied and illustrated based on X-ray projections. The illustrations available of this fossil by *Bergström, Stürmer & Winter (1980)* provide very few informative details. For this reason, it is not possible for us to confidently attribute any specimen to *P. devonica* until the holotype can be identified and redescribed. If the disappearance of the holotype is confirmed, it will be necessary to identify this binomen as a *nomen dubium*.

Based on their observations, *Bergström, Stürmer & Winter (1980)* concluded that *P. devonica* had no basal leg annulations (= annulated coxae 1) and possessed three short coxae on each leg and chelifores with reduced chelae. Hence, they interpretated this species as a pantopod, even suggesting affinities with Ammotheidae Dohrn, 1881 *sensu Stock (1994)* (= Ascorhynchoidea Hoek, 1881 *sensu Bamber, 2007*) (a group now regarded as paraphyletic; *Ballesteros et al., 2021*; *Bamber, El Nagar & Arango, 2024*; *Sabroux, Corbari & Hassanin, 2023*). This morphological interpretation is questionable. Re-examination of the radiographs published by *Bergström, Stürmer & Winter (1980)* shows that the so-called coxae are very narrow, more than usual for these podomeres (Fig. 35); we could as well interpret these structures as annulated coxae 1. The abdomen is poorly preserved, but its broad base and the outline on the radiographs of *Bergström, Stürmer & Winter (1980)* indicate it was comparatively large. Affinities of *P. devonica* with Pantopoda are therefore dubious at best.

Interestingly, *P. devonica* shares many characters with *Flagellopantopus blocki* and/or the specimen IGPD-HS437: this includes the thickened posterior margin of the trunk segments, the four dorsomedian tubercles on trunk segments, and the number of

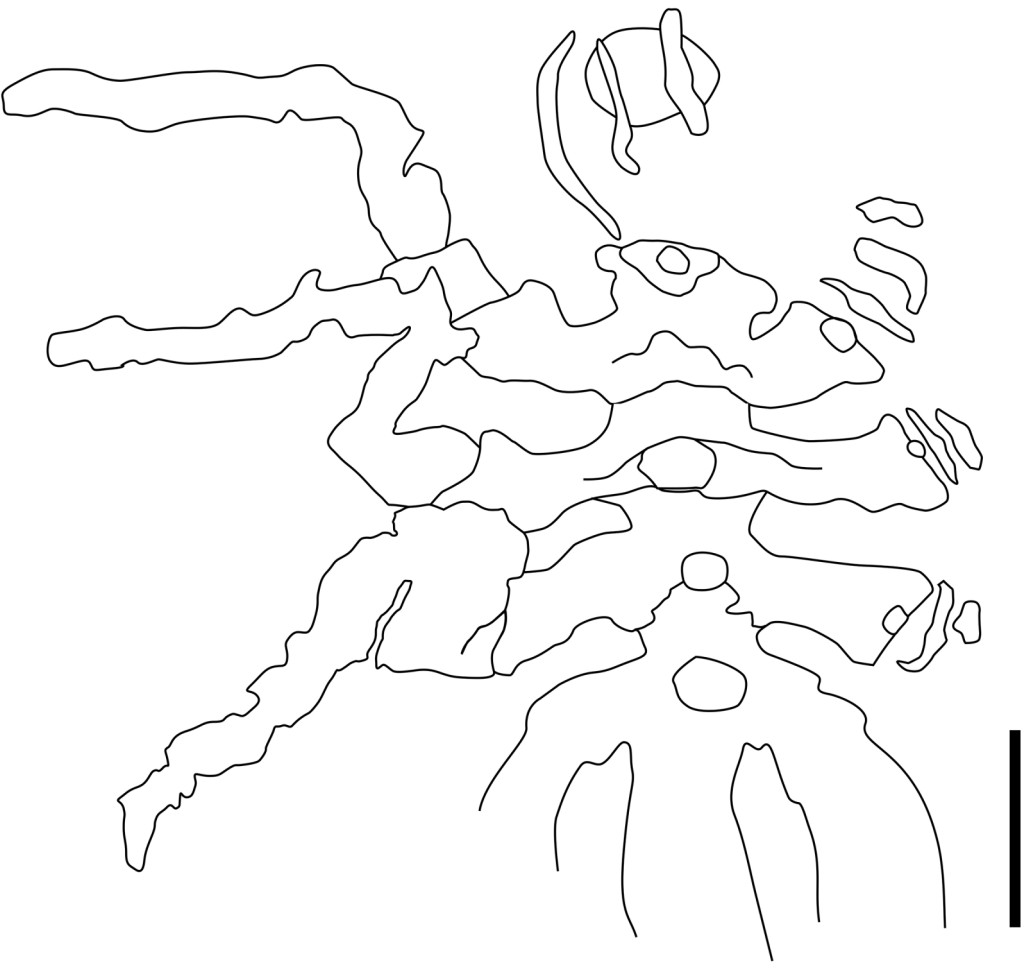

**Figure 35 Interpretative drawing of the holotype of *Palaeothea devonica*.** Drawing based on the X-ray images provided by *Bergström, Stürmer & Winter (1980)*. Scale bar 1 mm.

annulations at the leg bases (in *F. blocki*). It is also worth noting that the abdomen of *P. devonica* could also have the same dimensions as the massive abdomen of *F. blocki* or the specimen IGPD-HS437, although no unequivocal conclusion can be drawn from available illustrations. Perhaps *P. devonica* was of the same species as either one or the other of these two pycnogonids, though its smaller size would imply it was a juvenile or a different sex. Perhaps it could be another, third species. Unfortunately, these hypotheses cannot be further addressed in the absence of the holotype of *P. devonica*.

## Variability of the Pycnogonida walking leg ground plan

The most emblematic characteristic of modern sea spiders that makes them (almost) unmistakable in the field is their slender morphology with a narrow, often inconspicuous body onto which long legs articulate. This typical aspect has led to the sea spiders being coined as "no-bodies" (*e.g.*, *Brenneis et al., 2017*), or pantopods (Pantopoda Gerstäcker, 1863; literally "all made of legs"). Therefore, including pantopods within a broader phylogenetic framework among Pycnogonida and Arthropoda in general, involves a

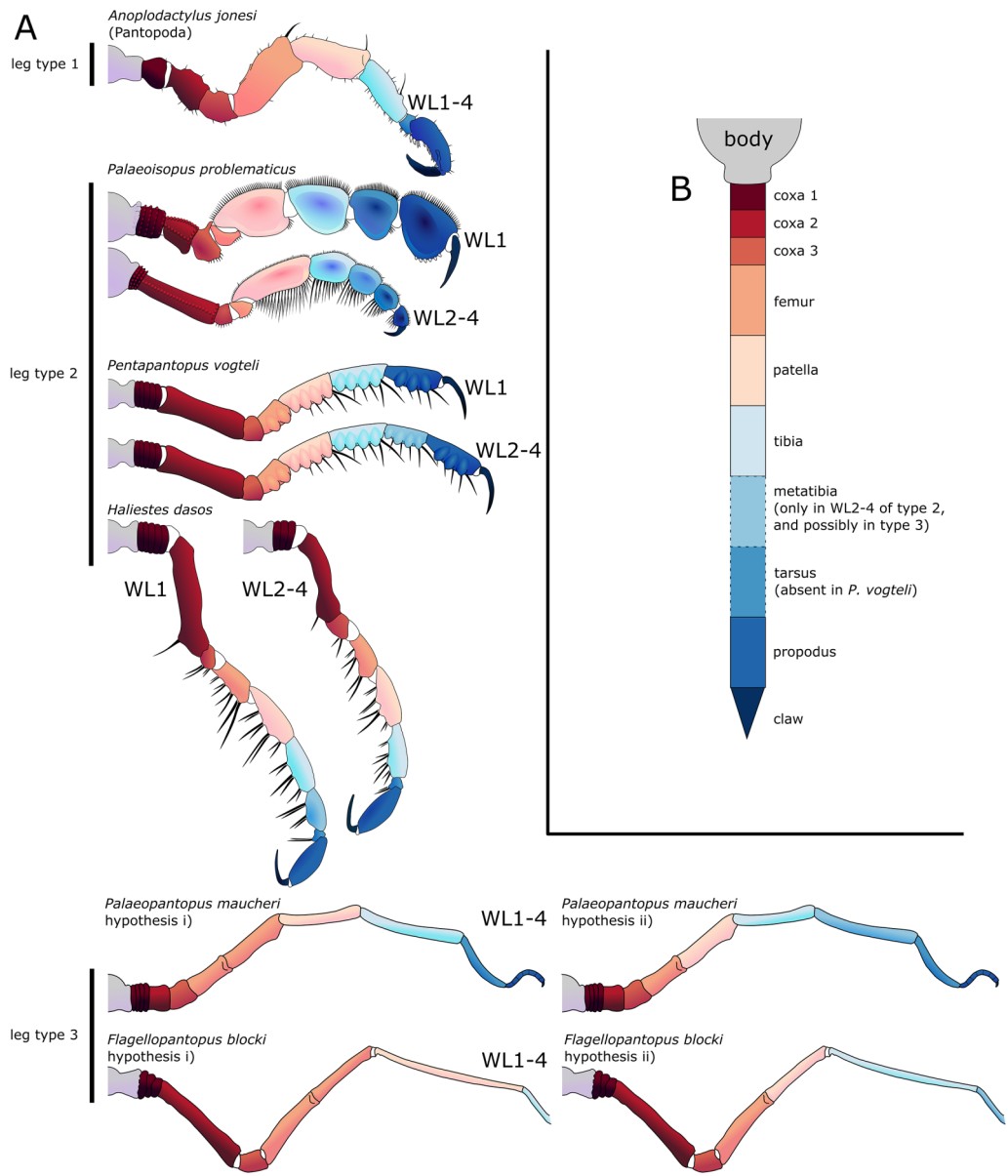

**Figure 36 Walking legs of sea spiders and hypothetical homologies among podomeres.** (A) Reconstruction and hypothesised homology (see text and Table 2 for explanations). (B) Colour code, following the nomenclature of *Sabroux et al. (2023)* and *Siveter et al. (2023)*. Figure modified from *Sabroux et al. (2023)* and *Siveter et al. (2023)*.

thorough comparison of the appendages. In most cases, the Hunsrück sea spiders have poorly preserved cephalic appendages; in contrast, walking (or swimming) legs were more massive and therefore often better preserved, sometimes exquisitely.

### Leg type 1: Pantopoda

The leg pattern of modern sea spiders (as, presumably, in all pantopods) is very stable (Fig. 36), with rare exceptions such as the parasitoid *Nymphonella tapetis*. From proximal to distal, the WL are composed of three short coxae; three long podomeres: femur, patella,

**Table 2 Characteristics of the three leg types in Pycnogonida.** Leg type 3 is presented following two alternative hypotheses (i) and (ii). See also Fig. 36.

| Leg characters | Leg type 1 | Leg type 2 | Leg type 3 (hypothesis i) | Leg type 3 (hypothesis ii) |
| --- | --- | --- | --- | --- |
| Coxa 1 | Simple | Annulated | Annulated | Annulated |
| Femur | Simple | Simple | Divided | Simple |
| Femur and patella | Mobile | Mobile | Mobile | Immobile? |
| Metatibia | Absent | Present in swimming legs 2–4 | Absent | Present in all walking legs |
| Rows of long ventral setae | Absent | Present | Absent | Absent |
| Transversal section | Cylindrical | Laterally flattened | Cylindrical | Cylindrical |

tibia (more often referred to femur, tibia 1 and tibia 2 in taxonomic literature); a variably elongated tarsus and propodus; and a terminal claw often flanked by two auxiliary claws. Hunsrück fossils show that this ground plan (to which we will refer to as leg type 1; Table 2) has not always been the sole leg pattern for sea spiders.

The most conspicuous and constant feature that differentiates Hunsrück fossils from leg type 1 is the annular structure at the base of legs. It was identified in *Palaeoisopus problematicus* and *Palaeopantopus maucheri* (*e.g.*, *Bergström, Stürmer & Winter, 1980*) and we also show clear evidence of its presence in *Flagellopantopus blocki* and *Pentapantopus vogteli*. As explained above, we suspect that *Palaeothea devonica* also had basal annulations. This structure is also observed at the base of the legs of *Haliestes dasos* (*Siveter et al., 2023*). *Siveter et al. (2023)* argued that these rings are likely homologous with the coxae 1 of leg type 1, relying on two arguments: 1) the position of legs in *H. dasos* suggests that the joint between the purported coxa 1 and 2 of this species has a promotor-remotor swing, which is typical of the coxa 1-coxa 2 articulation (*Manton, 1978*)–note that the Hunsrück material does not provide further evidence to support this observation; and 2) that the following podomere could be identified as coxa 2 through a ventrodistal protuberance that is typical of coxa 2 of extant pantopods, where the gonopores open. While *Bergström, Stürmer & Winter (1980)* interpreted the annulations at the base of legs of *P. problematicus* the same way as *Siveter et al. (2023)*, they proposed something different for *P. maucheri*, namely that they were "without doubt" part of the lateral processes seen also in *P. maucheri*. This hypothesis relies on the absence of lateral processes, an observation which we do not accept, as detailed above. We therefore regard the coxal rings as homologous among all the sea spiders that present them.

### Leg type 2: Palaeoisopus, Pentapantopus and Haliestes

We identify two types of legs among the Palaeozoic pycnogonids that both share the coxae 1 rings. The second pattern (leg type 2; see Table 2) is observed in *P. problematicus*. The paddle-like morphology of the walking legs in *P. problematicus* (Figs. 15, 36) suggests they were adapted for swimming. Conveniently, WL1 in *P. problematicus* had the same number of podomeres as pantopods, allowing us to propose an easily drawn podomere-by-podomere homology hypothesis (see also *Siveter et al., 2023*). Following this hypothesis,

the femora of *P. problematicus* are conspicuously shorter and compact compared to those of Pantopoda. The patella is the most elongated podomere, and the tibia, tarsus and propodus are similar in length and shape. The laterally flattened shape of the patella, tibia, tarsus and propodus in this fossil gives the leg its characteristic paddle-like structure. The propodus is followed by a lone, subchelate terminal claw (perhaps closing on the indentations on the opposite margin of the propodus). *Siveter et al. (2023)* showed a very similar pattern in *H. dasos* (Fig. 36), of which WL1 only differs from *P. problematicus* in that i) its transverse section is rather elliptic and not as strongly laterally compressed as in *P. problematicus*, ii) the femora are more elongated, though they remain short relative to the patella and tibia, and iii) the tarsus is reduced to a small podomere, similar to many of the extant sea spiders.

In WL2-4 of *P. problematicus* (as in *H. dasos*), there is one additional podomere (Figs. 16, 36). Since this additional podomere is "paddled", it must be somewhere between the patella and the propodus. *Siveter et al. (2023)* suggested based on length comparison that this additional podomere is to be found in *H. dasos* between the tibia and the tarsus, naming it the "metatibia". They suggested that the metatibia was also present in *P. problematicus*.

*P. vogteli* also presents a different number of podomeres between WL1 and WL2-4. Like *P. problematicus* and *H. dasos*, the legs are laterally flattened (Figs. 28, 29, 36), and the terminal claw is subchelate. Finally, these three fossils all have setae on the ventral surface of the legs; the distribution of these setae is sparse in *P. vogteli*, like in *H. dasos*. A fundamental difference with these two other fossils is that *P. vogteli* has one less podomere on all legs, *i.e.*, eight in WL1, and nine in WL2-4. The fact that the number of podomeres varies between the first and the following pairs of WLs suggests that *P. vogteli* also has metatibiae in WL2-4; so that a podomere is lacking from the overall count. This could be the tarsus: as this podomere is greatly reduced in *H. dasos* compared to *P. problematicus*, a possible scenario (Fig. 36) is that this podomere was progressively reduced and then completely lost/incorporated in *P. vogteli*.

### Leg type 3: Palaeopantopus and Flagellopantopus (and Palaeothea?)

At first glance, leg type 3 (see Table 2) found in *P. maucheri* and *F. blocki* appears to represent an intermediary state between type 1 and type 2, including both annulated coxae and cylindrical legs but the situation is more complex. It is difficult to propose a hypothesis of homology between leg podomeres of type 3 and those of types 1 or 2. In *P. maucheri* (Fig. 23), the number of podomeres exceeds 10. The terminal part of the legs seems to be composed of a series of small podomeres, at least four, possibly six (it is unclear whether this number varies in the different pairs of legs), in addition to the terminal claw. More proximally, there is at least one more "long" podomere in comparison to Pantopoda. Two of these long podomeres are most often aligned, with no evidence for mobility at the joint. This characteristic feature is also observed in *F. blocki* (Fig. 25), suggesting a podomere-by-podomere homology (Fig. 36). The terminal part of the legs of *F. blocki* is lacking, but based on this hypothesis it is possible to estimate the number of missing podomeres in that species.

We have considered two hypotheses of homology between these limbs and the other leg types. The first relies on the homology between the ventral and dorsal geniculations in leg types 1 and 3. The two aligned podomeres formed, in this case, a "divided" femur. It is followed by the patella, the tibia, and an elongated tarsus. The propodus would then be divided into at least four small podomeres and is followed by a terminal claw.

The second hypothesis of homology is based on inferring these aligned podomeres as two reflecting the plesiomorphic segmentation of the limb articles and assumes that the small terminal podomeres represent a divided propodus. Under this hypothesis, the number of podomeres is the same as the WL of type 2, suggesting the presence of a metatibia. Interestingly, there is no difference in the number of podomeres between WL1 and WL2-4, in contrast to leg type 2. This suggests that if there is a metatibia, it might not be homologous to that found in *P. problematicus*. This hypothesis is therefore less parsimonious than the first.

The situation found in *Palaeothea devonica* cannot be addressed with certainty given the dearth of information we have on the legs of this species. The presence of annulated coxae 1 and legs that appear cylindrical allow us to speculate that the limbs of this species are most likely to be of type 3. Similarly, the leg of the specimen IGPB-HS437 could correspond to type 3, but there is no clear evidence of the coxal rings nor of the division of the femur/additional WL podomere as observed in other species.

## Affinities of the Hunsrück sea spiders

Sea spiders have been classified in up to four orders: Palaeoisopoda, Palaeopantopoda, Nectopantopoda and Pantopoda (see Table 1; *Hedgpeth, 1954*, *1978*; *Bamber, 2007*). *Hedgpeth (1954)* included extant species in Pantopoda, a name that was until that time used as a synonym of the class Pycnogonida. The order Palaeopantopoda was named some 25 years before by *Broili (1930)* for the species *Palaeopantopodus maucheri*. Later, when he recognised *Palaeoisopus problematicus* as a sea spider, he also placed this species in the order (*Broili, 1932*). *Hedgpeth (1978)* divided Palaeopantopoda into two infraorders, Palaeopantopodina *Broili, 1930* and Palaeoisopodina *Hedgpeth, 1978*, which were much later raised to the ordinal level (*i.e.*, Palaeopantopoda and Palaeoisopoda) by *Bamber (2007)*. *Bamber (2007)* also established a fourth, monospecific order, Nectopantopoda, for *Haliestes dasos*. This classification is still in use (see for example *Bamber, El Nagar & Arango, 2024*), but this reflects more the scarcity of attempts to review it than a real consensus.

This work demonstrates that *P. problematicus*, *P. maucheri* and *Flagellopantopus blocki* all had a long, segmented abdomen, with often (if not always) a terminal telson. We suggest this was also probably the case for *Palaeothea devonica* and *Pentapantopus vogteli*. This character is arguably plesiomorphic as it mirrors the state in all potential fossil outgroups and it is plesiomorphic to arthropods as a whole. Consequently, the reduced, unsegmented, telson-less abdomen of pantopods is a robust autapomorphy for the group. All fossils studied here presents the typical annulated coxae 1 (it is not unambiguously observed only in the specimen IGPB-HS437). This also clearly distinguishes these fossils

from Pantopoda, which have legs of type 1 (*i.e.*, without annulated coxae 1; see Table 2, Fig. 36).

Leg morphology allows us to identify two distinct groups among the Devonian fossils. On one hand are sea spiders that have laterally flattened WL with ventral setae and a first leg pair with a reduced number of podomeres, that probably served a raptorial function (leg type 2; see Table 2, Fig. 36). This group included *P. problematicus* and *P. vogteli*, to which we can also add the Silurian species *H. dasos*. On the other hand are *P. maucheri*, *F. blocki*, and perhaps *P. devonica*, that have cylindrical legs (leg type 3; see Table 2, Fig. 36) and a broad, segmented abdomen.

Whether these two groups represent clades, or whether one or both are grades, remains uncertain. The group of *P. problematicus* has a number of characters that have the potential to be synapomorphic. The paddle-like distal podomeres of legs are potentially one: there is a trend in extant chelicerates to have flattened legs, but in these sea spiders this condition affects the proximal podomeres rather than the distal ones. However, the condition of fossils is often unknown. Another candidate is found in the modified, potentially raptorial first legs, but this is not unique to these sea spiders (*e.g.*, the eurypterid families Megalograptidae and Mixopteridae have such limbs; *Schmidt et al., 2022*). Similarly, the metatibia has the potential to be a synapomorphy, but the podomere composition and homology of many ancient chelicerates remains poorly understood. The group of *P. maucheri* is unambiguously characterised only by cylindrical legs and the annulated coxae 1, which are characters found in the two other Pycnogonida groups. The super-division of the propodus can only be identified in *P. maucheri*, and the division of the femur into two immobile podomeres is putative.

## CONCLUSION

The pycnogonids of the Hunsrück Slate are a unique testimony of the Palaeozoic diversity among Pycnogonida. The four species we redescribe, *Flagellopantopus blocki*, *Palaeoisopus problematicus*, *Palaeopantopus maucheri*, and *Pentapantopus vogteli*, show different morphologies within their body and the legs, suggesting two morphological groups (potentially corresponding to two clades) that surely had different biologies and ecologies. Unidentified specimens leave the possibility of even more diversity. None of these two groups have survived until today, where only pantopod pycnogonids can be observed.

In their tree-dating analysis, *Ballesteros et al. (2021)* estimated that crown-group Pantopoda diversified in the Silurian (at the latest) and that by the Late Devonian, all extant pantopod super-families listed by *Sabroux, Corbari & Hassanin (2023)* had already diversified. However, there is no trace of any of these taxa in the Devonian material or before, nor throughout the Devonian-Jurassic gap in the pycnogonid fossil record that ends with the faunas of La Voulte-sur-Rhône and Solnhofen, in which pantopods are the only representatives of Pycnogonida (*Charbonnier, Vannier & Riou, 2007*; *Sabroux et al., 2019*, *2023*). It remains uncertain whether the absence of pantopods among the rich Hunsrück material is a consequence of biased preservation (given that this material is primarily represented by the gigantic *P. problematicus*, and smaller forms are rarely found). Alternatively, or additionally, it is possible that pantopods inhabited different

environments (*e.g.*, abyssal waters) from the Hunsrück species. Lastly, we must consider that the diversification times calculated by *Ballesteros et al. (2021)* were based upon the then contemporary taxonomy and since then *Haliestes dasos* has been removed from the pantopod crown group (*Siveter et al., 2023*; *Sabroux et al., 2023*). This taxonomic revision will, inevitably, impact upon molecular clock estimates, including those of *Ballesteros et al. (2021)*.

## ACKNOWLEDGEMENTS

We are grateful to Manuela Aiglstorfer (Naturhistorisches Museum Mainz), Andreas Abele-Rassuly (Museum für Naturkunde, Berlin), Georg Heumann (Institut für Geowissenschaften, Bonn), Martin Nose and Mike Reich (Bayerische Staatssammlung für Paläontologie und Geologie München) for providing access and allowing loan of the studied material; to Kristin Mahlow, Alice Petzold and Marcel Gärtner (Museum für Naturkunde), Liz Martin-Silverstone (University of Bristol) and Julia A. Schultz (Institut für Geowissenschaften, Bonn) for their help in the acquisition and management of X-ray microtomography data; to Jakob Vinther (University of Bristol) for providing the RTI dome and photographic equipment; and to Gabriele Kühl for her comments on an early version of the draft. We are also grateful to Kenneth De Baets (Universytet Warszawki), Jason A. Dunlop (Museum für Naturkunde) and Lorenzo Lustri (Université de Lausanne) for their insightful reviews of the manuscript. Finally, this work could not have been achieved without the invaluable help of the late Christoph Bartels, who provided access to and loan of his precious collection (curated at the Institut für Geowissenschaften, University of Bonn; collection numbers IGPB-HS), hosted RS while visiting, and gave much valued advice during their examination.

### Funding

This project has received funding from the European Union's Horizon 2020 research and innovation programme under the Marie Skłodowska-Curie grant agreement No. 101023218 to RS; from the Leverhulme Trust Research Fellowship (RF-2022-167) to PCJD; from a University of Bristol University Research Fellowship (URF) to DP; and from the Natural Environment Research Council (grant NE/T000813/1), the Alexander von Humboldt Stiftung, and the Leverhulme Trust Research Project Grant (RPG-2023-234) to RJG. DP and PCJD were conjointly funded by the Natural Environment Research Council (grant NE/P013678/1) as part of the Biosphere Evolution, Transitions and Resilience programme co-funded by the Natural Science Foundation of China; by the John Templeton Foundation (62220; the opinions expressed in this publication are those of the authors and do not necessarily reflect the views of the John Templeton Foundation), and by a Gordon and Betty Moore Foundation (GBMF9741) grant. The funders had no role in study design, data collection and analysis, decision to publish, or preparation of the manuscript.

## Grant Disclosures

The following grant information was disclosed by the authors:

European Union's Horizon 2020 research and innovation programme under the Marie Skłodowska-Curie: 101023218.

Leverhulme Trust Research Fellowship: RF-2022-167.

University of Bristol University Research Fellowship (URF).

Natural Environment Research Council: NE/T000813/1.

Alexander von Humboldt Stiftung.

Leverhulme Trust Research Project: RPG-2023-234.

Natural Environment Research Council: NE/P013678/1.

Biosphere Evolution, Transitions and Resilience programme co-funded by the Natural Science Foundation of China.

John Templeton Foundation: 62220.

Gordon and Betty Moore Foundation: GBMF9741.

## Competing Interests

The authors declare that they have no competing interests.

## Author Contributions

- Romain Sabroux conceived and designed the experiments, performed the experiments, analyzed the data, prepared figures and/or tables, authored or reviewed drafts of the article, and approved the final draft.
- Russell J. Garwood conceived and designed the experiments, analyzed the data, authored or reviewed drafts of the article, and approved the final draft.
- Davide Pisani conceived and designed the experiments, authored or reviewed drafts of the article, and approved the final draft.
- Philip C. J. Donoghue conceived and designed the experiments, authored or reviewed drafts of the article, and approved the final draft.
- Gregory D. Edgecombe conceived and designed the experiments, analyzed the data, authored or reviewed drafts of the article, and approved the final draft.

## Data Availability

The RTI raw data, CT-scan raw data and RTI files are available at the University of Bristol's data.bris Research Data Repository:

Romain Sabroux, Russell Garwood, Davide Pisani, Philip Donoghue, Gregory D. Edgecombe (2024): New insights into the Devonian sea spiders of the Hunsrück Slate (Arthropoda: Pycnogonida). https://doi.org/10.5523/bris.d0qlgy38h4fq2t6p0hapiysh2.

## Supplemental Information

Supplemental information for this article can be found online at http://dx.doi.org/10.7717/peerj.17766#supplemental-information.

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
