# Peer review of "New insights into the Devonian sea spiders of the Hunsrück Slate (Arthropoda: Pycnogonida)"

_PeerJ, doi:10.7717/peerj.17766_

## Round 0.1 · original submission · Minor Revisions

You provide a spectacular revision of the Hunsrück Slate pycnogonids supported by the latest imaging techniques. This will be of great relevance to the community, paleontology, and other disciplines more generally. I apologize for the delay in making my decision but given the length of the manuscript and difficulties in finding reviewers to review on short notice, it was beyond my control. It would love to see this published, but some crucial points need to be addressed before publication:

Use of Hunsrück Slate: many authors still informally use the Hunsrück slate to refer to slate unit containing stratigraphic levels of exceptionally preservation within the Central Hunsrück Basin (Bundenbach – Herrstein – Gemünden area) but this is not a stratigraphic unit (compare Wilkin 2023). Furthermore, it has generally been inconsistently used. Some use it to refer to all slates of Pragian to Emsian ages within the area, while other consistently use it to refer to the slates from Central Hunsrück basin. The slate unit in this area does not just contain such levels of exceptional preservation but also units of background sedimentation. The entire slate unit containing those levels of exceptional preservation is now assigned to the middle Kaub Formation (e.g., Schindler et al. 2002). Based on ammonoid and dacryoconarid biostratigraphy, these levels are now consistently assigned to the lower Emsian (De Baets et al. 2013). Given the preservation and the fact that at least some of the specimens can be confidently assigned to the Wingertshell member at the Obereschenbach quarry (compare Poschmann & Dunlop 2002; Kühl et al. 2013). I assume most if not all your specimens derived from middle Kaub Formation and have an early Emsian age. Please discuss these aspects and specify the stratigraphic unit and age of the units in the Central Hunsrück basin relevant to you. The use of Hunsrück Slate is ok but should be clarified. Given you mention the importance for calibrating the evolutionary history of sea spiders, appropriate age assignment of these specimens is even more important.

Palaeoisopus: Please verify your statement about fusion of telson to a preanal segment and discuss include a discussion of abdominal segmentation more generally (see reviewer 1 and compare Vilpoux & Waloszek 2003).

Ecology: Please consider pointing out that multiple modern sea spider seems to have a parasitic life stage (e.g., Brenneis et al. 2017; Dunlop 2021) and how its appearance could have impacted their preservation potential. Interestingly, it seems that already at the time of the Hunsrück Slate some pycnogonids may have had symbioses or repeated interactions with other taxa.

Taphonomic scenario: I feel your discussion on taphonomy would benefit from included information on the non-exceptional “background” faunal. There is a considerable background sedimentation which often yields ammonoids and shelled organisms more generally but even those “background” organisms are occasionally exceptionally preserved within those intervals yielding exceptional preservation such as preservation of fragile whorls or embryonic shell (compare Sutcliffe 1997; Sutcliffe et al. 1999; De Baets et al. 2013). The rarity of exceptional preserved ammonoids or lack of soft-tissue preservation maybe corroborate your scenario for the rarity of pelagic sea spiders. Other biases might have exacerbated these patterns. Klug et al. (2008) did not explicitly compare size between Hunsrück Slate and Moroccan lagerstätten of similar age and environments but when looking at common elements, they are usually larger in the Hunsrück slate and higher proportion of shelled organisms (e.g., mollusks) described in the Moroccan lagerstätten – likely related to more 3D-preservation and shells details mostly lacking from the Hunsrück Slate specimens. Please consider discussing in greater detail the issues with interpreting fossil structures X-ray analyses alone (e.g., Otto 1999, 2000 and references therein).

Provide locality information: I am a bit surprised about the missing locality information for all specimens. I assume most derive from the Central Hunsrück basin (Bundenbach-Gemünden area). From the preservation alone, I assume they derive from the middle Kaub Formation which is confirmed at least for some specimens by cross-referencing previously published one (e.g., Wingertshell member of the Obereschenbach quarry in Poschmann and Dunlop 2006; Kühl et al. 2013 – both of which you cited already). Also, Broili mentions Bundenbach or Gemünden for the provenance of his specimens. I assume for most that locality information will be available and should therefore be added in systematic paleontology section and (at minimum) in the supplementary table.

Restructure systematic descriptions: in addition to adding geographic and stratigraphic information, I agree with reviewer 2 that the descriptions contain information that should be moved into the discussion or a remarks section. Descriptions should include only the observed morphology, while interpretations and explanations relying on the literature should be moved to the remarks section (or alternatively the discussion). This would make the paper easier to read. Please make sure all museum abbreviations are included – also those listed in supplementary materials (compare reviewer 1). Please consider providing an amended diagnosis for some of the re-described species (compare reviewer 2).

Homogenise synonymy lists: I agree with reviewer 1, that there are some inconsistencies in the synonymy list which need to be resolved. Make sure, they are consistent throughout.

Add more labels and figure calls in the description: I agree with reviewer 2 that the manuscript would benefit from referring more often and more precisely to figures as adding more labels.

Formatting and terminology issues: there are some phrases which could rephrased for clarity (see suggestions by reviewer 1 and 2); also, there are some typographical issues ( Bundenbach and Bundenbachia instead of Budenbach and Budenbachia; see annotated pdf) as well as potential issues which could lead to confusion (e.g., line 155: use of plate; compare reviewer 1).

Raw data availability: the raw image stack and ideally also the finished models in readable formats should be made available in appropriate repositories and explicitly state with digital object identifiers in the finished manuscript at the latest upon publication (e.g., Davies et al. 2017). Currently you only mention a link in the reviewing materials, but I was unable to access its content.

Please make sure these points as well as all other points raised including those in annotated pdfs are addressed. I look forward to receiving the revised manuscript and seeing it published.


Suggested references:

Brenneis, G., Bogomolova, E. V., Arango, C. P., & Krapp, F. (2017). From egg to “no-body”: an overview and revision of developmental pathways in the ancient arthropod lineage Pycnogonida. Frontiers in Zoology, 14, 1-22.

De Baets, K., Klug, C., Korn, D., Bartels, C., & Poschmann, M. (2013). Emsian Ammonoidea and the age of the Hunsrück Slate (Rhenish Mountains, Western Germany). Palaeontographica A, 299(1-6), 1-113.

Dunlop, J. A. (2021). Chelicerates as parasites. In The Evolution and Fossil Record of Parasitism: Identification and Macroevolution of Parasites (pp. 315-346). Cham: Springer International Publishing.

Klug, C., Kröger, B., Korn, D., Ruecklin, M., Schemm-Gregory, M., De Baets, K., & Mapes, R. H. (2008). Ecological change during the early Emsian (Devonian) in the Tafilalt (Morocco), the origin of the Ammonoidea, and the first African pyrgocystid edrioasteroids, machaerids and phyllocarids. Palaeontographica abteilung a-Palaozoologie-Stratigraphie, 283(4-6), 83-U58.

Otto, M. (1999). The problem of fossil interpretation. Lethaia, 32(2), 140-142.

Otto, M. (2000). Supposed soft tissue preservation in the Hunsrückschiefer (Lower Devonian, Rheinisches Schiefergebirge): the example of brachiopods. Paläontologische Zeitschrift, 74(1), 79-89.

Schindler, T., Sutcliffe, O.E., Bartels, C., Poschmann, M. & Wuttke, M. (2002): Lithostratigraphic subdivision and chronostratigraphic position of the middle Kaub Formation (Lower Emsian, Lower Devonian) of the Bundenbach area (Hunsrück, SW Germany).– Metalla 9 (2): 7388.

Sutcliffe, O.E. (1997): The sedimentology and ichnofauna of the Lower Devonian Hunsrück Slate, Germany: taphonomy and palaeobiological significance.– Unpubl. PhD thesis, Univ. Bristol, 297 pp.

Sutcliffe, O. E., Briggs, D. E., & Bartels, C. (1999). Ichnological evidence for the environmental setting of the Fossil-Lagerstatten in the Devonian Hunsruck Slate, Germany. Geology, 27(3), 275-278.

Vilpoux, K. & Waloszek, D. (2003). Larval development and morphogenesis of the sea spider Pycnogonum litorale (Ström, 1762) and the tagmosis of the body of Pantopoda. Arthropod Structure & Development, 32(4), 349-383.

Wilkin, J. T. (2023). The Hunsrück Slate Konservat‐Lagerstätte. Geology Today, 39(2), 72-76.


·

Basic reporting

This is going to be an important manuscript offering a thorough and comprehensive revision of the Devonian sea spiders from the famous Hunsrück Slate of Germany. As the authors correctly note, this is the most diverse Palaeozoic fauna of these unusual arthropods and offers a unique insight into the early evolution of this enigmatic arthropod group - including extinct taxa with body plans quite different from the living species. The authors are experienced researchers into fossil and Recent arthropods as well as evolutionary processes in the fossil record. The manuscript has been prepared to a very high standard with a generous number of high-quality illustrations. The text is well written and the manuscript is structured appropriately. I would strongly encourage acceptance subject to MINOR REVISION (see below).

Experimental design

The maunuscript does not involve experimental work per se, but the methods of study are appropriate and the authors have made some effort to examine and figure much of the available material using a variety of visualisation techniques, including Reflectance Transformation Imaging and computer tomography. The figures are thus excellent and certainly merit inclusion. Most of the specimens are historical and thus already available in public collections which would allow the work to be replicated. Overall, I believe the topic is appropriate for PeerJ. The aims of the study are straightforward, i.e. redescribe and revise the fossils and attempt to infer their palaeoecology and phylogenetic position, and their results integrate nicely into broader advances in our knowledge of early arthropods and the particularly thorny question of where the chelicerates (arachnids and their relatives) evolved from.

Validity of the findings

The authors' results appear to represent valid hypotheses and are well supported by their data, including comparative information from living species. The authors offer several novel interpretations of the morphology of some of these extinct sea spiders, which is of significance for understanding their evolutionary position, and argue (I think correctly) that the modern crown-group Pantopoda was not present among this early species assemblage. As the authors state, this will impact on the use of the Hunsrück fossils as calibration points for molecular dating. The reconstructions of their probable appearance are also nice and add to the wider picture of the life in the Hunsrück marine ecosystem. I particularly like the colour-coded limb drawings, which allow the reader to better understand the authors' interpretations of homologies between the leg segments.

Additional comments

ABSTRACT

line 18: I'm not quite sure what you mean by "timescale pycnogonid evolutionary history"? Do you mean their evolution in deep time and/or a time calibrated phylogeny? Maybe rephrase this?


INTRODUCTION

line 79: better "Five species have been recorded..." ?

line 188: better "If it could be demonstrated that..." ?

line 120: either "on the calibration of sea spider evolution to time" or "on the calibration of the sea spiders' evolution to time".

line 122: "in informing the taxonomy of fossil sea spiders which is poorly developed." reads a little strange, maybe in "resolving" or "understanding" the taxonomy ?


GEOLOGICAL SETTING

line 138: I think a better translation is "into the Rhineland-Palatinate fedral state" (i.e. a Bundesland in German)

line 147: not sure "qualitify" is a real word, do you mean "qualify" ?

line 150: better "...that were later exploited by people as early as the Roman..."


MATERIALS AND METHODS

line 155: I wonder if "...studied in detail 46 slabs..." would be better, "plate" makes me think of the photographic X-ray plates which were used in some early Hunsrück studies; see also line 177 where you do use slab.


RESULTS

lines 210-230: there are some inconsistencies in style in the synonymy list, especially. e.g. "pp 45-54" (gap before the page number), "pp70, 71" (no gap before the page number and a comma) and, "197-199" ("p." or "pp." not used). Please check the journal's recommended house style for synonymy lists and make sure this is applied consistently throughout for all the species.

line 222: here, for example, you have the page numbers before the author name.

line 240: would it be better to say "reaching up to 400 mm in legspan"; the actual body is much smaller.

line 244: I'm not sure if this is an issue here, but if any of the new material is not from the institutes you listed under Materials and Methods then you might want to add additional collections to lines 163-168. Not everyone looks at the supplementary info in detail.

line 253: better "...to which the palps articulate..." ?

line 255: "...on which the chelifores articulate..." (not articulates)

line 280: qualify "...in the living genus Rhynchothorax..."

line 303-340: Can you be certain that the Palaeoisopus telson is fused to a preanal segment? I have the feeling that Dieter Waloszek suggested that in Palaeoisopus the anus uniquely (or plesiomorphically) opened within the telson, but I can't recall if this was pers. comm. or formally published. In any case, in a discussion of abdominal segmentation (see also lines 850-860) I would have expected you to include:

Vilpoux, K. & Waloszek, D. (2003). Larval development and morphogenesis of the sea spider Pycnogonum litorale (Ström, 1762) and the tagmosis of the body of Pantopoda. Arthropod Structure & Development, 32(4), 349-383.

specifically their comparative figure 13 which draws heavily on Palaeoisopus and compares it to other arthropods.

line 329: "...more distally, is another..." (not iss)

line 343: "...typically has a dorsal spur..."

line 376 "(e.g., Sabroux et al., 2023)" delete unnecessary ( )

line 393: "...the distal claw of the legs..."

line 524: "...is, as far as we know, represented by..."

line 535: "...How the proboscis inserted on the head..."

line 595: "...left palp of the new specimen (fig. 29) potentially bears a terminal..."

line 621: "Its legs are not preserved..." (space missing)

line 644: "...margins of the trunk segments widen..."

line 669: " Two of them were already described..."

line 687 "...(figs 1, 17..." (space missing)

line 758: "...at the base of the legs of..."

line 761: "...coxa 1-coxa 2..." (space missing)

line 799: "...also had metatibiae in WL2-4..."? (which plural of tibia are you using?)


References

line 930: style, should it be "A critical appraisal of the placement of Xiphosura (Chelicerata) with account of known sources of phylogenetic error" ? (i.e. in don't capitalise all the words in the title)

line 935: "2023. PycnoBase" (space missing)

line 937: "The fossils of the Hunsrück Slate" ( S missing)

line 952: I think it should be "Frontiers in Zoology 14:6" and not just 14. The suggested citation from the journal publisher is "Front Zool 14, 6"

lines 262/264: there is a slight discrepency in the title of the journal (the second one includes "zu München"). Is this correct, i.e. the formal name changed slightly? Also the first one says "Mathematisch-Naturwissenschaftlichen" and the second one "mathematisch-naturwissenschaftlichen"? My limited German suggests the second one is probably correct as its being used as an adjective here, but please check the original citations. See also the 1933 reference.

line 977: italicise the species name "Ginkgophyllum grassetii"

line 1028: "An Ordovician pycnogonid (sea spider) with serially subdivided 'head' region."

line 1059: should be "On the relationships and phylogeny of fossil and Recent Arachnomorpha"


Figure legends

Fig. 11: "White arrowheads..."; PS I can't actually see any arrowheads on the figure itself.


Tables

Table 2: under leg type 2, better "laterally flattened" (not laterally flatten)?


Jason A. Dunlop
Berlin

·

Basic reporting

no comment

Experimental design

no comment.

Validity of the findings

no comment

Additional comments

The paper re-describes the species of pycnogonids of the Hunsrück slate. The authors utilize a mix of photographic and x-ray microtomography to reconstruct and describe the anatomies of the studied species. Beautiful schematic reconstructions of the anatomy are also provided. I have enjoyed the reading throughout, and this kind of in-depth description of the diversity of a group in a specific site is highly valuable to the paleontological community. This paper may easily be the "go-to paper" for researchers approaching the paleontology of pycnogonids for the first time. This is the reason why I suggest the author carefully consider two aspects of the paper that could increase the clarity of the paper.

1) Almost every description of the examined material incorporates considerations that should either be moved into the discussion or a remarks section. I personally suggest adding a remarks section for each species and moving the more discursive parts of the description into them, letting the description include only the observed morphology, devoid of interpretation and explanation relying on the literature. In my opinion, this will make the reading of such a huge paper more fluid.
2) Not enough labels and figure calls are present in the description. Again, this is a huge manuscript, and an extra effort to help the reader should be made. I suggest calling the figure more often during the description (possibly the figure number and letter and not only the figure number) and also inserting more labels since many figures are devoid of them.

Less important considerations are: a) Is there a need to do an amended diagnosis for some of the re-described species? b) You may want to add a systematic framework section.
I have attached a PDF with some examples of points 1 and 2 plus some other minor corrections.

My suggestions do not regard the methods either the discussion and conclusions but are meant merely to improve the clarity of the MS. As such, I suggest minor edits, accepting the paper, and I do not wish to read the manuscript again before publication if the authors will make the suggested corrections.

---

## Round 0.2 · accepted · Accept

Thank you for addressing our suggestions. The updated information on localities, Palaeoisopus, stratigraphy and the taphonomic scenario makes the manuscript easier to follow and of even greater relevance. Thank you for explaining your choices in greater length in the rebuttal. I agree with reviewer 1 that your manuscript can be accepted for publication – pending you include an updated link to the raw data (repository) within the manuscript during the proofing phase and it becomes accessible at the latest publication. I look forward to seeing this published.

·

Basic reporting

n/a

Experimental design

n/a

Validity of the findings

n/a

Additional comments

The revised manuscript has addressed the points raised in the previous round of reviews, and I believe it is now ready for publication as a excellent contribution towards our understanding of fossil sea spiders.